

# The consolidated European synthesis of CH₄ and N₂O emissions for EU27 and UK: 1990-2018

Ana Maria Roxana Petrescu[1], Chunjing Qiu[2], Philippe Ciais[2], Rona L. Thompson[3], Philippe Peylin[2], Matthew J. McGrath[2], Efisio Solazzo[4], Greet Janssens-Maenhout[4,] Francesco N. Tubiello[5], Peter Bergamaschi[4], Dominik Brunner[6], Glen P. Peters[7], Lena Höglund-Isaksson[8], Pierre Regnier[9], Ronny Lauerwald[9,23], David Bastviken[10], Aki Tsuruta[11], Wilfried Winiwarter[8,12], Prabir K. Patra[13], Matthias Kuhnert[14], Gabriel D. Orregioni[4], Monica Crippa[4], Marielle Saunois[2], Lucia Perugini[15], Tiina Markkanen[11], Tuula Aalto[11], Christine D. Groot Zwaaftink[3], Yuanzhi Yao[16], Chris Wilson[17,18], Giulia Conchedda[5], Dirk Günther[19], Adrian Leip[4], Pete Smith[14], Jean-Matthieu Haussaire[6], Antti Leppänen[20], Alistair J. Manning[21], Joe McNorton[22], Patrick Brockmann[2] and Han Dolman[1]

[1]Department of Earth Sciences, Vrije Universiteit Amsterdam, 1081HV, Amsterdam, the Netherlands
[2]Laboratoire des Sciences du Climat et de l'Environnement, 91190 Gif-sur-Yvette, France
[3]Norwegian Institute for Air Research (NILU), Kjeller, Norway
[4]European Commission, Joint Research Centre, 21027 Ispra (Va), Italy
[5]Food and Agriculture Organization of the United Nations, Statistics Division. 00153 Rome, Italy.
[6]Empa, Swiss Federal Laboratories for Materials Science and Technology, 8600 Dübendorf, Switzerland
[7]CICERO Center for International Climate Research, Oslo, Norway
[8]International Institute for Applied Systems Analysis (IIASA), 2361 Laxenburg, Austria
[9]Biogeochemistry and Modeling of the Earth System, Université Libre de Bruxelles, 1050 Bruxelles, Belgium
[10]Department of Thematic Studies - Environmental Change, Linköping University Sweden
[11]Finnish Meteorological Institute, P. O. Box 503, FI-00101 Helsinki, Finland
[12]Institute of Environmental Engineering, University of Zielona Góra, Zielona Góra, 65-417, Poland
[13]Research Institute for Global Change, JAMSTEC, Yokohama 2360001, Japan
[14]Institute of Biological and Environmental Sciences, University of Aberdeen, 23 St Machar Drive, Aberdeen, AB24 3UU, UK
[15]Centro Euro-Mediterraneo sui Cambiamenti Climatici (CMCC), Viterbo, Italy
[16]International Centre for Climate and Global Change, School of Forestry and Wildlife Sciences, Auburn University, USA
[17]Institute for Climate and Atmospheric Science, University of Leeds, Leeds, UK.
[18]National Centre for Earth Observation, University of Leeds, Leeds, UK.
[19]Umweltbundesamt (UBA), 14193 Berlin, Germany
[20]University of Helsinki, Institute for Atmospheric and Earth System Research/Physics, Faculty of Science, 00560 Helsinki, Finland
[21]Hadley Centre, Met Office, Exeter, EX1 3PB, UK
[22]European Centre for Medium-Range Weather Forecasts (ECMWF), Reading, RG2 9AX, UK
[23]Université Paris-Saclay, INRAE, AgroParisTech, UMR ECOSYS, Thiverval-Grignon, France

*Correspondence to*: A.M. Roxana Petrescu (a.m.r.petrescu@vu.nl)

**Abstract**

Reliable quantification of the sources and sinks of greenhouse gases, together with trends and uncertainties, is essential to monitoring the progress in mitigating anthropogenic emissions under the Paris Agreement. This study provides a consolidated synthesis of CH₄ and N₂O emissions with consistently derived state-of-the-art bottom-up (BU) and top-down (TD) data sources for the European Union and UK (EU27+UK). We integrate recent emission inventory data, ecosystem process-based model results, and inverse modelling estimates over the period 1990-2018. BU and TD products are compared with European National GHG Inventories (NGHGI) reported to the UN climate convention secretariat UNFCCC in 2019. For uncertainties, we used for NGHGI the standard deviation obtained by varying parameters of inventory calculations, reported by the Member States following the IPCC guidelines recommendations. For atmospheric inversion models (TD) or other inventory datasets (BU), we defined uncertainties from the spread



between different model estimates or model specific uncertainties when reported. In comparing NGHGI with other approaches, a key source of bias is the activities included, e.g. anthropogenic versus anthropogenic plus natural fluxes. In inversions, the separation between anthropogenic and natural emissions is sensitive to the geospatial prior distribution of emissions. Over the 2011-2015 period, which is the common denominator of data availability between

all sources, the anthropogenic BU approaches are directly comparable, reporting mean emissions of 20.8 Tg $CH_4$ yr$^{-1}$ (EDGAR v5.0) and 19.0 Tg $CH_4$ yr$^{-1}$ (GAINS), consistent with the NGHGI estimates of 18.9 ± 1.7 Tg $CH_4$ yr$^{-1}$. TD total inversions estimates give higher emission estimates, as they also include natural emissions. Over the same period regional TD inversions with higher resolution atmospheric transport models give a mean emission of 28.8 Tg $CH_4$ yr$^{-1}$. Coarser resolution global TD inversions are consistent with regional TD inversions, for global inversions with

GOSAT satellite data (23.3 Tg $CH_4$ yr$^{-1}$) and surface network (24.4 Tg $CH_4$ yr$^{-1}$). The magnitude of natural peatland emissions from the JSBACH-HIMMELI model, natural rivers and lakes emissions and geological sources together account for the gap between NGHGI and inversions and account for 5.2 Tg $CH_4$ yr$^{-1}$. For $N_2O$ emissions, over the 2011-2015 period, both BU approaches (EDGAR v5.0 and GAINS) give a mean value of anthropogenic emissions of 0.8 and 0.9 Tg $N_2O$ yr$^{-1}$ respectively, agreeing with the NGHGI data (0.9 ± 0.6 Tg $N_2O$ yr$^{-1}$). Over the same period,

the average of the three total TD global and regional inversions was 1.3 ± 0.4 and 1.3 ± 0.1 Tg $N_2O$ yr$^{-1}$ respectively, compared to 0.9 Tg $N_2O$ yr$^{-1}$ from the BU data. The TU and BU comparison method defined in this study can be 'operationalized' for future yearly updates for the calculation of $CH_4$ and $N_2O$ budgets both at EU+UK scale and at national scale. The referenced datasets related to figures are visualized at https://doi.org/10.5281/zenodo.4288969 (Petrescu et al., 2020).

## 1. Introduction

The global atmospheric concentrations of methane ($CH_4$) has increased by 160% and that of nitrous oxide ($N_2O$) by 22% since the pre-industrial period (WMO, 2019) and are well documented as observed by long-term ice-core records (Etheridge et al., 1998, CSIRO). According to the NOAA atmospheric data

(https://www.esrl.noaa.gov/gmd/ccgg/trends_ch4/ last access June 2020) the $CH_4$ concentration in the atmosphere continues to increase and, after a small dip in 2017, has an average growth of 10 ppb / year, representing the highest rate observed since the 1980s[1] (Nisbet et al. 2016, 2019). This increase was attributed to anthropogenic emissions from agriculture (livestock enteric fermentation and rice cultivation) and fossil fuel related activities, combined with a contribution from natural tropical wetlands (Saunois et al., 2020, Thompson et al. 2018, Nisbet et al., 2019). The

recent increase in atmospheric $N_2O$ is more linked to agriculture in particular due to the application of nitrogen fertilizers and livestock manure on agricultural land (FAO, 2020, 2015; IPCC, 2019b, Tian et al., 2020).

National GHG inventories (NGHGI) are prepared and reported on annual basis by Annex I countries[2] based on IPCC Guidelines using national activity data and different levels of sophistication (tiers) for well-defined sectors.

---

[1] The 1980s rapid development of gas industry in former USSR.

[2] Annex I Parties include the industrialized countries that were members of the OECD (Organization for Economic Co-operation and Development) in 1992 plus countries with economies in transition (the EIT Parties), including the Russian Federation, the Baltic States, and several central and eastern European states (UNFCCC, https://unfccc.int/parties-observers, last access: February 2020).



These inventories contain annual time series of each country GHG emissions from the 1990 base year[3] until two years
before the year of reporting and were originally set to track progress towards their reduction targets under the Kyoto
Protocol (UNFCCC, 1997). Non-Annex I countries provide some information in Biennial Update Reports (BURs) as
well as National Communications (NCs), but neither BURs nor NCs report annual time series or use harmonized
formats. The IPCC tiers represent the level of sophistication used to estimate emissions, with Tier 1 based on global
or regional default values, Tier 2 based on country-specific parameters, and Tier 3 based on more detailed process-
level modelling. Uncertainties in NGHGI are calculated based on ranges in observed (or estimated) emission factors
and variability of activity data, using the error propagation method (95% confidence interval) or Monte-Carlo
methods, based on clear guidelines (IPCC, 2006).

NGHGIs follow principles of transparency, accuracy, consistency, completeness and comparability
(TACCC) under the guidance of the UNFCCC (UNFCCC, 2014). Methodological procedures are taken from the 2006
IPCC guidelines (IPCC, 2006). The IPCC 2019 Refinement (IPCC, 2019a), that may be used to complement the 2006
IPCC guidelines, has updated sectors with additional sources and provides guidance on the possible and voluntary use
of atmospheric data for independent verification of GHG inventories. Complementary to NGHGI, research groups
and international institutions produce estimates of national GHG emissions, with two families of approaches:
atmospheric inversions (Top-Down, TD) and GHG inventories based on the same principle that NGHGI but using
different methods and input data (Bottom-Up, BU). These complementary approaches are necessary. First, TD
approaches act as an independent check on BU approaches, and facilitate a deeper understanding of the scientific
processes driving different GHG budgets. Second, NGHGIs only cover a subset of countries, and it is therefore
necessary to construct BU estimates independently for all countries. The BU estimates are often used as priors for TD
estimates and to track emissions over time, either globally or country level, such as in the UNEP Emissions Gap
Report (UNEP, 2019). There is no guideline to estimate uncertainties in TD or BU approaches. The uncertainties are
usually assessed from the spread of different estimates within the same approach, though some groups or institutions
report uncertainties for their estimates using a variety of methods, for instance, by varying parameters or input data.
However, this gets complicated when dealing with complex process based models.

NGHGI official numbers are not always straightforward to compare with other independent estimates.
Independent estimates often have different system boundaries and a different focus. BU estimates often have a lot of
overlap in terms of methods and other input data, and through harmonization, the differences between BU estimates
and NGHGIs can be bridged. On the other hand, TD estimates are much more independent and provide the best
independent check on NGHGIs. While NGHGI goes through a review process, the UNFCCC procedures do not
incorporate mandatory independent, large-scale observation-derived verification, but allow the use of atmospheric
data for external checks within the data quality control, quality assurance and verification process (IPCC 2006
Guidelines, Chapter 6 QA/QC procedures). So far, only a few countries (e.g. Switzerland, UK, New Zealand and

---

[3] For most Annex I Parties, the historical base year is 1990. However, parties included in Annex I with an economy in transition during the early
1990s (EIT Parties) were allowed to choose one year up to a few years before 1990 as reference because of a non-representative collapse during the
breakup of the Soviet Union (e.g., Bulgaria, 1988, Hungary, 1985–1987, Poland, 1988, Romania, 1989, and Slovenia, 1986).



Australia) have used atmospheric observations (TD) to complement their national inventory data (Bergamaschi et al., 2018).

A key priority in the current policy process is to facilitate the global stock-take exercise of the Paris agreement, the first one coming in 2023 and to assess collective progress towards achieving the near- and long-term objectives, considering mitigation, adaptation and means of implementation. The global stock-take is expected to create political momentum for enhancing commitments in Nationally Determined Contributions (NDCs) under the Paris Agreement. Key components of the global stock-take are the NGHGI submitted by countries under the Enhanced transparency framework of the Paris Agreement. Under the framework, for the first time, developing countries will be required to submit their inventories and also commit to provide regular reports to UNFCCC, alongside developed countries, that will continue to submit also on an annual basis. Some developing countries will face challenges to provide and then update inventories.

The work presented here represents dozens of distinct datasets and models, in addition to the individual country submissions to the UNFCCC for all European countries (NGHGIs), which while following the general guidance laid out in IPCC (2006) still differ in specific approaches, models, and parameters, in addition to differences underlying activity datasets. A comprehensive investigation of detailed differences between all datasets is beyond the scope of this paper, though attempts have been previously made for specific subsectors (e.g. agriculture Petrescu et al., 2020) and in dedicated gas-specific follow-ups to this manuscript. As this is the most comprehensive comparison of NGHGIs and research datasets (including both TD and BU approaches) for the European continent to date, we focus here on the rich set of questions that such a comparison raises without necessarily yet offering detailed solutions: How to compare the detailed sectoral NGHGI to the observation-based estimates? Which new information the observation-based estimates are likely to bring (mean fluxes, trend, ensemble variability)? What to expect from such a complex study and how to proceed forward?

We compare official anthropogenic NGHGI emissions with research datasets, and wherever needed harmonizing research data on total emissions to ensure consistent comparisons of anthropogenic emissions. We analyze differences and inconsistencies between emissions, and make recommendations towards future actions to evaluate NGHGI data. While NGHGI include uncertainty estimates, individual spatially disaggregated research datasets of emissions often lack quantification of uncertainty. Here, we use the median[4] and minimum/maximum (min/max) range of different research products of the same type to get a first estimate of uncertainty.

## 2. $CH_4$ and $N_2O$ data sources and estimation approaches

We analyze $CH_4$ and $N_2O$ emissions in the EU27+UK from inversions (TD) and anthropogenic emissions from various BU approaches that cover specific sectors. These data (Table 2) span the period from 1990 and 2018, with the same data available for shorter time periods. The data are from peer-reviewed literature and from unpublished

---

[4] The reason for using median instead of mean for the ensembles is because there is a large spread between global inversions and we don't want to be biased by outliers/extremes.



research results from the VERIFY project (Table 1 and Appendix A). They are compared with NGHGI official submissions up to 2017 and with the UNFCCC-NRT inventory to capture 2018 estimates (Near Real Time, EEA 2019). References are given in Tables 2, 3 and the detailed description of all products in Appendix A1-A3. From BU

approaches, we used inventories of anthropogenic emissions covering all sectors (EDGAR v5.0 and GAINS), inventories limited to agriculture (CAPRI and FAOSTAT), one biogeochemical model of natural peatland emissions (JSBACH-HIMMELI), literature data for geological emissions on land (excluding marine seepage) (Etiope et al., 2019; Hmiel et al., 2020) and for lakes and reservoirs (Del Sontro et al., 2018). Emissions from gas hydrates and termites are not included as they are close to zero in the EU27+UK (Saunois et al., 2020). Emissions from LULUCF biomass burning emissions of $CH_4$ account for 3 % of the total emissions in EU27+UK. These estimates are described

in 2.2. From TD approaches, we used both regional and global inversions, the latter having a coarser spatial resolution. These estimates are described in 2.3.

For $N_2O$ emissions, we used the same global BU inventories as for $CH_4$, and natural emissions from inland waters (rivers, lakes and reservoirs) from (Maavara et al., 2019; Lauerwald et al., 2019). According to Yuanzhi Yao (pers. comm.), about 66 % of the $N_2O$ emitted by Europe's natural rivers are considered anthropogenic indirect

emissions, caused by leaching and run-off of N-fertilizers from the agriculture sector. We did not account for natural $N_2O$ emissions from unmanaged soils (Tian et al., 2019, estimated pre-industrial soil emissions in Europe at a third of the level of the most recent decade - emissions that in pre-industrial times may have been influenced by human management activities, or based on natural processes that have been abolished since). For $N_2O$ inversions, we used one regional inversion FLEXINVERT_NILU and three global inversions (Friedlingstein et al., 2019; Tian et al.,

2020). Agricultural sector emissions of $N_2O$ were presented in detail by Petrescu et al., 2020. In this current study these emissions belong to CAPRI model and FAOSTAT, with the latter additionally covering non-$CO_2$ emissions from biomass fires in LULUCF. Fossil fuel related and industrial emissions were obtained from GAINS (see Appendix A1). Table AA in Appendix A presents the methodological differences of current study with respect to Petrescu et al., 2020.


Table 1: Sectors used in this study and data sources providing estimates for these sectors.

| Anthropogenic (BU)[5] $CH_4$ and $N_2O$ | Natural (BU)[6] $CH_4$ | Natural** (BU) $N_2O$ | TD ($CH_4$ and $N_2O$) |
|---|---|---|---|
| 1. Energy (NGHGI, GAINS, EDGAR v5.0) | | | No sectoral split – total emissions (**FLEXPART - FLExKF-TM5-4DVAR; TM5-4DVAR; FLEXINVERT_NILU; CTE-CH4 InTEM-NAME InGOS inversions** |
| 2. Industrial Products and Products in Use (IPPU) (NGHGI, GAINS, EDGAR v5.0) | | | |

---

[5] For consistency with the NGHGI, here we refer to the five reporting sectors as defined by the UNFCCC and the Paris Agreement decision (18/CMP.1), and the IPCC Guidelines (IPCC,2006), and their refinement (IPCC, 2019a), with the only exception that the IPCC groups together Agriculture and LULUCF sector in one sector (Agriculture, Forestry and Other land Use - AFOLU)

[6] With natural we refer to unmanaged natural emissions (wetlands, geological, inland waters) not reported under the UNFCCC LULUCF sector.



| | | | |
|---|---|---|---|
| 3. Agriculture* **(NGHGI, CAPRI, GAINS, EDGAR v5.0, FAOSTAT, ECOSSE, DayCent)** | | | **GCP-CH₄ 2019 anthropogenic partition from inversions GCP-CH₄ 2019 Natural partition from inversions GN₂OB 2019)** |
| **4.** LULUCF total emissions **(Fig. 1,2,4,5, B1a for CH₄ and Fig. 6,7,9,B1b for N₂O)** | | | |
| 5. Waste **(NGHGI, GAINS, EDGAR v5.0)** | | | |
| | Peatlands, inland waters (lakes and reservoirs) and geological fluxes **(JSBACH-HIMMELI, non-wetland waters_ULB, Hmiel et al., Etiope et al.)** | Inland water (lakes, rivers and reservoirs) fluxes **(non-wetland inland waters_ULB)** | |

\* Anthropogenic (managed) agricultural soils can also have a level of natural emissions.

\*\*Natural soils (unmanaged) can have both natural and anthropogenic emissions.


The units used in this paper are metric tonne (t) [1kt = $10^9$ g; 1Mt = $10^{12}$g] of $CH_4$ and $N_2O$. The referenced data used for the figures' replicability purposes are available for download at https://doi.org/10.5281/ zenodo.4288969 (Petrescu et al., 2020). We focus herein on EU27+UK. In the VERIFY project, we have constructed in addition a web tool which allows for the selection and display of all plots show in this paper (as well as the companion paper on $CO_2$), not only for the regions shown here but for a total of 79 countries and groups of countries in Europe. The website, located on the VERIFY project website: http://webportals.ipsl.jussieu.fr/VERIFY/FactSheets/, is accessible with a username and password distributed by the project. Figure 1 includes also data from countries outside the EU but located within geographical Europe (Switzerland, Norway, Belarus, Ukraine and Rep. of Moldova).

### 2.1. CH₄ and N₂O anthropogenic emissions from NGHGI

UNFCCC NGHGI (2019) emissions are country estimates covering the period 1990-2017. They were kept separate to be compared with other BU and TD data. We supplemented the NGHGI estimates with the NRT – Near Real Time (EEA, 2019) to capture one additional year with preliminary estimates[7]. NRT represents the approximated GHG inventory (also referred to as "proxy estimates") with an early estimate of the GHG emissions for the preceding year, as required by Regulation (EU) 525/2013 of the European Parliament and of the Council.

Anthropogenic $CH_4$ emissions from the four UNFCCC sectors (Table 1, excl. LULUCF) were grouped together. As anthropogenic NGHGI $CH_4$ emissions from the LULUCF sector are very small for EU27+UK (2.6 % in 2017 incl. biomass burning) we exclude them in Figures 4 but include them in the total UNFCCC estimates from Figure 1,2,3,5 and 6. Only a few countries[8] under the NGHGI volunteered to report "wetland" emissions, following

---

[7] t-1 refers to an early estimate of the GHG emissions for the preceding year, as required by Regulation (EU) 525/2013 of the European Parliament and of the Council.

[8] Denmark, Finland, Germany, Ireland, Latvia, Sweden, France, Estonia and Spain. In total these nine countries report in 2017 11.2 kton $CH_4$ from managed wetlands (UNFCCC 2019, CRF Table4(II)D: https://unfccc.int/documents/194946, last access September 2020).

the recommendations of the 2014 IPCC Wetlands supplement (IPCC, 2014) and these emissions were not included in the NGHGI total,  following the IPCC (2006) guidelines as the reference for NGHGI and in absence of a detailed description of what they cover. According to NGHGI data between 2008 and 2017, the wetland emissions in the EU27+UK reported under LULUCF (CRF table 4(II) accessible for each EU27+UK country[9]) include only managed wetlands which represent one fourth of the total wetland area in EU27+UK (G. Grassi pers. comm.) and sum up to

0.1 Tg $CH_4$ (Petrescu et al., 2020).

Anthropogenic $N_2O$ emissions are predominantly related to agriculture (for EU27+UK, 69 % in 2017) but are also found in the other sectors (Tian et al. 2020). In addition, $N_2O$ has natural emissions, which are defined as the pre-industrial background, that is before the use of synthetic N-fertilizers and intensive agriculture, and derive from natural processes in soils but also in lakes, rivers and reservoirs (Maavara et al., 2019; Lauerwald et al., 2019; Tian et

al., 2020).

### 2.2. $CH_4$ and $N_2O$ anthropogenic and natural emissions from other bottom-up sources

We used four global $CH_4$ and $N_2O$ BU anthropogenic emissions inventories CAPRI, FAOSTAT, GAINS and EDGAR v5.0 (Table 2, 3) These estimates are not completely independent from NGHGIs (see Figure 4 in Petrescu et

al., 2020) as they integrate their own sectorial modelling with the UNFCCC data (e.g. common activity data and IPCC emission factors) when no other source of information is available.

Anthropogenic emissions from these datasets follow, or can be matched, to Table 1 sectors. The $CH_4$ biomass and biofuel burning emissions are included in NGHGI under the UNFCCC LULUCF sector, although they are identified as a separate category by the Global Carbon Project $CH_4$ budget synthesis (Saunois et al., 2020). For both

$CH_4$ and $N_2O$, CAPRI (Britz and Witzke, 2014; Weiss and Leip, 2012) and FAOSTAT (FAO, 2020) report only agricultural emissions. None of the BU inventories reported uncertainties, except for the 2015 values of EDGAR v5.0 (Solazzo et al., 2020 submitted to ACP) and for an earlier FAOSTAT dataset only up to 2010 (Tubiello et al., 2013 and Appendix B).

The $CH_4$ natural emissions belong to "peatlands", and "other natural emissions", the latter including

geological sources and inland waters (lakes and reservoirs), following Saunois et al, 2020. For peatlands, we used the JSBACH-HIMMELI framework and the ensemble of thirteen monthly gridded estimates of peatland emissions based on different land surface models as calculated for Saunois et al. (2020), all described in Appendix A2. In EU27+UK, geological emissions were calculated by scaling the regional emissions from Etiope et al. 2019 (37.4 Tg $CH_4$ $yr^{-1}$) to the global ratio of emissions from Hmiel et al. (2020), obtaining an estimate of 1.3 Tg $CH_4$ $yr^{-1}$ (marine and land

geological). Marine seepage emissions were excluded. This rescaled geological source represents 24 % of the total EU27+UK natural $CH_4$ emissions. Inland waters (lakes and reservoirs, based on Lauerwald et al., 2019 and Del Sontro et al., 2018) (Appendix A2) are the largest natural component (48 %), the rest (28 %) being attributed to peatlands. Overall, in EU27+UK the natural emissions thus accounted for 5.2 Tg $CH_4$ $yr^{-1}$.

---

[9]    https://unfccc.int/process-and-meetings/transparency-and-reporting/reporting-and-review-under-the-convention/greenhouse-gas-inventories-annex-i-parties/national-inventory-submissions-2019



The N$_2$O anthropogenic emissions from BU datasets belong predominantly to two main categories, as presented in Table 5: 1) direct emissions from the agricultural sector where synthetic fertilizers and manure were applied, and from manure management and 2) indirect emissions on non-agricultural land and water receiving anthropogenic N through atmospheric N deposition, leaching and run-off (also from agricultural land). Furthermore, emissions from industrial processes are declining over time but originate from fossil fuel combustion, air pollution abatement devices, specific chemical reactions, wastewater treatment and land use change. In this study, we do not consider the natural emissions from soils, since these emissions are relatively small for temperate regions, including Europe and cannot be singled out in landscapes largely dominated by human activities. Therefore, the only "natural" fluxes considered in this study are emissions from inland waters (lakes, rivers and reservoirs, Maavara et al., 2019; Lauerwald et al., 2019, Appendix A3) even if, more than half of the emissions (56 % globally, Tian et al., 2020, and 66 % for Europe, Yao pers. comm.) are due to eutrophication following N-fertilizer leaching to inland waters. Emissions from natural soils, in this study are considered as "anthropogenic" because, according country specific NIRs, all land in EU27+UK is considered to be managed, except 5% of France EU territory.

### 2.3. CH$_4$ and N$_2$O emission data from top-down inversions

Inversions combine atmospheric observations, transport and chemistry models and prior estimates of GHG sources all with their uncertainties, to estimate emissions. Emission estimates from inversions depend on the data set of atmospheric measurements and the choice of the atmospheric model, as well as on other settings (e.g. prior emissions and their uncertainties). Inversions outputs were taken from original publications without evaluation of their performance through specific metrics (e.g. fit to independent cross validation atmospheric measurements (Bergamaschi et al., 2013, 2018; Patra et al., 2016). Some of the inversions solve explicitly for sectors, others solve for all fluxes in each grid cell and separate sectors using prior grid-cell fractions (see details in Saunois et al. 2020 for global inversions).

For CH$_4$, we use nine regional TD inversions and 22 global TD inversions listed in Table 2. These inversions are not independent from each other: some are variants from the same modeling group, many use the same transport model, and most of them use the same atmospheric data. Different prior data is generally used in models, which produces a greater range of posterior emission estimates (Appendix B, Table B4). The subset of InGOS inversions (Bergamaschi et al., 2018a) belongs to a project where all models used the same atmospheric data over Europe covering the period 2006-2012. The global inversions from Saunois et al. 2020 were all updated to 2017.

The regional inversions generally use both higher resolution a priori data and higher resolution transport models, and e.g. TM5-JRC runs simultaneously over the global domain at coarse resolution and over the European domain at higher resolution, with atmospheric CH$_4$ concentration boundary conditions taken from global fields. For CH$_4$, 11 global inversions use GOSAT for the period 2010-2017, eight global inversions use SURF since 2000, two global use SURF since 2010 and one SURF since 2003 (see "Appendix 4 Table " in Saunois et al. 2020 and Table 2 below). None of the regional inversions use GOSAT prior data as all base their prior data on SURF stations.

*Table 2: Data sources for CH$_4$ and N$_2$O emissions used in this study*



| Method | Name | CH₄ | N₂O | Contact / lab | References |
|---|---|---|---|---|---|
| **CH₄ and N₂O Bottom-up anthropogenic** | | | | | |
| UNFCCC NGHGI (2019) | UNFCCC CRFs | CH₄ emissions 1990-2017 | N₂O emissions 1990-2017 | MS inventory agencies | UNFCCC CRFs https://unfccc.int/process-and-meetings/transparency-and-reporting/reporting-and-review-under-the-convention/greenhouse-gas-inventories-annex-i-parties/national-inventory-submissions-2019 |
| UNFCCC | UNFCCC MS-NRT | t-1 proxy estimate for 2018 | t-1 proxy estimate for 2018 | EEA | EEA Report, Approximated EU GHG inventory: proxy GHG estimates for 2018, https://www.eea.europa.eu/publications/approximated-eu-ghg-inventory-proxy |
| BU | EDGAR v5.0 | CH₄ sectoral emissions 1990-2015 | N₂O sectoral emissions 1990-2015 | EC-JRC | Crippa et al., 2019a Crippa et al., 2019 EU REPORT Janssens-Maenhout et al., 2019 Solazzo et al., 2020 (in review ACP) |
| BU | CAPRI | CH₄ agricultural emissions 1990-2013 | N₂O agricultural emissions 1990-2013 | EC-JRC | Britz and Witzke, 2014 Weiss and Leip, 2012 |
| BU | GAINS | CH₄ sectoral emissions 1990-2015 | N₂O sectoral emissions 1990-2015 (every five years) | IIASA | Höglund-Isaksson, L. 2012 Höglund-Isaksson, L. 2017 Höglund-Isaksson, L. et al., 2020 Gomez-Sanabria, A. et al., 2018 Winiwarter et al., 2018 |
| BU | FAOSTAT | CH₄ agriculture and land use emissions 1990-2017 | N₂O agricultural emissions 1990-2017 | FAO | Tubiello et al. 2013 FAO, 2015, 2020 Tubiello, 2019 |
| BU | ECOSSE | | Direct N₂O emissions from agricultural soils 2000-2015 | UNIABDN | Bradbury et al., 1993 Coleman., 1996 Jenkinson., 1977, 1987 Smith et al., 1996, 2010a,b |
| BU | DayCent | | N₂O emissions from direct agricultural soils avg. 2011-2015 | EC-JRC | Orgiazzi et al., 2018 Lugato et al., 2018, 2017 Quemada et al., 2020 |
| **CH₄ and N₂O bottom-up natural** | | | | | |
| BU | JSBACH-HIMMELI | CH₄ emissions from peatlands 2005-2017 | | FMI | Raivonen et al., 2017 Susiluoto et al., 2018 |



| BU | Non-wetland inland waters | One average value for CH$_4$ fluxes from lakes and reservoirs with uncertainty 2005-2011 | N$_2$O average value for emissions from lakes, rivers, reservoirs Average of 2010-2014 | ULB | Maavara et al., 2017, 2019 Lauerwald et al., 2019 Deemer et al., 2016 Del Sontro et al., 2018 Mccauley et al., 1989 |
|---|---|---|---|---|---|
| BU | Geological emissions, including marine and land geological) | Total preindustrial-era geological CH$_4$ emissions | | Hmiel et al., 2020 Etiope et al., 2019 | Hmiel et al., 2020 https://www.nature.com/articles/s41586-020-1991-8) Etiope et al., 2019 |
| | **CH$_4$ and N$_2$O Top-down inversions** | | | | |
| | **Regional inversions over Europe ( high transport model resolution )** | | | | |
| TD | FLEXPART - FLExKF-TM5-4DVAR | Total CH$_4$ emissions from inversions with uncertainty 2005-2017 | | EMPA | Brunner et al., 2012 Brunner et al., 2017 Background concentrations from TM5-4DVAR, Bergamaschi et al., 2018a |
| TD | TM5-4DVAR | CH$_4$ emissions from inversions, split into total, anthropogenic and natural 2005-2017 | | EC-JRC | Bergamaschi et al., 2018a |
| TD | FLEXINVERT_NILU | CH$_4$ total emissions from inversions 2005-2017 | N$_2$O total emissions,2005-2017 | NILU | Thompson and Stohl, 2014 |
| TD | CTE-CH$_4$ | Total CH$_4$ emissions from inversions for Europe with uncertainty 2005-2017 | | FMI | Brühl et al., 2014 Howeling et al., 2014 Giglio et al., 2013 Ito et al., 2012 Janssens-Maenhout et al., 2013 Krol et al., 2005 Peters et al., 2005 Saunois et al., 2020 Stocker et al., 2014 Tsuruta et al., 2017 |
| TD | InTEM-NAME | CH$_4$ emissions only plotted for the UK | | MetOffice UK | Jones et al., 2007 Cullen et al., 1993 Arnold et al., 2018 |
| TD | InGOS inversions | Total CH$_4$ emissions from inversions 2006-2012 | | EC-JRC and InGOS project partners | Bergamaschi et al., 2018a TM5-4DVAR: Meirink et al., 2008; Bergamaschi et al. 2010; 2015 TM5-CTE: Tsuruta et al., 2017 LMDZ-4DVAR: Hourdin and Armengaud, 1999; Hourdin et al., 2006 TM3-STILT: Trusilova et al., 2010, Gerbig et al., 2003; Lin et al., 2003; Heimann and Koerner, 2003 |

|  |  |  |  |  | NAME: Manning et al. 2011; Bergamaschi et al., 2015 CHIMERE: Berchet et al. 2015a; 2015b; Menut et al., 2013; Bousquet et al., 2011\ COMET: Eisma et al., 1995; Vermeulen et al., 1999; Vermeulen et al., 2006 |
|---|---|---|---|---|---|
| **Global inversions from the Global Carbon Project CH₄ and N2O budgets (Saunois et al. 2020, Tian et al., 2020)** | | | | | |
| TD | GCP-CH₄ 2019 anthropogenic partition from inversions | 22 models for CH₄ inversions, both *SURF and GOSAT* 2000-2017 |  | LSCE and GCP-CH₄ contributors | Saunois et al., 2020 and model specific references in Appendix B, Table B4 |
| TD | GCP-CH₄ 2019 Natural partition from inversions | 22 models with optimized wetland CH₄ emissions 2000-2017 |  | LSCE | Saunois et al., 2020 and model specific references in Appendix B, Table B3 |
| TD | GN₂OB 2019 |  | Inverse N₂O emissions - 3 Inversions PYVAR TOMCAT MIROC4-ACTM 1998-2016 | GN₂OB 2019 and contributors | Thompson et al., 2019 Tian et al., 2020 |

For N$_2$O, we use one regional inversion (FLEXINVERT_NILU for 2005-2017 period) and three global inversions for the period 1998-2016 from Thompson et al. (2019), listed in Table 2. These inversions are not completely independent from each other since most of them use the same prior information (Appendix B, Table B4).

The regional inversion uses a higher resolution transport model for Europe, with atmospheric N$_2$O concentration boundary conditions taken from global fields. As all inversions derived total rather than anthropogenic emissions, emissions from inland waters (lakes, rivers and reservoirs) estimated by Maavara et al. (2019) and Lauerwald et al. (2019) were subtracted from the total emissions. Note that the estimates of Maavara et al. (2019) and Lauerwald et al. (2019) include anthropogenic emissions from N-fertilizer leaching accounting for 66% of the inland water emissions

in EU27+UK. The natural N$_2$O emissions are small, but should be better quantified in the future to allow for a more accurate comparison between BU (anthropogenic sources only) and TD estimates.

The largest share of N$_2$O emissions comes from the agricultural soils (direct and indirect emissions from the applications of fertilizers, whether synthetic or manure) contributing in 2017 69 % of the total N$_2$O emissions (excl. LULUCF) in EU27+UK. In Table 3 we present the allocation of emissions by activity type covering all agricultural

activities and natural emissions, following the IPCC classification. We notice that each data product has its own particular way of grouping emissions, and does not necessarily cover all emissions activities. Main inconsistencies between models and inventories are observed with activity allocation in the two models (ECOSSE and DayCent). ECOSSE only estimates direct N$_2$O emissions, and does not estimate downstream emissions of N$_2$0, for example indirect emissions from nitrate leached into water courses, which also contributes to an underestimation of total N$_2$O

emissions. Field burning emissions are as well not included by most of the data sources.





*Table 3: Adapted from Petrescu et al., 2020: Agriculture and natural $N_2O$ emissions - Allocation of emissions to different sectors by different data sources*

| Emission sources/Data providers | UNFCCC NGHGI (2019) | UNFCCC MS-NRT | EDGAR v5.0 | CAPRI | GAINS | FAOSTAT | ECOSSE | DayCent | Inland waters |
|---|---|---|---|---|---|---|---|---|---|
| **Direct $N_2O$ emissions from manure management** | 3.B.2 minus 3.B.2.5 Manure management | 3.B Manure management | 4.B Manure management | N2OMAN Manure management | 3B Manure management | Manure management | n/a | n/a | n/a |
| **Direct $N_2O$ emissions from managed soils** | 3.D.1.1 and 3.D.1.2 – direct $N_2O$ emissions from managed soils (inorganic N and organic N fertilizers) 3.D.1.4 Crop residues 3.D.1.6 Cultivation of organic soils | 3.D. Agricultural soils | 4.D.1 – direct soil emissions | N2OAPP – manure application on soils N2OSYN – synthetic fertilizer application N2OHIS - histosols N2OCRO – crop residues | 3.D.a.1 - Soil: Inorganic fertilizer and crop residues 3.D.a.2 - organic fertilizer 3.D.a.6 - histosols | Synthetic fertilizers Crop residues Cultivation of organic soils Prescribed burning of savannas | Direct $N_2O$ emissions | Direct emissions from manure application + Direct $N_2O$ emissions (fertilizers ?) | n/a |
| **Direct $N_2O$ emissions from grazing animals** | 3.D.1.3 – Urine and Dung Deposited by Grazing Animals | n/a | 4.D.2 - Manure in pasture/range/paddock | N2OGRA - grazing | 3.D.a.3 - grazing | Manure left on pasture | n/a | Direct and indirect $N_2O$ emissions from grazing animals | n/a |
| **Indirect $N_2O$ emissions** | 3.B.2.5. – Indirect $N_2O$ Emissions from leaching from manure management 3.D.2 Indirect emissions from soils (3.D.2.1 atmospheric deposition - volatilized N + 3.D.2.2 leaching and run-off) | n/a | 4.D.3 – Indirect $N_2O$ from agriculture | N2OLEA - leaching N2OAMM – ammonia volatilization | 3.D.b.1 - atmospheric deposition 3.D.b.2 - leaching | Manure applied to soils | Atmospheric N deposition | Atmospheric N deposition | Runoff and leaching N-fertilizers |
| **Field burning of agricultural residues** | 3.F. Field Burning of Agricultural Residues | 3.F. Field burning of agricultural residues | 4.F. – agricultural waste burning | n/a | n/a | Field burning of crop residues | n/a | n/a | n/a |
| **Natural (unmanaged) $N_2O$ emissions** | n/a | n/a | n/a | n/a | n/a | n/a | n/a | n/a | Emissions from lakes, rivers and estuaries |



## 3. Results and discussion

### 3.1. Comparing CH₄ anthropogenic emissions estimates from different approaches\

#### 3.1.1. *Estimates of European and regional total CH₄ fluxes*

We present results of total $CH_4$ fluxes from EU27+UK and five main regions in Europe: North, West, Central, East (non-EU) and South. The countries included in these regions are listed in Appendix A, table A. Figure 1 shows the total $CH_4$ fluxes from NGHGI for both base year 1990 and mean of 2011-2015 period. This period was the common denominator for which data was available, including 2 years of the Kyoto Protocol first reporting period (2011-2012) and reaching the year of the Paris Agreement was adopted. We aim with the selection of this period to bring together all information over a 5-year period for which values are known in 2018. In fact, this can be seen as a reference for what we can achieve in 2023, the year of the first global stocktake, where for most UN Parties the reported inventories will be known until 2021. Given that the global stocktake is only repeated every 5 years, a five-year average is clearly of interest.

The total NGHGI estimates include emissions from all sectors and we plot and compare them with fluxes from global datasets, BU models and inversions. We note that for all five regions, the NGHGI reported $CH_4$ emissions decreased, by 21 % in South Europe, by up to 54 % in East Europe, and by 35% for the European Union with respect to the 1990 value. This is encouraging in the context of meeting EUs commitments under the PA (at least 50% and towards 55% compared with 1990 levels stated by the amended proposal for a regulation of the European parliament and of the council on establishing the framework for achieving climate neutrality and amending Regulation (EU) 2018/1999 (European Climate Law) ([https://ec.europa.eu/clima/sites/clima/files/eu-climate-action/docs/prop_reg_ecl_en.pdf](https://ec.europa.eu/clima/sites/clima/files/eu-climate-action/docs/prop_reg_ecl_en.pdf)) and reaching carbon neutrality by 2050). It also shows that not only at EU27+UK level, but also at regional European level, the emissions from BU (anthropogenic and natural) and TD estimates agree well with reported NGHGI data despite the high uncertainty observed in the TD models. This uncertainty is represented here by the variability in the model ensembles and denotes the range of the extremes (min and max) of estimates within each model ensemble. From Figure 1 we clearly note that Northern Europe is dominated by natural (wetlands) emissions while Western, Central and Southern Europe emissions are dominated by anthropogenic sectors (e.g. agriculture).

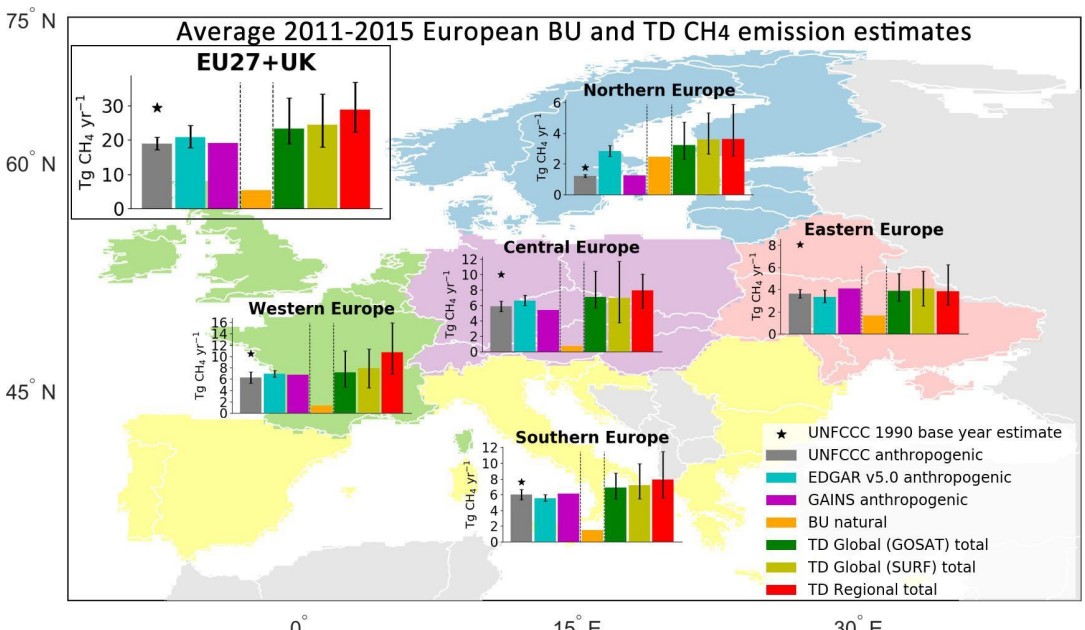

*Figure 1: Five years average 2011-2015 total CH₄ emission estimates (incl. LULUCF) for EU27+UK and five European regions (North, West, Central, South and East non-EU). Eastern European region does not include*

*European Russia and the UNFCCC uncertainty for the Republic of Moldova was not available. Northern Europe includes Norway. Central Europe includes Switzerland. The data belongs to UNFCCC NGHGI (2019) submissions (grey) and base year 1990 (black star), two BU inventories (GAINS and EDGAR v5.0), natural unmanaged emissions (sum of peatland, geological and inland waters emissions) and three TD total estimates (regional European inversions (excluding InGOS unavailable for 2013-2015) and GOSAT and SURF estimates from global inverse models). The*

*relative error on the UNFCCC value represents the NGHGI (2018) reported uncertainty computed with the error propagation method (95% confidence interval); is 9.3 % for the EU27+UK, 10 % for Eastern Europe non-EU, 7.8 % for Northern Europe, 10.9 % for Southern Europe, 16.1 % for Western Europe and 11 % for Central Europe.*

The EDGAR v5.0 estimate for Northern Europe is twice as high when compared to NGHGI and GAINS, and
this is because of CH₄ emissions from the fuel production and distribution (IPCC sector 1B) and waste sectors. Most Scandinavian countries rely for their power and heat supply on biogenic fuels which introduces more uncertainty in the use of activity data and emission factors. The allocation of auto-producers as explained in section 3.2 could be another reason for differences. The waste sector emissions for Norway, Sweden, Finland and Estonia are different but still consistent with the landfills emissions from EDGAR v4.3.2, which are known to be up to twice as high as the
nationally reported value (Janssens-Maenhout et al., 2019). For Eastern Europe we note that BU anthropogenic estimates have the same magnitude as the TD. We hypothesize that this could be due to a less dense network of surface stations.



In line with Bergamaschi et al., 2018a we highlight the potential significant contribution from natural unmanaged sources (peatlands, geological and inland water), which for EU27+UK accounted for 5.24 Tg $CH_4$ yr$^{-1}$

(Figure 1). Taking into account these natural unmanaged $CH_4$ emissions, and adding it to the range of the anthropogenic estimates (19 – 21 Tg $CH_4$ yr$^{-1}$) the total BU estimates become broadly consistent for all European regions with the range of the TD estimates (23 – 28 Tg $CH_4$ yr$^{-1}$).

### 3.1.2. NGHGI sectoral emissions and decadal changes

According to the UNFCCC (2019) NGHGI estimates, in 2017 the EU27+UK emitted GHGs totaling 3.9 Gt $CO_2e$ (incl. LULUCF), of this total, $CH_4$ emissions accounted for ~11% (0.4 Gt $CO_2e$ or 18.1 Mt $CH_4$ yr$^{-1}$) (Appendix, B, Figure B1a) with France, UK and Germany contributed together 36% of total $CH_4$ emissions.

The data in Figure 2 shows anthropogenic $CH_4$ emissions and their change from one decade to the next, from UNFCCC NGHGI (2019), with the contribution from different UNFCCC sectors. In 2017, NGHGI report $CH_4$ from

agricultural activities to be 52 % (± 10 %) of the total EU27+UK $CH_4$ emissions, followed by emissions from waste, 27 % (± 23 %). The large share of agriculture in total anthropogenic $CH_4$ emissions also holds at global level (IPCC SRCCL 2019). Between the 1990s and the 2000s, the net -17.7 % reduction originates largely from energy and waste with IPPU (metal and chemical industry) and LULUCF having negligible change. Between the 2000s and 2010-2017, the -15.5 % reduction is distributed more evenly across sectors, with waste having the largest contribution, and

industry showing no change. The two largest sectors composing total EU27+UK emission are Agriculture and Waste, but Energy and Waste are showing the higher reductions over the last decade.

The reduction observed in the waste sector is partly due to the adoption of the first EU methane strategy published in 1996 (COM_1996_557_EN_ACTE_f.pdf.en.pdf). EU legislation addressing emissions in waste sector proved to be successful and brought about the largest reductions. Directive 1999/31/ EC on the landfill of waste (also

referred to as the Landfill Directive) required the MS to separate waste, minimizing the amount of biodegradable waste disposed untreated in landfills and to install landfill gas recovery at all new sites. Based on the 1999 Directive, the new 2018/1999 EU Regulation on the Governance of the Energy Union requires the European Commission to propose a strategic plan for methane, which will become an integral part of the EU's long-term strategy. In the waste sector, the key proposal included the adoption of EU legislation requiring the installation of methane recovery and use

systems at new and existing landfills. Other suggested actions included measures aimed at the minimization, separate collection and material recovery of organic waste (Olczak and Piebalgs, 2019).

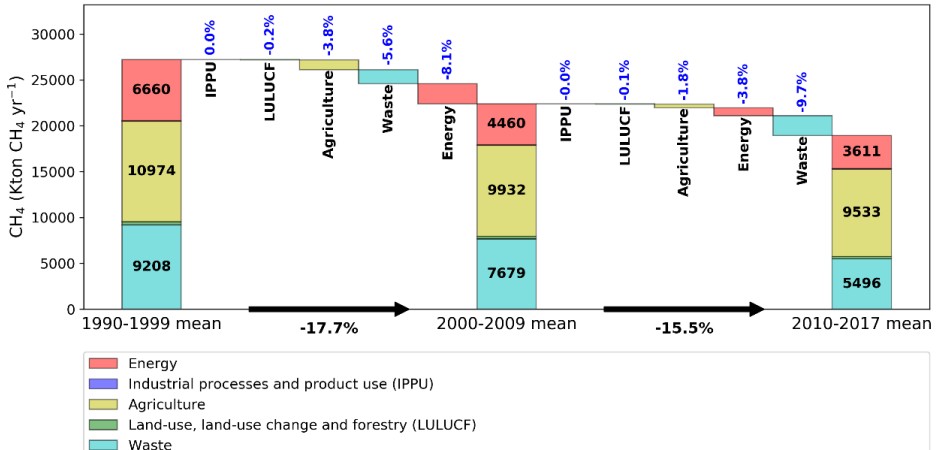

*Figure 2: The contribution of changes (%) in CH4 anthropogenic emissions in the five UNFCCC sectors to the overall change in decadal mean, as reported to UNFCCC NGHGI (2019). The three stacked columns represent the average*

*CH4 emissions from each sector during three periods (1990-1999, 2000-2009 and 2010-2017) and percentages represent the contribution of each sector to the total reduction percentages (black arrows) between periods.*

### 3.1.3. NGHGI estimates compared with bottom-up inventories

The data in Figure 3 present the total anthropogenic CH4 emissions from four BU inventories and UNFCCC

NGHGI (2019) excl. those from LULUCF. According to NGHGI, anthropogenic emissions from the total EU27+UK of the four UNFCCC sectors (Table 1, excl. LULUCF) amounted to 18.2 Tg CH4 in the year 2017, 10.7 % of the total GHG emissions in CO2-eq. In Figure 3a, we observe that EDGARv5.0 and GAINS show consistent trends with NGHGI (excl. LULUCF), but GAINS reports consistently lower estimates (10 %) and EDGARv5 consistently higher estimates (8 %) compared to NGHGI. In contrast to the previous version, EDGAR v4.3.2, which was found by

Petrescu et al. 2020 to be consistent with NGHGI (2018) data, EDGAR v5.0 reports higher estimates but within the 9.4 % UNFCCC uncertainty range. The trends in emissions agree better between the two BU inventories and NGHGI over 1990-2015:, with linear trends of -1.5 % yr$^{-1}$ in NGHGI compared to -1.5 % yr$^{-1}$ in GAINS and -1.4 % yr$^{-1}$ in EDGAR v5.0.

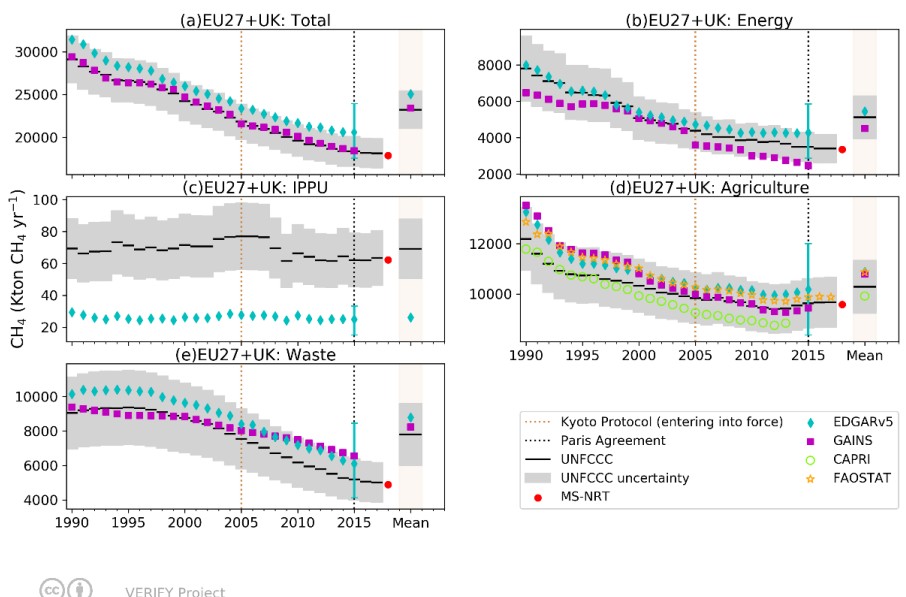

*Figure 3: Total anthropogenic CH₄ emissions (excl. LULUCF):a) of EU27+UK and total sectoral emissions as: b) Energy, c) Industry and Products in Use (IPPU), d) Agriculture and e) Waste from UNFCCC NGHGI (2019) submissions and MS-NRT 2018 compared to global bottom-up inventory models for agriculture (CAPRI, FAOSTAT) and all sectors excl. LULUCF (EDGAR v5.0, GAINS). CAPRI reports one estimate for Belgium and Luxembourg. The relative error on the UNFCCC value represents the UNFCCC NGHGI (2018) MS-reported uncertainty computed with the error propagation method (95% confidence interval); is 9.4 % for the total EU27+UK, 23 % for Energy and Waste, 27 % for IPPU and 10 % for Agriculture. Uncertainty for EDGAR v5.0 was calculated for 2015 and is min 14 % and max 27 %; it represents the 95 % confidence interval of a lognormal distribution. The means represent the common overlapping period 1990-2015. Last reported year in this study refers to 2017 (UNFCCC and FAOSTAT), 2015 (EDGAR v5.0 and GAINS), and 2013 (CAPRI).*

Sectoral time series of anthropogenic CH₄ emissions (excl. LULUCF) and their means are shown in Figures 3b,c,d and e. For the energy sector (Figure 3b), both EDGAR v5.0 and GAINS match well NGHGI trend thanks to updated methodology that derives emission factors bottom-up and accounts for country-specific information about associated petroleum gas generation and recovery, venting and flaring (Höglund-Isaksson, 2017). After 2005, GAINS reports consistently lower emissions than UNFCCC due to a phase-down of hard coal production in Czech Republic, Germany, Poland and the UK, a decline in oil production in particular in the UK, and declining emission factors reflecting reduced leakage from gas distribution networks as old town gas networks are replaced. The consistently higher estimates (+6 % compared to the UNFCCC mean) of EDGAR v5.0 might be due to the use of default emission factors for oil and gas production based on data from the US (Janssens-Maenhout et al. (2019). Next to that, several other reasons could be the cause for the differences (e.g. use of Tier 1 emission factors for coal mines, assumptions





for material in the pipelines (in the case of gas transport) and the activity data). EDGAR v5.0, for example, uses the gas pipeline length as a proxy for the activity data however this may not be appropriate for the case of the official data, which could consider the total amount of gas being transported or both methods according to the countries. Using pipeline length may overestimate the emissions because the pipeline is not always at 100% capacity thus a larger amount of methane is assumed to be leaked. For coal mining, emissions are a function of the different types of

processes being modelled.

The IPPU sector (Figure 3c), which has only a small share of the total emissions, is not reported in GAINS, while EDGAR v5.0 estimates are less than half of the emissions reported by NGHGI 2019 in this sector. The discrepancy for this sector has negligible impact on discrepancy for the total $CH_4$ emission. However, we identified that the low bias of EDGAR v5.0 could be explained by fewer activities included in EDGAR v5.0 (e.g. missing

solvent, electronics and other manufacturing goods) accounting for 5.5 % of the total IPPU emissions in 2015 reported to UNFCCC. The reason for the remaining difference could be explained by the allocation of emissions from auto-producers[10] in EDGAR v5.0 to the Energy sector (following the IPCC 1996 guidelines), while in NGHGI are reported under the IPPU sector (following the 2006 IPCC guidelines).

As CAPRI and FAOSTAT just report emissions from agriculture, we only included them in Fig. 3d. The data

shows that the four data sources (EDGAR v5.0, GAINS, CAPRI and FAOSTAT) show good agreement, with CAPRI at the lower range of emissions (Petrescu et al., 2020) and on average 3% lower than UNFCCC, and EDGAR v5.0 at the upper range. The reason for EDGAR v5.0 having the highest estimate (contrary to Petrescu et al., 2020 where NGHGI were the highest and EDGAR v4.3.2 the second highest) is likely due to the activity data updates in EDGAR v5.0 based on FAOSTAT values, compared to EDGAR v4.3.2). When looking at the time series mean, EDGAR v5.0,

GAINS and FAOSTAT show a similar value, +5 % higher than the NGHGI. This shows good consistency between the three BU estimates and UNFCCC likely due to the use of similar activity data and emission factors (EFs) cfr. Figure 4 in Petrescu et al., 2020.

For the waste sector (Figure 3e) EDGAR v5.0 shows consistent higher estimates compared to the NGHGI data, while GAINS emissions have an increasing trend after 2000 (mean 1990-2015 value 6% higher than NGHGI).

The two inventories, EDGAR v5.0 2020 update for landfills and GAINS used an approach based on the decomposition of waste into different biodegradable streams, with the aim of applying the methodology described in the 2019 Refinement of the 2006 IPCC guidelines and the IPCC waste model (IPCC, 2019) using the First-Order-Decay (FOD) method. The main differences between the two datasets come from i) sources for total waste generated per person, ii) assumption for the fraction composted and iii) the oxidation. The two inventories may have used different strategies

to complete the waste database when inconsistencies were observed in the EUROSTAT database or in the waste trends in UNFCCC.

---

[10] auto-producers of electricity and heat: cogeneration by industries and companies for housing management (central heating and other services) (Olivier et al., 2017 PBL report)



*3.1.4. NGHGI estimates compared to atmospheric inversions*

*Regional inversions*

Figure 5 compares TD regional estimates with NGHGI anthropogenic data for CH$_4$ and with natural BU emissions. We present TD estimates of total emissions (Fig. 5a) as well as estimates of anthropogenic emissions only (Fig. 5b), which are calculated by subtracting the natural emissions from the total inversions.

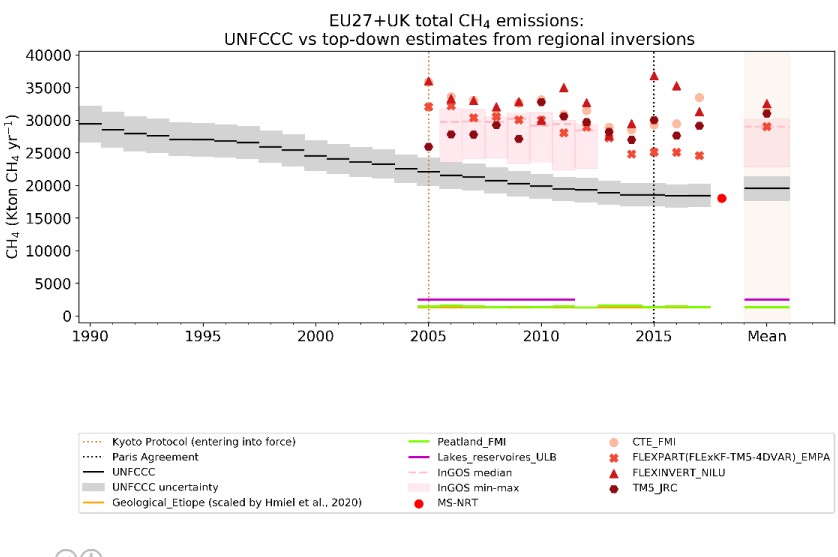


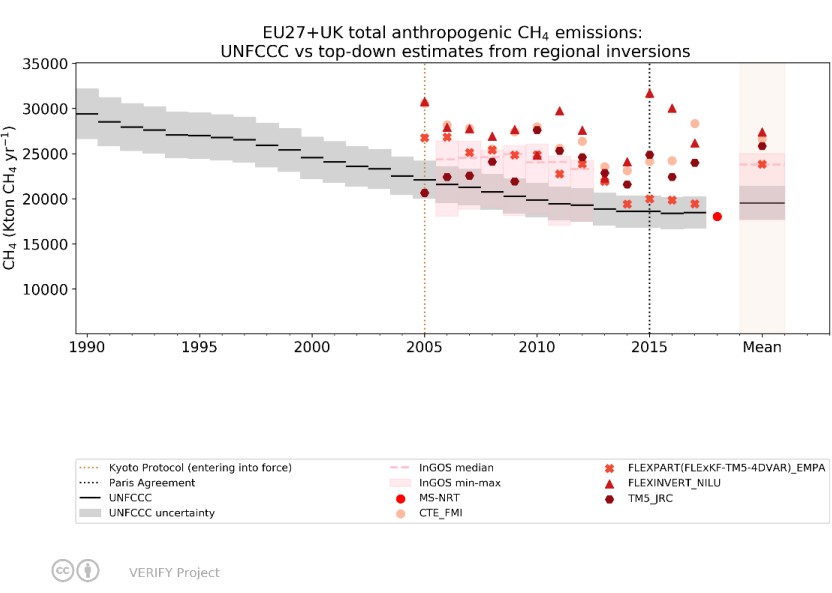





*Figure 4: a) Comparison of **total** CH₄ emissions from top-down regional inversions with UNFCCC NGHGI (2019)*

*data and inland water (lakes_reservoirs_ULB, pink), peatland (from JSBACH-HIMMELI, green), and geological*

*emissions (yellow); b) comparison of **anthropogenic** CH₄ emissions from top-down regional inversions with UNFCCC*

*NGHGI (2019) data. Anthropogenic emissions from these inversions are obtained by removing natural emissions*

*shown in Figure 4a. The MS-NRT LULUCF estimate does not include the following countries: Austria, Belgium,*

*Estonia, Croatia, Hungary, Luxembourg, Latvia, Malta, Slovenia. UNFCCC NGHGI (2018) reported uncertainty*

*computed with the error propagation method (95% confidence interval) is 9.29 % and represents the UNFCCC*

*NGHGI (2018) MS-reported uncertainty for all sectors (incl. LULUCF). The time series mean was computed for the*

*common period 2006-2012.*

The TD estimates of European CH₄ emissions of Figure 4 use four European regional models (2005-2017) and an ensemble of five different inverse models (InGOS, Bergamaschi et al., 2015) for 2006-2012.

For the common period 2006-2012, the four inverse models give a total CH₄ emissions mean of 25.8 (24.0-27.4) Tg CH₄ yr$^{-1}$ compared to anthropogenic total of 20.3 ± 1.9 Tg CH₄ yr$^{-1}$ in NGHGI (Fig. 4a). The large positive difference between TD and NGHGI suggests a potentially significant contribution from natural sources (peatlands, geological sources and inland waters), which for the same period report a total mean of 5.2 Tg CH₄ yr$^{-1}$. However, it needs to be emphasized that wetland emission estimates have large uncertainties and show large variability in the

spatial (seasonal) distribution of CH₄ emissions but for Europe their inter-annual variability is not very strong (mean of 13 years from JSBACH-HIMMELI peatland emissions 1.4 ± 0.1 Tg CH₄ yr$^{-1}$). Overall, they do represent an important source and could dominate the budget assessments in some regions such as Northern Europe (Figure 1). We also note that the TD trends do not necessarily match those of NGHGIs and this might be due to strong seasonality of emissions coming from the natural priors in the inversions (Saunois et al., 2020)

The natural emissions from inland waters (based on Lauerwald et al., 2019, see appendix A2) contribute 2.53 Tg CH₄ yr$^{-1}$, or 48 % of the total natural CH₄ emissions (sum of lakes and reservoirs, geological and peatlands emissions). Peatlands (Raivonen et al. 2017 and Susiluoto et al. 2018) account for 1.38 Tg CH₄ yr$^{-1}$, i.e. 27 % of the total natural CH₄ emissions, and geological sources sum up to 1.27 Tg CH₄ yr$^{-1}$, i.e. 25 % of the total natural CH₄ emissions. It should be noted that geological emissions are an important component of the EU27+UK emissions

budget, although not of concern for climate warming if their source strength has not changed since pre-industrial times (Hmiel et al., 2020). According to the IPCC 2006 guidelines (IPCC, 2006) CH₄ emissions from wetlands are reported by the MS to the NGHGI under the LULUCF sector and considered anthropogenic. They are included in the total LULUCF values (Figure 1,2,4 and 5) and in 2017 only eight EU countries (Germany, Denmark, Spain, Estonia, Finland, Ireland, Latvia and Sweden) reported CH₄ emissions from wetlands accounting only for 11.2 kton CH₄ yr$^{-1}$.

In an attempt to quantify the anthropogenic CH₄ component in the European TD estimates, in Figure 4b we subtract from the total TD emissions the BU peatland emissions from the regional JSBACH-HIMMELI model and those from geological and inland water sources. It remains however uncertain to perform these corrections due to the prior inventory data allocation of emissions to different sectors (e.g. anthropogenic or natural), which can induce uncertainty of up to 100 % if for example an inventory allocates all emissions to natural emissions and the correction





is made by subtracting the natural emissions. The inversion that simulates the closest anthropogenic estimate to the UNFCCC NGHGI (2019) is FLExKF-TM5-4DVAR_EMPA. In 2017, it reports 19.4 Tg $CH_4$ $yr^{-1}$ while NGHGI report 18.5 Tg $CH_4$ $yr^{-1}$. Regarding trends, only FLExKF-TM5-4DVAR_EMPA shows a linear decreasing trend of -2.1 % $yr^{-1}$, compared to the NGHGI data trend of -1.3% $yr^{-1}$ over their overlap period of 2005-2017 while other inversions show no significant trend. From this attempt we clearly note that not so many of the inversions showed the clear

decline of NGHGI. As NGHGI emissions are dominated by anthropogenic fluxes and decline with almost 30% compared to 1990, this should be seen as well in the corrected anthropogenic inversions. Therefore, we need to further investigate how well the NGHGI reflect reality or how well the TD estimates capture the trends.

***Global inversions estimates***

Figures 5 compares TD global estimates, with NGHGI data and gives for information the wetland emissions from global wetland models (Saunois et al., 2020). We present TD estimates of total emissions (Fig. 5a) as well as estimates of anthropogenic emissions (Fig. 5b). The global inversion models were split according to the type of observations used, 11 of them using satellites (GOSAT) and 11 using surface stations (SURF). Wetlands emissions provided by 22 global TD inversions from the Global Methane Budget (Saunois et al., 2020) are post-processed with

prior ratios estimates for wetlands $CH_4$ emissions (Appendix B, Table B4).

For the common period 2010-2016 for the EU27+UK, the two ensembles of regional and global models give a total $CH_4$ emission mean (Figure 5a) of 22.6 Tg $CH_4$ $yr^{-1}$ (GOSAT) and 23.7 Tg $CH_4$ $yr^{-1}$ (SURF) compared to 19.0 ± 1.7 Tg $CH_4$ $yr^{-1}$ for NGHGI (Figure 5a). The mean of the natural wetland emissions from the global inversions is 1.3 Tg $yr^{-1}$ and partly explains the positive difference between total emissions from inversions and NGHGI

anthropogenic emissions.

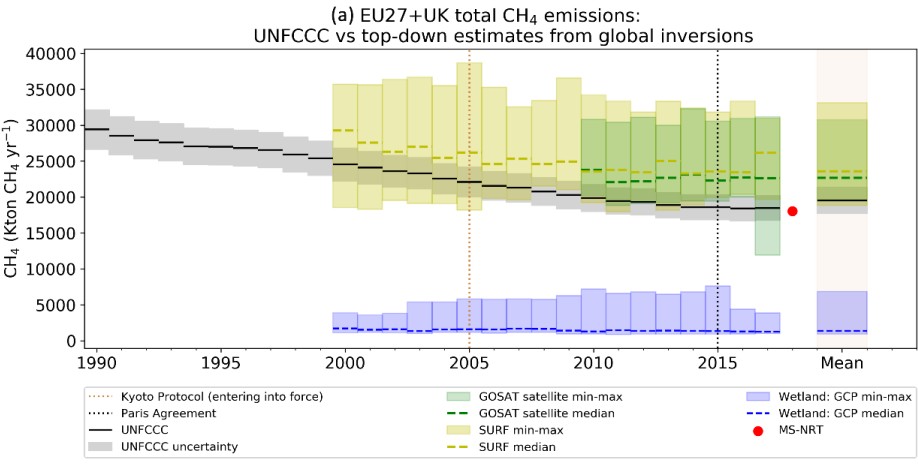

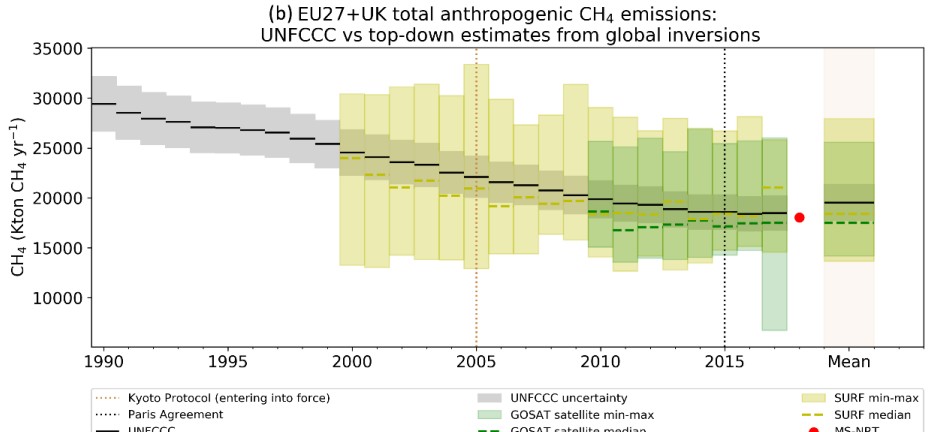

*Figure 5: a) **Total** CH₄ emissions from TD global ensembles based on surface stations (SURF) (yellow) and satellite concentration observations (GOSAT) (green) from 22 global models compared with UNFCCC NGHGI (2019) data (incl. LULUCF); b) **Anthropogenic** CH₄ emissions from top-down global inversions based on surface stations (SURF) (yellow) and on satellite concentration observations (GOSAT) (green) from different estimates. Anthropogenic emissions from these inversions were obtained by removing the sum of the natural emissions (peatland, inland waters and geological fluxes shown in figure 4a) from the total estimates. For consistency with the global data we plot the global wetland emissions from the GCP inversions (blue). UNFCCC NGHGI (2018) MS-reported uncertainty computed with the error propagation method (95% confidence interval) is 9.29 % and represents the UNFCCC NGHGI (2018) uncertainty for all sectors (incl. LULUCF). The time series mean was computed for the common period 2010-2016.*

In an attempt to quantify the European TD anthropogenic CH₄ component, in Figure 5b we subtract from the total TD CH₄ emissions once again the peatland emissions from the regional JSBACH-HIMMELI model and those from geological and inland waters sources. The reason for correcting both regional and global inversions with the European peatland emissions from the JSBACH-HIMMELI model, lays in the fact that they are in the range of the global wetland emissions estimates for Europe (Saunois et al., 2020). Their median for all years (1.43 Tg CH₄ yr$^{-1}$, averaged over 2005-2017), is close to the BU estimates of peatland emissions from the JSBACH-HIMMELI model (1.44 Tg CH₄ yr$^{-1}$, averaged over 2005-2017). For 2010-2016 common period, the two ensembles of regional and global models give an anthropogenic CH₄ emission mean (Figure 5b) of 17.4 Tg CH₄ yr$^{-1}$ (GOSAT) and 23.7 Tg CH₄ yr$^{-1}$ (SURF) compared to 19.0 ± 1.7 Tg CH₄ yr$^{-1}$ for NGHGI (Fig. 5b).

In 2017, the TD ensemble that simulates the closest anthropogenic estimate (Figure 5b) to the UNFCCC NGHGI (2019) is GOSAT, with the median of GOSAT inversions (16.4 Tg CH₄ yr$^{-1}$) falling within the uncertainty range of UNFCCC (18.4 ± 1.7 Tg CH₄ yr$^{-1}$).

Regarding trends, for total CH₄ emissions (Figure 5a), the SURF and GOSAT ensemble show a decreasing trend of -1.2 % yr$^{-1}$ and -0.6 % yr$^{-1}$, respectively, over the period covered by each of them (SURF: 2000-2016; GOSAT: 2010-2017). For anthropogenic CH₄ emissions (Figure 5b), the SURF ensemble shows a decreasing trend of -1.4 %





yr$^{-1}$ compared to -1.5 % yr$^{-1}$ for the NGHGI over 2000-2016, while the GOSAT ensemble shows a decreasing trend of -0.8 % yr$^{-1}$ compared to -0.9 % yr$^{-1}$ for the NGHGI over 2010-2017.

### 3.2 Comparing $N_2O$ anthropogenic emissions estimates from different approaches

*3.2.1. Estimates of European and regional total $N_2O$ fluxes*

Similarly, as done for $CH_4$ (section 3.1.1. and Figure 1), we present results of total $N_2O$ fluxes from EU27+UK and five main regions in Europe. Figure 7 summarizes the total $N_2O$ fluxes from NGHGI (incl. LULUCF) for both base year 1990 and mean of 2011-2015 period.

The total UNFCCC estimates include emissions from all sectors. We plot these and compare them with fluxes from global datasets, BU models and TD inversions. We note that for all five regions, the $N_2O$ emissions decreased between 29 % (Northern Europe) to 43 % (Western Europe) and for EU27+UK  37 % with respect to NGHGI 1990 value. It also shows that at regional European level, the emissions from BU (anthropogenic and natural) and TD estimates agree well with reported NGHGI data within the high uncertainty reported by UNFCCC (~80%) or observed 555 in the TD model range. This TD uncertainty is represented here by the variability in the model ensembles and denotes the range of the extremes (min and max) of estimates within each model ensemble. There is significant uncertainty in Northern Europe, where the TD estimates indicate either a source or a sink (Figure 6). The current observation network is sparse, which currently limits the capability of inverse models to quantify GHG emissions at country or regional scale.

For all other regions BU anthropogenic emissions agree well with NGHGI given uncertainties, though we note consistently higher estimates from TD regional and global models estimates. The difference is too high to be attributed to the natural emission, which is related here to inland waters as only source, and which ranges for all five regions between 0.2 – 1.3 kton $N_2O$ yr$^{-1}$. The blue bar representing the natural emissions has a lower value estimates (Maavara et al., 2019 and Lauerwald et al., 2019), while the maximum value was calculated according to Yao et al 565 2020. The higher values in Yao et al. (2020) are primarily due to $N_2O$ emissions from small streams, which are not included in Maavara et al. (2019), while both studies agree fairly well regarding $N_2O$ emissions from larger rivers (Yao et al., 2020).

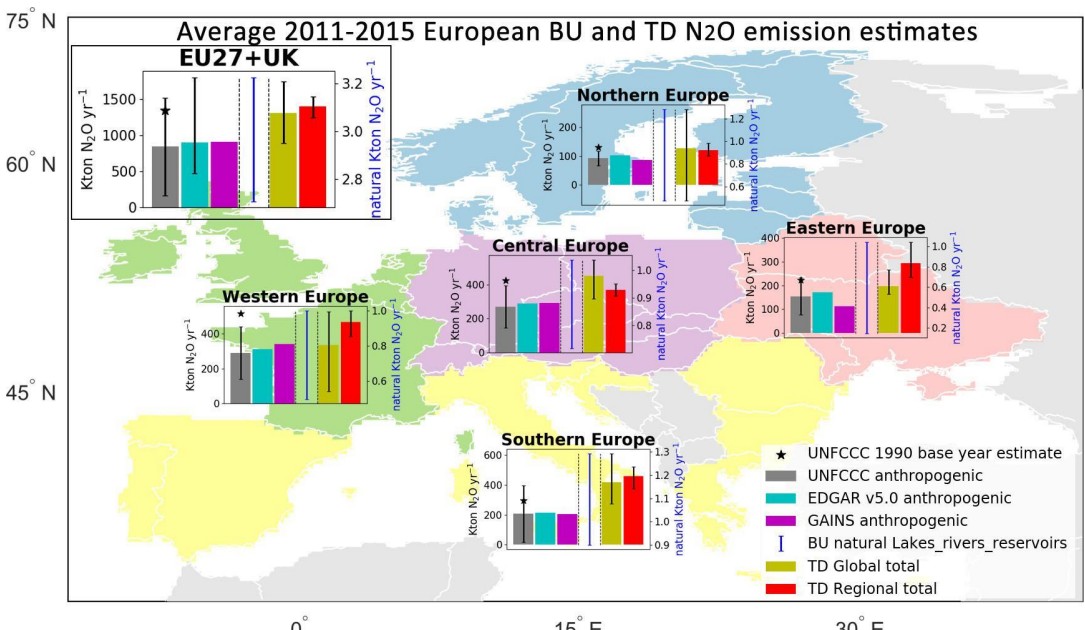

*Figure 6: Five years (2011-2015) average N$_2$O emission estimates for EU27+UK and five European regions*
*(Northern, Western Central, Southern and Eastern non-EU). Eastern European region does not include European*
*Russia and the UNFCCC uncertainty for the Republic of Moldova was not available. Northern Europe includes*
*Norway. Central Europe includes Switzerland. The data belongs to UNFCCC NGHGI (2019) submissions (grey) and*
*base year 1990 (black star), two BU inventories (GAINS and EDGAR v5.0), natural unmanaged emissions (sum of*
*peatland, geological and inland waters emissions) and three TD total estimates (regional European inversions and*
*GOSAT and SURF estimates from global inverse models). The relative error on the UNFCCC value represents the*
*UNFCCC NGHGI (2018) MS-reported uncertainty computed with the error propagation method (95% confidence*
*interval); is 80.0 % for the EU27+UK, 50.3 % for Eastern Europe non-EU, 26.6 % for Northern Europe, 91.6 % for*
*Southern Europe, 51.9 % for Western Europe and 46.0 % for Central Europe.*

### 3.2.2. NGHGI sectoral emissions and decadal changes

According to the UNFCCC (2019) NGHGI estimates for 2017 the EU27+UK emitted GHGs totaling 4.2 Gt
CO$_2$e (excl. LULUCF), of this total, N$_2$O emissions accounted for ~6% (0.2 Gt CO$_2$e or 0.8 Mt N$_2$O yr$^{-1}$) (Figure 7).
France, UK and Germany contributed together 41% of total N$_2$O emissions, respectively slightly higher than for CH$_4$
(Appendix B, Figure B1b).

The data in Figure 7 shows anthropogenic CH$_4$ emissions and their change from one decade to the next, from
UNFCCC NGHGI (2019), with the contribution from different UNFCCC sectors. In 2017, NGHGI reported
anthropogenic emissions from the EU27+UK for the four UNFCCC sectors (excl. LULUCF) (Table 1), to be 0.8 Tg
N$_2$O yr$^{-1}$. The agricultural N$_2$O emissions accounted for 76 % (± 107 %) of total EU27+UK emissions followed by

emissions from the energy sector with 12 % (± 23 %). We exclude fire emissions as they only account for 1.8 % of
the total $N_2O$ emissions in EU27+UK.

Between the 1990s and the 2000s, the net -17.7 % reduction originates largely from IPPU and agriculture
sectors, which contributed -13.5 % and -4.2 % respectively. For the period between the 2000s and 2010-2017, the net
-15.2 % reduction was again mainly attributed to the IPPU sector (-14.1 %), despite very small increases from the
LULUCF and waste sectors (+0.6 %).

We note that in 2017, the amount of emissions from the IPPU sector had already decreased by 98 % compared
to 1990 and was only 3.5 kton $N_2O$ $yr^{-1}$. Although the IPPU sector contributes in 2017 only 4% to total $N_2O$ emissions,
it was the sector with the largest reduction. IPPU sector emissions are mainly linked to the production of nitric acid
(e.g. used in fertilizer production) and adipic acid (e.g. used in nylon production). In the late 1990's and early 2000's
the five European adipic acid plants were equipped with efficient abatement technology, cutting emissions by 95-99
%, largely through voluntary agreements of the companies. Much of the remaining IPPU emissions, from nitric acid
plants, were cut in a similar manner around 2010, a development that has been connected with the introduction of the
European Emission Trading System that made it economically interesting for companies to apply emission abatement
technologies (catalytic reduction of $N_2O$ in the flue gas) to reduce their emissions.

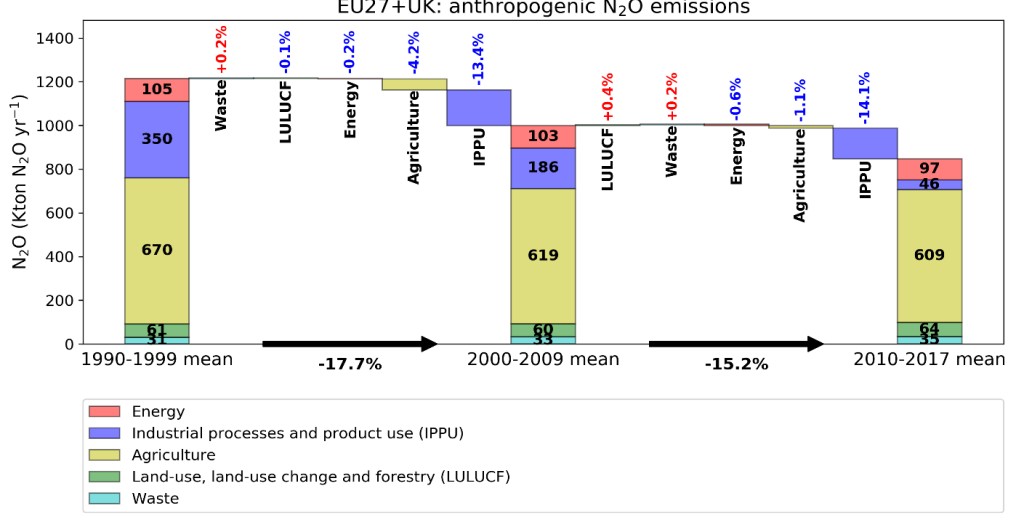

*Figure 7: The contribution of changes in $N_2O$ anthropogenic emissions in the five UNFCCC sectors to the overall*
*change in decadal mean, as reported to UNFCCC NGHGI (2019). The emissions follow the atmospheric convention,*
*where positive numbers represent an emission to the atmosphere. The three stacked columns represent the average*
*$N_2O$ emissions from each sector during three periods (1990-1999, 2000-2009 and 2010-2017) and percentages*
*represent the contribution of each sector to the total reduction percentages between periods.*


### 3.2.3. NGHGI estimates compared with bottom-up inventories

Figure 8 compares the six bottom-up inventories with UNFCCC NGHGI (2019) data, and shows that all of them fall on the NGHGI line (Figure 8a), noting that GAINS only provides emissions every five years. Each inventory shows a very good agreement with each other and the NGHGI estimates until 2005. After 2005 the slight increased

trend is influenced by the IPPU (Figure 8c) and Waste (Figure 8e) sectors, with estimates of both EDGAR v5.0 and GAINS for total anthropogenic $N_2O$ emissions in the year 2015 being 15.6 % higher than UNFCCC NGHGI (2019). For agriculture (Figure 8d) five models/inventories, show a very good match with the NGHGI. Over 1990-2015, we found linear trends of -0.7 % $yr^{-1}$ in NGHGI, GAINS and EDGAR v5.0. This provides further evidence that the sources rely on the same basic activity data from FAOSTAT and follow the IPCC EF Tier 1 or 2 approach (Petrescu et al.,

2020). In contrast, ECOSSE estimates do not use the FAO fertilizer application rate data base, but instead calculates ideal fertilizer application rates from the nitrogen demand of the crops. This means that it can severely under-estimate the applied fertilizer amounts for some areas (e.g. Netherlands, Denmark or North-West Germany), and the results are more indicative of emissions under idealized fertilizer application rates. Additionally, as mentioned above, the model simulates only the direct emissions.

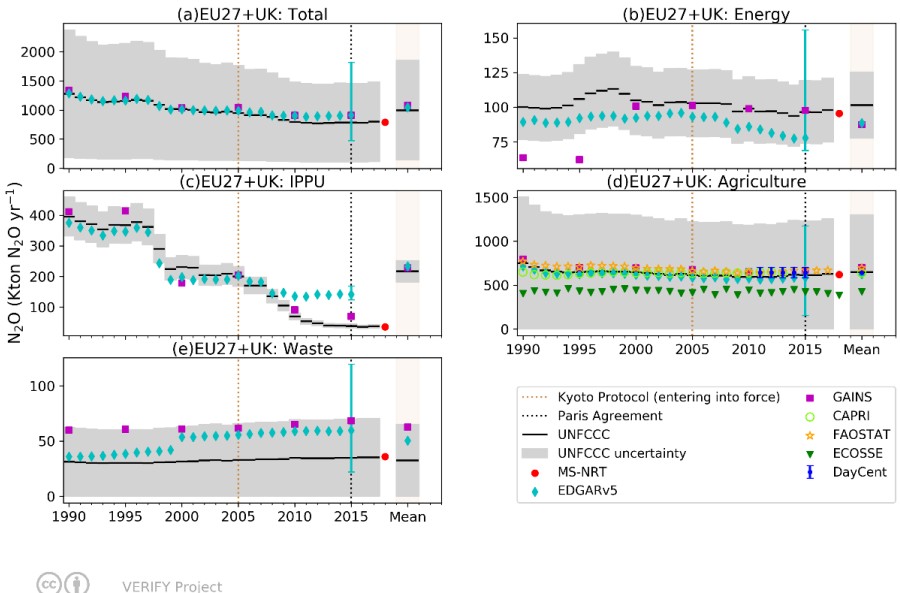


*Figure 8: EU27+UK anthropogenic $N_2O$ emissions from UNFCCC NGHGI (2019) submissions and MS-NRT 2018 compared to global BU inventories for agriculture (CAPRI, FAOSTAT, DayCent) and all sectors excl. LULUCF (EDGAR v5.0, GAINS). Sectors: Energy, IPPU, Agriculture, Waste. LULUCF not included. CAPRI reports one value for Belgium and Luxembourg. UNFCCC NGHGI (2018) MS-reported uncertainty was computed with the error*

*propagation method (95% confidence interval) and is 86 % for the total EU27+UK (excl. LULUCF), 23 % for Energy, 16 % for IPPU, 107 % for Agriculture and 626 % (in the figure 100%) for Waste. Uncertainty for EDGAR v5.0 was*





*calculated for 2015 and is min 37 % and max 73 %; it represents the 95 % confidence interval of a lognormal distribution. Last reported year in this study refers to 2017 (UNFCCC), 2015 (EDGAR v5.0), 2015 (GAINS, every 5 years).*

In the NGHGI (2018) submissions, the EU27+UK Tier 1 total uncertainty (based on the IPCC chapter 3 error propagation method described in detail by Petrescu et al., 2020) for the waste sector was 626 %. The sectoral activity responsible for this high uncertainty was the wastewater treatment and discharge (913%) and this remains one of the most uncertain sources of $N_2O$ having the highest emissions in the waste sector. Emissions are known to vary markedly in space and time even within a single wastewater treatment plant (Gruber et al., 2020), a fact that only recently has

been properly accounted for in the inventory guidelines (IPCC, 2019a). However, the total emissions from the waste sector account for only 4.4 % of the total EU27+UK $N_2O$ emissions (excl. LULUCF).

### *3.3.4. NGHGI estimates compared to atmospheric inversions*

Figure 9 compares inversion estimates of total regional (FLEXINVERT_NILU) and global (three models) $N_2O$ inversions with UNFCCC NGHGI (2019). The min-max range of all inversions is within the 2-sigma uncertainty

of NGHGI, with the median of global inversions being on average 42 % or 0.4 Tg $N_2O$ yr$^{-1}$ higher than NGHGI. Over the period 2005 – 2017, the regional FLEXINVERT_NILU is 65 % higher than UNFCCC NGHGI (2019). From the three global inversions, two show consistently higher estimates (MIROC4-ACTM and TOMCAT) as well as high variability. Regarding trends, only FLEXINVERT_NILU shows a decreasing trend of -2.1 % yr$^{-1}$ over 2005-2017, compared to UNFCCC NGHGI (2019) decreasing trend of -1.2 % yr$^{-1}$. The global PYVAR inversion agrees best in

its mean value (1.0 Tg $N_2O$ yr$^{-1}$) with the UNFCCC estimate (0.9 Tg $N_2O$ yr$^{-1}$) but not in its trend. The higher emissions from the TD estimates could be attributed to the seasonal cycle (e.g. fertilizer application) not accounted for in the NGHGI reporting.

For the $N_2O$ we do not present the corrected anthropogenic value because the only natural flux, from inland waters, is very low (2.7 kton $N_2O$ yr$^{-1}$) and when subtracted from the 4 inversions the change is almost negligible. Part

of the inland water natural estimate is considered anthropogenic in Europe and is due to the leaching of N-fertilizers from agriculture. It accounts for 66 % (Yao pers. comm.) of the total inland waters emissions.



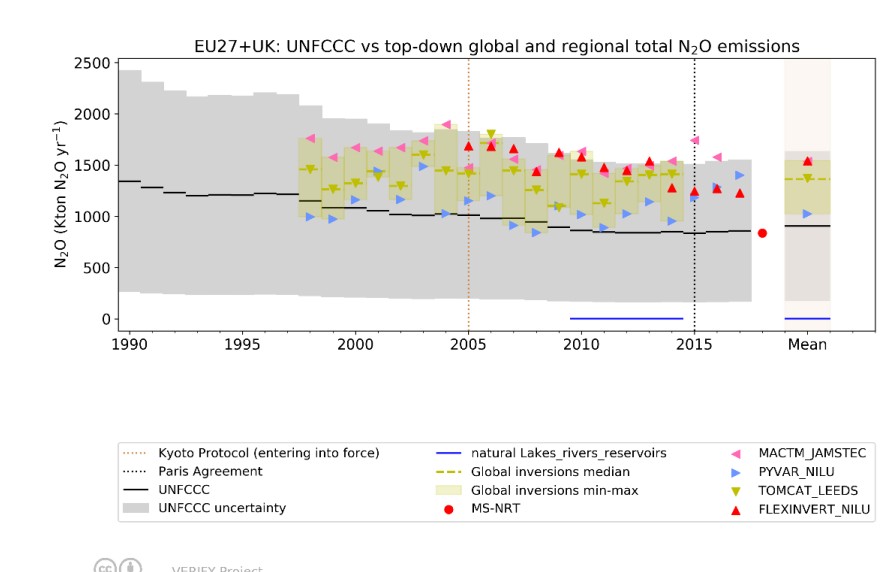

*Figure 9: Total N$_2$O emissions from UNFCCC NGHGI (2019) (incl. LULUCF) and MS-NRT 2018, compared to FLEXINVERT regional inversion over Europe and GCP global inversions (TOMCAT, PYVAR and MIROC4-ACTM), In blue the natural inland waters (lakes_rivers_reservoirs_ULB) N$_2$O emissions. UNFCCC NGHGI (2018) reported uncertainty computed with the error propagation method (95% confidence interval) is 80 % and represents the UNFCCC NGHGI (2018) uncertainty for all sectors (incl. LULUCF). Uncertainty for EDGAR v5.0 was calculated for 2015 and is min 37 % and max 73 %; it represents the 95 % confidence interval of a lognormal distribution. Last reported year in this study refers to 2014 (TOMCAT), 2016 (MIROC4-ACTM), 2017 (UNFCCC NGHGI (2019), FLEXINVERT and PYVAR), and 2018 (MS-NRT). For MS-NRT, the following countries are missing information from the LULUCF sector: Austria, Estonia, Croatia, Hungary, Latvia, Malta, Slovenia. The time series mean for TD products was computed for the overlapping period 2005-2014.*

## 4. Data availability

All raw data files reported in this work which were used for calculations and figures are available for public download at https://doi.org/10.5281/zenodo.4288969 (Petrescu et al., 2020). The data we submitted are reachable with one click (without the need for entering login and password), with a second click to download the data, consistent with the two click access principle for data published in ESSD (Carlson and Oda, 2018). The data and the DOI number are subject to future updates and only refers to this version of the paper.

## 5. Summary and concluding remarks

This study represents the first comprehensive European verification that compares total and sectoral European CH$_4$ and N$_2$O emission estimates from BU (anthropogenic and natural) with TD estimates in order to assess their use for verification purposes with the UNFCCC NGHGI reporting. Above, in the results sections, we discussed





differences between estimates. Identification of source specific uncertainty is key in understanding these differences and will lead to the reduction of the overall uncertainty in GHG inventories. More specifically, we present the first EU27+UK and European regional 2011-2015 averaged results for $CH_4$ (Figure 1) and $N_2O$ (Figure 6) compared to the NGHGI emissions (incl. LULUCF) for the same period and the one reported in 1990, in the framework of the future global stock-take estimate.

Regarding sources of inconsistencies between $CH_4$ BU estimates and NGHGI data (Figure 3), at EU27+UK level they are mainly caused by the use of different methodologies in calculating emissions as highlighted in Petrescu et al. (2020). Both BU inventories and the NGHGI use similar activity data and the default EFs reported in the IPCC 2006 guidelines make all data sources agree rather well, thus inventory spread may not be indicative of the uncertainty. For global consistency purposes, EDGAR v5.0 mostly uses Tier 1 approaches in calculating emissions and uncertainties, a fact which triggers differences with other data sources (GAINS for all sectors and CAPRI for agriculture). Within the UNFCCC reporting process, the two most important emission sectors after agriculture are energy and waste and contribute to the highest reduction percentages (see Figure 1). Another reason for small inconsistencies between datasets is the allocation of emissions to different sectors, different data sources using different versions of IPCC guidelines (e.g. EDGAR v5.0).

For $N_2O$ anthropogenic emissions, all BU data sources show good agreement with the UNFCCC NGHGI (2019) data in both trends and means (Figure 8) and agriculture remains the largest emitter (e.g. urea fertilizers). The uncertainties reported by NGGI are very large and will need further improvement.

Regarding the TD estimates, our exercise shows that comparison between $CH_4$ inversions estimates and NGHGI is highly uncertain because of the large spread in the inversion results. As TD inversions do not fully distinguish between all emission sectors used by NGHGI and report either total emissions or a coarse sectorial partitioning, their comparison to NGHGI is only possible for total emissions. It is also necessary to make an adjustment for natural emissions, which are included in TD inversions but not reported by the NGHGIs. However, the natural $N_2O$ emissions does not explain the difference between BU and TD (452 kton $N_2O$) and more research is needed to identify the source of discrepancies.

Some studies (Fronzek et al., 2018) showed that ensembles work well in simulating variables with high uncertainty. In general regional inversions show less spread then the global inversions as they used recent updates of transport models and higher resolution transport. Total $CH_4$ from regional inversions show a min-max range of 8.7 Tg $CH_4$ $yr^{-1}$ compared to 12.4 Tg $CH_4$ $yr^{-1}$ from global GOSAT inversions and 13.5 Tg $CH_4$ $yr^{-1}$ from global SURF inversions (Figures 4 and 5). The global models are less well constrained as they have lower resolution (hence larger representation errors) and often use fewer observations.

A key challenge for the inversion $CH_4$ community remains the separation of emissions in specific source sectors, as derived total emissions may also include natural emissions (or removals), while in the case of $N_2O$ this won't be possible due to the use of definitions (e.g. "natural" $N_2O$ emissions are defined as the level of emission in the pre-industrial period). It is therefore not possible to separate between $N_2O$ from natural or anthropogenic sources because natural (or unmanaged) soils can have both natural and anthropogenic emissions, while anthropogenic (managed) agricultural soils can also have a level of natural emissions according to the definition of natural. Therefore,





the goal of TD inversions for estimating $N_2O$ emissions should mainly focus on the trends. Furthermore, the accuracy of derived emissions and the spatial scales at which emissions can be estimated depend on the quality and density of measurements and the quality of the atmospheric models (Bergamaschi et al., 2018b). Significant further developments of the global observations system and the top-down methods would be required to support the implementation of the Paris Agreement.

We provide for EU27+UK a consolidated synthesis of the relative uncertain $CH_4$ and $N_2O$ emissions making use of consistently derived BU and TD estimates over the region of Europe, which might illustrate the importance of regional consistent analyses that form the basis of the Multilateral Facilitative Consideration of Progress under the enhanced transparency framework of the Paris Agreement. However, the implementation of the Paris Agreement requires accurate quantification of GHG emissions in order to track the progress of all parties with their "Nationally Determined Contributions" and to assess collective progress towards achieving the purpose of this Agreement and its long-term goals (stocktake). As this will be mainly achieved and build upon BU methodologies developed by the IPCC, we need to take into consideration the potential to quantify GHG emissions by using "top-down" methods ("inverse modelling") (Bergamaschi et al., 2018b). One advantage of the inverse estimate is that it provides total emission estimates. Therefore, the capability to quantify anthropogenic emissions depends on the magnitude of natural sources and sinks and the capability to quantify them.

As stated in the introduction, our aim was to identify in this synthesis the issues which cause the differences between NGHGI, BU and TD to further improve and build a pathway to a verification system (BU use of activity data, emission factor and emission allocation ($CO_2$ fossil and $CH_4$), very large NGHGI reported uncertainties which need to be reassessed ($N_2O$) and higher TD estimates then inventories ($CH_4$ and $N_2O$).

## 6. Appendices

### Appendix A: Data sources, methodology and uncertainty descriptions

The country specific plots are found at: http://webportals.ipsl.jussieu.fr/VERIFY/FactSheets/ v1.24

### VERIFY project

VERIFY's primary aim is to develop scientifically robust methods to assess the accuracy and potential biases in national inventories reported by the parties through an independent pre-operational framework. The main concept is to provide observation-based estimates of anthropogenic and natural GHG emissions and sinks as well as associated uncertainties. The proposed approach is based on the integration of atmospheric measurements, improved emission inventories, ecosystem data, and satellite observations, and on an understanding of processes controlling GHG fluxes (ecosystem models, GHG emission models).

Two complementary approaches relying on observational data-streams will be combined in VERIFY to quantify GHG fluxes:

1) atmospheric GHG concentrations from satellites and ground-based networks (top-down atmospheric inversion models) and

2) bottom-up activity data (e.g. fuel use and emission factors) and ecosystem measurements (bottom-up models).

For $CO_2$, a specific effort will be made to separate fossil fuel emissions from ecosystems fluxes. For $CH_4$ and $N_2O$, we will separate agricultural from fossil fuel and industrial emissions. Finally, trends in the budget of the three GHGs will be analysed in the context of NDC targets.

The objectives of VERIFY are:

**Objective 1**. Integrate the efforts between the research community, national inventory compilers, operational centres in Europe, and international organisations towards the definition of future international standards for the verification of GHG emissions and sinks based on independent observation.

**Objective 2**. Enhance the current observation and modelling ability to accurately and transparently quantify the sinks and sources of GHGs in the land-use sector for the tracking of land-based mitigation activities.

**Objective 3.** Develop new research approaches to monitor anthropogenic GHG emissions in support of the EU commitment to reduce its GHG emissions by 40% by 2030 compared to the year 1990.

**Objective 4.** Produce periodic scientific syntheses of observation-based GHG balance of EU countries and practical policy-oriented assessments of GHG emission trends, and apply these methodologies to other countries.

For more information on project team and products/results check https://verify.lsce.ipsl.fr/.


Table A: *Country grouping use for reconciliation purposes between BU and TD emissions.*

| Country name – geographical Europe | BU-ISO3 | Aggregation from TD-ISO3 |
|---|---|---|
| Luxembourg | LUX | |
| Belgium | BEL | BENELUX |
| Netherlands | NLD | BNL |
| Bulgaria | BGR | BGR |
| Switzerland | CHE | |
| *Lichtenstein* | *LIE* | *CHL* |
| Czech Republic | CZE | Former Czechoslovakia |
| Slovakia | SVK | CSK |
| Austria | AUT | AUT |
| Slovenia | SVN | North Adriatic countries |
| Croatia | HRV | NAC |
| Romania | ROU | ROU |
| Hungary | HUN | HUN |
| Estonia | EST | |
| Lithuania | LTU | Baltic countries |



| Latvia | LVA | BLT |
|---|---|---|
| Norway | NOR | NOR |
| Denmark | DNK | |
| Sweden | SWE | |
| Finland | FIN | DSF |
| Iceland | ISL | ISL |
| Malta | MLT | MLT |
| Cyprus | CYP | CYP |
| France (Corsica incl.) | FRA | FRA |
| *Monaco* | *MCO* | |
| *Andorra* | *AND* | |
| Italy (Sardinia, Vatican incl.) | ITA | ITA |
| *San Marino* | *SMR* | |
| United Kingdom (Great Britain + N Ireland) | GBR | UK |
| *Isle of Man* | *IMN* | |
| Iceland | | |
| Ireland | IRL | IRL |
| Germany | DEU | DEU |
| Spain | ESP | IBERIA |
| Portugal | PRT | IBE |
| Greece | GRC | GRC |
| *Russia (European part)* | *RUS European* | |
| *Georgia* | *GEO* | *RUS European+GEO* |
| *Russian Federation* | *RUS* | *RUS* |
| Poland | POL | POL |
| *Turkey* | *TUR* | *TUR* |
| EU27+UK (Austria, Belgium, Bulgaria, Cyprus, Czech Republic, Germany, Denmark, Spain, Estonia, Finland, France, Greece, Croatia, Hungary, Ireland, Italy, Lithuania, Latvia, Luxembourg, Malta, Netherlands, Poland, Portugal, Romania, Slovakia, Slovenia, Sweden, United Kingdom) | AUT, BEL, BGR, CYP, CZE, DEU, DNK, ESP, EST, FIN, FRA, GRC, HRV, HUN, IRL. ITA, LTU, LVA, LUX, MLT, NDL, POL, PRT, ROU, SVN, SVK, SWE, GBR | E28 |



| | | |
|---|---|---|
| Western Europe (Belgium, France, United Kingdom, Ireland, Luxembourg, Netherlands) | BEL, FRA, UK, IRL, LUX, NDL | WEE |
| Central Europe (Austria, Switzerland, Czech Republic, Germany, Hungary, Poland, Slovakia) | AUT, CHE, CZE, DEU, HUN, POL, SVK | CEE |
| Northern Europe (Denmark, Estonia, Finland, Lithuania, Latvia, Norway, Sweden) | DNK, EST, FIN, LTU, LVA, NOR, SWE | NOE |
| *South-Western Europe (Spain, Italy, Malta, Portugal)* | *ESP, ITA, MLT, PRT* | *SWN* |
| *South-Eastern Europe (all) (Albania, Bulgaria, Bosnia and Herzegovina, Cyprus, Georgia, Greece, Croatia, Macedonia, the former Yugoslav, Montenegro, Romania, Serbia, Slovenia, Turkey)* | *ALB, BGR, BIH, CYP, GEO, GRC, HRV, MKD, MNE, ROU, SRB, SVN, TUR* | *SEE* |
| *South-Eastern Europe (non-EU) (Albania, Bosnia and Herzegovina, Macedonia, the former Yugoslav, Georgia, Turkey, Montenegro, Serbia)* | *ALB, BIH, MKD, MNE, SRB, GEO, TUR* | *SEA* |
| *South-Eastern Europe (EU) (Bulgaria, Cyprus, Greece, Croatia, Romania, Slovenia)* | *BGR, CYP, GRC, HRV, ROU, SVN* | *SEZ* |
| *Southern Europe (all) (SOE) (Albania, Bulgaria, Bosnia and Herzegovina, Cyprus, Georgia, Greece, Croatia, Macedonia, the former Yugoslav, Montenegro, Romania, Serbia, Slovenia, Turkey, Italy, Malta, Portugal, Spain)* | *ALB, BGR, BIH, CYP, GEO, GRC, HRV, MKD, MNE, ROU, SRB, SVN, TUR, ITA, MLT, PRT, ESP* | *SOE* |
| *Southern Europe (non-EU) (SOY) Albania, Bosnia and Herzegovina, Georgia, Macedonia, the former Yugoslav, Montenegro, Serbia, Turkey)* | *ALB, BIH, GEO, MKD, MNE, SRB, TUR,* | *SOY* |
| Southern Europe (EU) (SOZ) (Bulgaria, Cyprus, Greece, Croatia, Romania, Slovenia, Italy, Malta, Portugal, Spain) | BGR, CYP, GRC, HRV, ROU, SVN, ITA, MLT, PRT, ESP | SOZ |
| Eastern Europe (non-EU) (Belarus, Moldova, Republic of, Russian Federation, Ukraine) | BLR, MDA, RUS, UKR | EAE |
| *EU-15 (Austria, Belgium, Germany, Denmark, Spain, Finland, France, United Kingdom, Greece, Ireland, Italy, Luxembourg, Netherlands, Portugal, Sweden)* | *AUT, BEL, DEU, DNK, ESP, FIN, FRA, GBR, GRC, IRL, ITA, LUX, NDL, PRT, SWE* | *E15* |



| | | |
|---|---|---|
| EU-27 (Austria, Belgium, Bulgaria, Cyprus, Czech Republic, Germany, Denmark, Spain, Estonia, Finland, France, Greece, Croatia, Hungary, Ireland, Italy, Lithuania, Latvia, Luxembourg, Malta, Netherlands, Poland, Portugal, Romania, Slovakia, Slovenia, Sweden) | AUT, BEL, BGR, CYP, CZE, DEU, DNK, ESP, EST, FIN, FRA, GRC, HRV, HUN, IRL. ITA, LTU, LVA, LUX, MLT, NDL, POL, PRT, ROU, SVN, SVK, SWE | E27 |
| All Europe (Aaland Islands, Albania, Andorra, Austria, Belgium, Bulgaria, Bosnia and Herzegovina, Belarus, Switzerland, Cyprus, Czech Republic, Germany, Denmark, Spain, Estonia, Finland, France, Faroe Islands, United Kingdom, Guernsey, Greece, Croatia, Hungary, Isle of Man, Ireland, Iceland, Italy, Jersey, Liechtenstein, Lithuania, Luxembourg, Latvia, Moldova, Republic of, Macedonia, the former Yugoslav, Malta, Montenegro, Netherlands, Norway, Poland, Portugal, Romania, Russian Federation, Svalbard and Jan Mayen, San Marino, Serbia, Slovakia, Slovenia, Sweden, Turkey, Ukraine) | ALA, ALB, AND, AUT, BEL, BGR, BIH, BLR, CHE, CYP, CZE, DEU, DNK, ESP, EST, FIN, FRA, FRO, GBR, GGY, GRC, HRV, HUN, IMN, IRL, ISL, ITA, JEY, LIE, LTU, LUX, LVA, MDA, MKD, MLT, MNE, NDL, NOR, POL, PRT, ROU, RUS, SJM, SMR, SRB, SVK, SVN, SWE, TUR, UKR | EUR |

*countries highlighted in *italic* are not discussed in the current 2019 synthesis mostly because unavailability of NGHGI data (non-Annex I countries[11]) but are present on the web-portal: http://webportals.ipsl.jussieu.fr/VERIFY/FactSheets/. Results of Annex I countries (NOR, CHE, ISL) and non-EU EAE countries/groups are represented in Figure 4.



---

[11]Non-Annex I countries are mostly developing countries. The reporting to UNFCCC is implemented through national communications (NCs) and biennial update reports (BURs): https://unfccc.int/national-reports-from-non-annex-i-parties.





*Table AA: Main methodological changes (**in bold**) of current study with respect to Petrescu et al., 2020; n/a cells*

*mean that there is no data available.*

| Publication year | Gas | Bottom-up anthropogenic CH₄ / N₂O emissions | | | Bottom-up natural CH₄ / N₂O emissions | Top-down CH₄ / N₂O emissions | | Uncertainty and other changes |
|---|---|---|---|---|---|---|---|---|
| | | Inventories | Global databases | Emission models | Emission models | Regional models | Global models | |
| 2020<br><br>Petrescu et al. (2020) AFOLU bottom-up synthesis | CH₄ | National emissions from UNFCCC (2018) 1990-2016<br><br>*AFOLU sector (Agriculture and LULUCF)* EU28 data for four years (1990, 2005, 2010 and 2016) | EGDAR v4.3.2 1990-2012<br><br>EDGAR FT2017 1990-2016<br><br>FAOSTAT 1990-2016<br><br>*Agriculture sector* EU28 data for four years (1990, 2005, 2010 and last reported year) | CAPRI 1990-2013<br><br>GAINS 1990-2015<br><br>*Agriculture sector* EU28 data for four years (1990, 2005, 2010 and last reported year) | Natural (wetlands) CH₄ emissions model ensemble GCP (2018) Puolter et al. (2017)<br><br>Time series 1990-2017 | n/a | n/a | UNFCCC (2018) uncertainty estimates for 2016 (error propagation 95% interval method)<br><br>EDGAR v.4.3.2. reports only for 2012 |
| | N₂O | National emissions from UNFCCC (2018) 1990-2016<br><br>*Agriculture sector* EU28 data for four years (1990, 2005, 2010 and 2016) | EGDAR v4.3.2 1990-2012 EDGAR FT2017 1990-2016 FAOSTAT 1990-2016<br><br>*Agriculture sector* EU28 data for four years (1990, 2005, 2010 and last reported year) | CAPRI 1990-2013<br><br>GAINS 1990-2015<br><br>*Agriculture sector* EU28 data for four years (1990, 2005, 2010 and last reported year) | n/a | n/a | n/a | UNFCCC (2018) uncertainty estimates for 2016 EDGAR v.4.3.2. reports only for 2012 |
| **2021**<br><br>this study synthesis bottom-up **and top-down** | CH₄ | National emissions from UNFCCC (**2019**) 1990-**2017**<br><br>***All UNFCCC sectors EU27+UK time series* and 2018 MS-NRT** | EGDAR v 5.0 1990-**2015**<br><br>FAOSTAT (**only agriculture**) 1990-**2017**<br><br>*Anthropogenic EU27+UK time series* (excl. LULUCF) | CAPRI 1990-2013<br><br>GAINS 1990-2015<br><br>*Agriculture sector EU27+UK* **Times series** | **Non-wetland inland waters Average 2005-2011**<br><br>**Geological fluxes Total pre-industrial era**<br><br>**JSBACH-HIMMELI 2005-2017** | **Total CH₄ column Time series 2005-2017: FLEXPART - FLExKF**<br><br>**TM5-4DVAR**<br><br>**FLEXINVERT _NILU**<br><br>**CTE-FMI** | **Anthropogenic and natural partitions**<br><br>**GCP-GCB 2019 2000-2017** | UNFCCC (2018) uncertainty estimates for 2016 (error propagation 95% interval method)<br><br>EDGAR v.4.3.2. reports only for **2015**<br><br>**For model ensembles reported as variability in extremes (min/max)** |





| | | | | | | | |
|---|---|---|---|---|---|---|---|
| | | estimate (EEA, 2019) *Regional EU27+UK totals* (incl. NOR, CHE, UKR, MLD and BLR) | **Regional EU27+UK totals** (incl. NOR, CHE, UKR, MLD and BLR) Excl. LULUCF | | | **InTEM-NAME Only for UK** **InGOS inversions 2006-2012** | |
| | $N_2O$ | National emissions from UNFCCC (2019) 1990-2017 ***All UNFCCC sectors EU27+UK time series and 2018 MS-NRT estimate (EEA, 2019)*** *Regional EU27+UK totals* **(incl. NOR, CHE, UKR, MLD and BLR)** | EGDAR v 5.0 1990-**2015** (excl. LULUCF) FAOSTAT **(only agriculture)** 1990-**2017** *Anthropogenic EU27+UK time series* **Regional EU27+UK totals** (incl. NOR, CHE, UKR, MLD and BLR) Excl. LULUCF | *Agriculture* CAPRI 1990-2013 **ECOSSE 1990-2018** | **$N_2O$ missions from lakes, rivers, reservoirs Average 2010-2014** | **Total $N_2O$ column Time series FLEXINVERT _NILU 2005-2017** | **Total $N_2O$ column Time series GCP - GN₂OB 2019 PYVAR TOMCAT MIROC4-ACTM 1998-2016** | UNFCCC (2018) uncertainty estimates for 2016 (error propagation 95% interval method) EDGAR v.4.3.2. reports only for **2015** **For model ensembles reported as variability in extremes (min/max)** |

## A1: Anthropogenic CH₄ emissions (sectors Energy, IPPU, Agriculture, LULUCF and Waste)

*Bottom-up CH₄ emissions estimates*

*UNFCCC NGHGI (2019)*

Under the UNFCCC convention and its Kyoto Protocol national greenhouse gas (GHG) inventories are the most important source of information to track progress and assess climate protection measures by countries. In order

to build mutual trust in the reliability of GHG emission information provided, national GHG inventories are subject to standardized reporting requirements, which have been continuously developed by the Conference of the Parties (COP)[12]. The calculation methods for the estimation of greenhouse gases in the respective sectors is determined by the methods provided by the 2006 IPCC Guidelines for National Greenhouse Gas Inventories (IPCC, 2006). These Guidelines provide detailed methodological descriptions to estimate emissions and removals, as well as provide

recommendations to collect the activity data needed. As a general overall requirement, the UNFCCC reporting

---

[12] The last revision has been made by COP 19 in 2013 (UNFCCC, 2013)



guidelines stipulate that reporting under the Convention and the Kyoto Protocol must follow the five key principles of transparency, accuracy, completeness, consistency and comparability (TACCC). The reporting under UNFCCC shall meet the TACCC principles. The three main GHGs are reported in time series from 1990 up to two years before the due date of the reporting. The reporting is strictly source category based and is done under the Common Reporting

Format tables (CRF), downloadable from the UNFCCC official submission portal:

https://unfccc.int/process-and-meetings/transparency-and-reporting/reporting-and-review-under-the-convention/greenhouse-gas-inventories-annex-i-parties/national-inventory-submissions-2019

The UNFCCC NGHGI anthropogenic $CH_4$ emissions include estimates from 4 key sectors for the EU27+UK: 1 Energy, 2 Industrial processes and product use (IPPU), 3 Agriculture and 5 Waste. The tiers method a country

applies depends on the national circumstances and the individual conditions of the land, which explains the variability of uncertainties among the sector itself as well as among EU countries. The LULUCF $CH_4$ emissions are very small but are included in some figures (see Table 1).

**Uncertainty** methodology for the NGHGI UNFCCC submissions are based on the Chapter 3 of 2006 IPCC Guidelines for National Greenhouse Gas Inventories and is explained in Appendix B of Petrescu et al., 2020.


*EDGAR v5.0*

The Emissions Database for Global Atmospheric Research (EDGAR) is an independent global emission inventory of greenhouse gases (GHG) and air pollutants developed by the Joint Research Centre of the European Commission (https://edgar.jrc.ec.europa.eu/index.php). The non-$CO_2$ component in EDGAR v5.0 covers a long time

series of emissions starting in 1970 till the t-4. $CH_4$ emissions are estimated for all anthropogenic emission sectors with the exception of Land Use, Land Use Change and Forestry (LULUCF) at country and annual level in a consistent and comparable way for all world countries. Emissions are computed using activity data from international statistics (e.g. IEA (2017), FAO (2017), USGS (2019), etc.), emission factors from IPCC guidelines (IPCC, 2006) and scientific literature, technology and abatement measures incorporation. Once the emission database is compiled for all countries,

sectors and pollutants, annual emission data are disaggregated to monthly emissions applying sector and country-specific yearly emission profiles (Crippa et al., 2019a). In addition, monthly emissions are spatially distributed over global gridmaps with a resolution of 0.1x0.1 degree making use of sector specific spatial proxies (Janssens-Maenhout et al., 2019).

The latest version of the EDGAR database EDGARv5.0 contains estimated $CH_4$ emissions from 1970 till

2015 based fully on statistical data (https://edgar.jrc.ec.europa.eu/overview.php?v=50_GHG). EDGAR v5.0 updated waste emissions were quantified using the First-Order-Decay method, combining nationally defined inputs (for waste generation rates and compositions) and IPCC's regional default values for parameters associated with waste degradation processes (specific mass of biodegradable organic carbon, the methane volumetric fraction in the obtained landfill gas and the half life time for each waste component). The total landfilled waste was split into six

streams: food and organic waste type, paper and cardboard, textiles, rubber, wood, sludge and similar effluents. This was done, making use of the EUROSTAT waste database (Eurostat 2020) but also by employing data from waste composition for municipal/household type waste from Silpa (Silpa et al., 2018).



**Uncertainties:** The methodological description is explained in detail in Appendix B Petrescu et al., 2020, and Solazzo et al., 2020 (in review ACP).


*CAPRI*

CAPRI is an economic, partial equilibrium model for the agricultural sector, focused on the EU (Britz and Witzke, 2014[13]; Weiss and Leip, 2012[14]). CAPRI stands for 'Common Agricultural Policy Regionalised Impact analysis', and the name hints at the main objective of the system: assessing the effect of CAP policy instruments not only at the EU or Member State level but at sub-national level. The model is calibrated for the base year (currently 2012) and then baseline projections are built, allowing the ex-ante evaluation of agricultural policies and trade policies on production, income, markets, trade and the environment.

Among other environmental indicators, CAPRI simulates $CH_4$ emissions from agricultural production activities (enteric fermentation, manure management, rice cultivation, agricultural soils). Activity data is mainly based on FAOSTAT and EUROSTAT statistics and estimation of emissions follows IPCC 2006 methodologies, with a higher or lower level of detail depending on the importance of the emission source. Details on CAPRI methodology for emissions calculations is referenced in the Annex Table A1.

**Uncertainties** are not available for the CAPRI estimates.

*GAINS*

Specific sectors and abatement technologies in GAINS vary by the specific emitted compound, with source sector definition and emission factors largely following the IPCC methodology at the Tier 1 or Tier 2 level. GAINS includes in general all anthropogenic emissions to air, but does not cover emissions from forest fires, savannah burning and land use / land use change. Emissions are estimated for 174 countries/regions, with the possibility to aggregate to a global emission estimate, and spanning a timeframe from 1990 to 2050 in five-year intervals. Activity drivers for macroeconomic development, energy supply and demand, and agricultural activities are entered externally, GAINS extends with knowledge required to estimate "default" emissions (emissions occurring due to an economic activity without emission abatement) and emissions and costs of situations under emission control (see Amann et al., 2001).

The GAINS model covers all source sectors of anthropogenic methane ($CH_4$) emissions; agricultural sector emissions from livestock, rice cultivation and agricultural waste burning, energy sector emissions from upstream and downstream sources in fossil extraction and use, and emissions from handling and treatment of solid waste and wastewater source sectors. A description of the modelling of $CH_4$ emissions in GAINS is presented in Höglund-Isaksson (2020). Generation of solid waste and the carbon content of wastewater are derived within the model in consistency with the relevant macroeconomic scenario. The starting point for estimations of anthropogenic $CH_4$ is the methodology recommended in the IPCC (2006 and revision in 2019) guidelines, for most source sectors using country-specific information to allow for deriving country- and sector/technology- specific emission factors at a Tier 2 level. Consistent methodologies were further developed to estimate emissions from oil and gas systems (Höglund-Isaksson,

---

[13] https://www.capri-model.org/docs/CAPRI_documentation.pdf

[14] https://www.sciencedirect.com/science/article/pii/S0167880911004415





2017) and solid waste (Höglund-Isaksson, 2018; Gómez-Sanabria et al., 2018). Emission factors are specified in a consistent manner across countries for given sets of technology and with past implementation of emission abatement measures reflected as changes in technology structures. The resulting emission estimates are well comparable across

geographic and temporal scales. The GAINS approach to calculate waste emissions is developed in consistency with the First-Order-Decay method recommended by IPCC (2006 and 2019 revision), applying different decay periods when estimating emissions from flows of different types of organic waste, i.e., food & garden, paper, wood, textile and other. Data on waste generation, composition and treatment are taken from EUROSTAT (2019) and complemented with national information from the UNFCCC (2019) Common Reporting Format tables on the amounts

of waste diverted to landfills of various management levels and to treatment e.g., recycling, composting, biodigestion and incineration.

**Uncertainties:** Uncertainty is prevalent among many different dimensions both in the estimations of emissions, abatement potentials and costs. When constructing global bottom-up emission inventories at a detailed country and source level, it is inevitable that some information gaps will be bridged using default assumptions. As it is difficult to

speculate about how such sources of uncertainty affect resulting historical and future emission estimates, we instead address uncertainty in historical emissions by making comparisons to estimates by other publicly available and independently developed bottom-up inventories and various top-down estimates consistent with atmospheric measurements and inverse model results. Although existing publicly available global bottom-up inventories adhere to the recommended guidelines of the IPCC (2006), the flexibility in these is large and results will depend on the

availability and quality of gathered source information. There is accordingly a wide range of possible sources of uncertainty built into estimations in such comprehensive efforts. Having a pool of independently developed inventories, each with its own strengths and weaknesses, can improve the understanding of the scope for uncertainty, in particular when compared against top-down atmospheric measurements.

*FAOSTAT*

FAOSTAT: The Food and Agricultural Organization of the United Nations (FAO), provides $CH_4$ emissions from agriculture available at: http://www.fao.org/faostat/en/#data/GT/visualize, and emissions from land use, available at http://www.fao.org/faostat/en/#data/GL/visualize. The FAOSTAT emissions database is computed following Tier 1 IPCC 2006 Guidelines for National GHG Inventories (http://www.ipcc-

nggip.iges.or.jp/public/2006gl/index.html). Country reports to FAO on crops, livestock and agriculture use of fertilizers are the source of activity data. Geospatial data are the source of AD for the estimates from cultivation of organic soils, biomass and peat fires. GHG emissions are provided by country, regions and special groups, with global coverage, relative to the period 1961-present (with annual updates, currently to 2017) and with projections for 2030 and 2050, expressed as $CO_2e$ for $CH_4$, by underlying agricultural emission sub-domain and by aggregate (agriculture

total, agriculture total plus energy, agricultural soils). LULUCF emissions consist of $CH_4$ (methane) associated with biomass and peat fires.

**Uncertainties** were computed by Tubiello et al., 2013 but are not available in the FAOSTAT database.



### *Top-down CH₄ emission estimates*

#### *FLEXPART – FLExKF*

FLExKF applies an Extended Kalman Filter (Brunner et al. 2012) in combination with backward Lagrangian transport simulations using the model FLEXPART (Stohl et al. 2005; Pisso et al. 2019). It optimizes surface-atmosphere fluxes by assimilating atmospheric observations in a sequential manner, which allows for an analytical solution for relatively large inversion problems (long time periods, number stations O(100)). Since model-observation residuals typically follow a log-normal distribution, the method optimizes log-transformed emissions, which also

guarantees a positive solution. Source-Receptor Matrices (Seibert and Frank, 2004) were computed at 0.25° x 0.25° resolution with FLEXPART driven by ECMWF Era Interim meteorological fields in the same way as for FlexInvert. Backward simulations were limited to 10 days prior to each observation and to the domain 15°W – 35°E, 30°N – 75°N. Fluxes were estimated for this domain on a monthly basis at 0.5° x 0.5° resolution. For the version used in this study, FLExKF-TM5-4DVAR_EMPA, the background mole fraction was taken from a global TM5-4DVAR

assimilation run (Bergamaschi et al. 2018a) where the above domain was cut out following the two-step approach of Rödenbeck et al. (2009).

**Uncertainties:** The uncertainty in the posterior fluxes is composed of random and systematic errors. The random uncertainties are represented by the posterior error covariance matrix provided by the Kalman Filter, which combines errors in the prior fluxes with errors in the observations and model representation. Systematic uncertainties primarily

arise from systematic errors in modelled atmospheric transport and in background mole fractions, but also include aggregation errors, i.e. errors arising from the way the flux variables are discretized in space and time.

#### *FLEXINVERT*

     The FlexInvert framework is based on Bayesian statistics and optimizes surface-atmosphere fluxes using the

maximum probability solution (Rodgers, 2000). Atmospheric transport is modelled using the Lagrangian model FLEXPART (Stohl et al., 2005; Pisso et al., 2019) run in the backwards time mode to generate a so-called Source-Receptor Matrix (SRM). The SRM describes the relationship between the change in mole fraction and the fluxes discretized in space and time (Seibert and Frank, 2004) and was calculated for 8 days prior to each observation. For use in the inversions, FLEXPART was driven using ECWMF operational analysis wind fields. The state vector

consisted of prior fluxes discretized on an irregular grid based on the SRMs (Thompson et al. 2014). This grid has finer resolution (in this case the finest was 0.25°×0.25°) where the fluxes have a strong influence on the observations and coarser resolution where the influence is only weak (the coarsest was 2°×2°). The fluxes were solved at 10-days temporal resolution. The state vector also included scalars for the background contribution. The background mixing ratio, i.e., the contribution to the mixing ratio that is not accounted for in the 8-day SRMs, was estimated by coupling

the termination points of backwards trajectories (modelled using virtual particles) to initial fields of methane simulated with the Lagrangian FLEXPART-CTM model, which was developed at Empa based on FLEXPART (Stohl et al., 2005; Pisso et al., 2019). In these simulations, we applied the data assimilation method described by Groot Zwaaftink et al. (2018) that constrains modelled fields with surface observations through nudging.






**Uncertainties:** The posterior fluxes are subject to systematic errors primarily from: 1) errors in the modelled atmospheric transport; 2) aggregation errors, i.e. errors arising from the way the flux variables are discretized in space and time; 3) errors in the background methane fields; and 4) the incomplete information from the observations and hence the dependence on the prior fluxes. In addition, there is, to a smaller extent, some error due to calibration offsets between observing instruments. Uncertainties in the observation space were inflated to take into account the model representation errors

*InGOS and TM5-4DVAR*

The atmospheric models used within the European FP7 project InGOS (Integrated non-CO2 Greenhouse gas Observing System) are described by Bergamaschi et al., 2018a and Supplement ([https://www.atmos-chem-phys.net/18/901/2018/acp-18-901-2018-supplement.pdf](https://www.atmos-chem-phys.net/18/901/2018/acp-18-901-2018-supplement.pdf)). The models include global Eulerian models with a zoom over Europe (TM5-4DVAR, TM5-CTE, LMDZ), regional Eulerian models (CHIMERE), and Lagrangian dispersion models (STILT,NAME,COMET). The horizontal resolutions over Europe are$\sim$1.0–1.2$\circ$ (longitude)$\times\sim$0.8–1.0$\circ$

(latitude) for the global models (zoom) and $\sim$0.17–0.56$\circ$ (longitude)$\times\sim$0.17–0.5$\circ$ (longitude) for the regional models. Most models are driven by meteorological fields from the European Centre for Medium-Range Weather Forecasts (ECMWF) ERA-Interim reanalysis (Dee et al., 2011). In the case of STILT, the operational ECMWF analyses were used, while for NAME meteorological analyses of the Met Office Unified Model (UM) were employed. The regional

models use boundary conditions (background $CH_4$ mole fractions) from inversions of the global models (STILT from TM3, COMET from TM5-4DVAR, CHIMERE from LMDZ) or estimate the boundary conditions in the inversions (NAME) using baseline observations at MaceHead as prior estimates. In the case of NAME and CHIMERE, the boundary conditions are further optimised in the inversion. The inverse modelling systems applied in this study use different inversion techniques. TM5-4DVAR, LMDZ, and TM3-STILT use 4DVAR variational techniques, which

allow optimisation of emissions of individual grid cells. These 4DVAR techniques employ an adjoint model in order to iteratively minimise the cost function using a quasi-Newton (Gilbert and Lemaréchal, 1989) or conjugate gradient (Rödenbeck, 2005) algorithm. The NAME model applies a simulated annealing technique, a probabilistic technique for approximating the global minimum of the cost function. In CHIMERE and COMET, the inversions are performed analytically after reducing the number of parameters to be optimised by aggregating individual grid cells before the

inversion. TM5-CTE applies an ensemble Kalman filter (EnKF) (Evensen, 2003), with a fixed-lag smoother (Peters et al., 2005).

**Uncertainty:** In general, the estimated model uncertainties depend on the type of station and for some models (TM5-4DVAR and NAME) also on the specific synoptic situation. In InGOS the uncertainty of the ensemble was calculated as 1σ estimate. Bergamaschi et al. (2015) showed that the range of the derived total $CH_4$ emissions from north-western

and eastern Europe using four different inverse modelling systems was considerably larger than the uncertainty estimates of the individual models because the latter typically use Bayes' theorem to calculate the reduction of assumed a prior emission uncertainties by assimilating measurements (propagating estimated observation and model errors to the estimated emissions). An ensemble of inverse models may provide more realistic overall uncertainty estimates, since estimates of model errors are often based on strongly simplified assumptions and do not represent the

total uncertainty.



### InTEM – NAME

The Inverse Technique for Emission Modelling (InTEM) (Arnold et al., 2018) uses the NAME (Numerical Atmospheric dispersion Modelling Environment) (Jones et al, 2007) atmospheric Lagrangian transport model. NAME is driven by analysis 3-D meteorology from the UK Met Office Unified Model (Cullen, 1993). The horizontal and vertical resolution of the meteorology has improved over the modelled period from 40 km to 12 km (1.5 km over the UK). InTEM is a Bayesian system that minimises the mismatch between the model and the atmospheric observations given the constraints imposed by the observation and model uncertainties and prior information with its associated uncertainties. The direction (latitude and longitude) and altitude varying background concentration and observation station bias are solved for within the inverse system along with the spatial distribution and magnitude of the emissions. The time-varying prior background concentration for the DECC network stations is derived from the MHD observations when they are very largely sensitive only to Northern Canada (Arnold et al., 2018). The prior bias (that can be positive or negative) for each station is set to zero with an uncertainty of 1 ppb. The observations from each station are assumed to have an exponentially decreasing 12-hr time correlation coefficient and, between stations, a 200 km spatial correlation coefficient. The observations are averaged into 2-hr periods. The uncertainty of the observations is derived from the variability of the observations within each 2-hr period. The modelling uncertainty for each 2-hr period at each station varies and is defined as the larger of; the median pollution events in that year at that station, or 16.5% of the magnitude of the pollution event. These values have been derived from analysis of the observations of methane at multiple heights at each station across the DECC network. Each inversion is repeated 24 times, each time 10% of the observations per year per station are randomly removed in 5-day intervals and the results and uncertainty averaged.

**Uncertainty:** This random removal of observations allows a greater exploration of the uncertainty, given the potential for some of the emission sources to be intermittent within the time-period of the inversion.

### CTE-CH4 Europe, CTE-SURF and CTE-GOSAT

CarbonTracker Europe $CH_4$ (CTE-$CH_4$) (Tsuruta et al., 2017) applies an ensemble Kalman filter (Peters et al. 2005) in combination with the Eulerian transport model TM5 (Krol et al. 2005). It optimizes surface fluxes weekly , and assimilates atmospheric $CH_4$ observations. TM5 was run at 1° x 1° resolution over Europe and 6° x 4° resolution globally, constrained by 3-hourly ECMWF ERA-Interim meteorological data. The photochemical sink of $CH_4$ due to tropospheric and stratospheric OH, and stratospheric Cl and O($^1$D) was pre-calculated based on Houweling et al. (2014) and Brühl and Crutzen (1993) and not adjusted in the optimization scheme.

Three experiments were conducted, which differ in (1) sets of prior fluxes, (2) sets of assimilated observations, and (3) optimization resolution over the Northern Hemisphere. CTE-FMI uses sets of prior fluxes from LPX-Bern DYPTOP (Stocker et al., 2014) for biospheric, EDGAR v4.2 FT2010 (Janssens-Maenhout et al., 2013) for anthropogenic, GFED v4 (Giglio et al., 2013) for biomass-burning, Ito and Inatomi (2012) for termites and Tsuruta et al. (2017) for ocean sources. CTE-SURF and CTE-GOSAT use sets of prior fluxes from Global Carbon Project (Saunois et al., 2020). CTE-FMI and CTE-SURF assimilated ground-based surface $CH_4$ observations, while CTE-GOSAT assimilated GOSAT XCH$_4$ retrievals from NIES v2.72. CTE-FMI optimized fluxes at 1° x 1° resolution over Northern Europe, northeast Russia and southeast Canada, 6° x 4° resolution over other parts of the Northern



Hemisphere land, and region-wise (combined TransCom regions and soil-type) over the Southern Hemisphere and ocean. CTE-SURF and CTE-GOSAT fluxes were optimized at 1° x 1° resolution over Europe and region-wise

elsewhere globally.

**Uncertainty:** The prior uncertainty is assumed to be a Gaussian probability distribution function, where the error covariance matrix includes errors in prior fluxes, observations and transport model representations. The uncertainty for the prior fluxes were assumed to be 80 % of the fluxes over land and 20 % over ocean, with correlation between grid cells or regions to be 100-500 km over land and 900 km over ocean. The uncertainty for observations and transport

model representations vary between observations, with min. aggregated uncertainty to be 7.5 ppb for surface observations and 15 ppb for GOSAT data. The posterior uncertainty is calculated as standard deviation of the ensemble members, where the posterior error covariance matrix is driven by the ensemble Kalman filter.

### *MIROC4-ACTM:*

The MIROC4-ACTM time dependent inversions solve for emissions from 53 regions for $CH_4$ and N2O. The inversion framework is based on Bayesian statistics and optimizes surface-atmosphere fluxes using the maximum probability solution. Atmospheric transport is modelled using the JAMSTEC's Model for Interdisciplinary Research on climate, version 4 based atmospheric chemistry-transport model (MIROC4-ACTM) (Watanabe et al. 2008; Patra et al. 2018). The Source-Receptor Matrix (SRM) is calculated by simulating unitary emissions from 53 or 84 basis

regions, for which the fluxes are optimised. The SRM describes the relationship between the change in mole fraction at the measurement locations for the unitary basis region fluxes. The MIROC4-ACTM meteorology was nudged to the JMA 55-year reanalysis (JRA55) horizontal wind fields and temperature. The calculation of photo-chemical losses is performed online. The hydroxyl (OH) radical concentration for reaction with $CH_4$ vary monthly but without any interannual variations. The simulated mole fractions for the total a priori fluxes are subtracted from the observed

concentrations before running the inversion calculation (as in Patra et al., 2016 for $CH_4$ inversion). Both the inversion results are contributed to the GCP-CH4 and GCP-N2O activities (Saunois et al., 2020; Thompson et al., 2019; Tian et al., 2020).

**Uncertainties:** The posterior fluxes are subject to systematic errors primarily from: 1) errors in the modelled atmospheric transport; 2) aggregation errors, i.e. errors arising from the way the flux variables are discretized in space

(84 regions) and time (monthly-means); 3) errors in the background mole fractions (assumed to be a minor factor); and 4) the incomplete information from the sparse observational network and hence the dependence on the prior fluxes. In addition, there is, to a much smaller extent, some error due to calibration offsets between observing instruments, which is more pertinent for $N_2O$ than for other GHGs. We have validated model transport in troposphere using $SF_6$ for the inter-hemispheric exchange time, and the using $SF_6$ and $CO_2$ for the age of air in the stratosphere. The simulated

$N_2O$ concentrations are also compared with aircraft measurements in the upper troposphere and lower stratosphere for evaluating the stratosphere-troposphere exchange rates. Comparisons with ACE-FTS vertical profiles in the stratosphere and mesosphere indicate good parameterisation of $N_2O$ loss by photolysis and chemical reactions, and thus the lifetime, which affect the global total $N_2O$ budgets. Random uncertainties are calculated by the inverse model depending on the prior flux uncertainties and the observational data density and data uncertainty. Only 37 sites are





used in the inversion and thus the reduction in priori flux uncertainties have been minimal. The net fluxes from the inversion from individual basis regions are less reliable compared to the anomalies in the estimated fluxes over a period of time.

**Global Carbon Project – Global Methane Budget (GMB)**

GMB uses an ensemble of 22 top-down global inversions for anthropogenic $CH_4$ emissions presented in Saunois et al. (2020) for the Global Methane Budget. These inversions were simulated by nine atmospheric inversion systems based on various chemistry transport models, differing in vertical and horizontal resolutions, meteorological forcing, advection and convection schemes, and boundary layer mixing. Surface-based inversions were performed over the period 2000-2017 while satellite-based inversions cover the GOSAT data availability 2010-2017. The

protocol established for these simulations was not stringent as the prior emission flux data set was not mandatory, and each group selected their constraining observations. More information can be found in Saunois et al. (2020) in particular in their Table 6 and S6.

**Uncertainties**: currently there are no uncertainties reported for the GMB models. This study uses the median and the min/mas as uncertainty range estimation from the 22 models ensemble. In general uncertainties might be due to factors

like: different transport models, physical parametrizations, prior fluxes, observation data sets etc.

## A2: Natural $CH_4$ emissions
**Bottom-up $CH_4$ emissions estimates**

**$CH_4$ emissions from inland waters**

    The $CH_4$ estimate from inland waters represents a climatology of average annual diffusive and ebulitive $CH_4$ emissions from lakes and reservoirs at the spatial resolution of 0.1°. The climatology is based on five alternative estimates, all relying on the high-resolution HydroLAKES database (Messager et al., 2016), and of which we report the mean and the standard deviation as a measure of uncertainty. Four of these estimates are based on predictions of

$CH_4$ emission rates from N and P concentrations. These concentrations were computed for each lake and reservoir of the HydroLAKES dataset (> 1.4 millions), using the mechanistic-stochastic model (MSM) of Maavara et al. (2017, 2019) and Lauerwald et al. (2019); see methodology for inland water $N_2O$ emissions for further details. The four estimates result from two empirical equations relating $CH_4$ emissions to chlorophyll-a concentrations (Deemer et al., 2016; DelSontro et al., 2018) and two equations relating chlorophyll-a concentrations to nutrient concentrations (both

from McCauley et al., 1989) in lakes and reservoirs. The fifth estimate is based on direct upscaling from observed $CH_4$ emission rates (155 lakes and reservoirs), which we have classified into rates reported for small lakes (<0.3 $km^2$), larger (>0.3 $km^2$) lakes, and reservoirs. In addition, we applied a coarse regionalization distinguishing the Boreal (>54°N) from the Temperate to Sub-Tropical (<54°N) zone.

**JSBACH-HIMMELI**



The model framework, JSBACH-HIMMELI (Raivonen et al., 2017; Susiluoto et al., 2018) is used to estimate wetland and mineral soil emissions, and an empirical model is used to estimate the emissions from inland water bodies.

JSBACH-HIMMELI is a combination of two models, JSBACH, that is the land-surface model of MPI-ESM (Reick et al., 2013), and HIMMELI, that is a specific model for northern peatland emissions of $CH_4$ (Raivonen et al., 2017). HIMMELI (HelsinkI Model of MEthane buiLd-up and emIssion for peatlands) has been developed especially for estimating $CH_4$ production and transport in northern peatlands. It simulates both $CH_4$ and $CO_2$ fluxes and can be used as a module within different modelling environments (Raivonen et al., 2017; Susiluoto et al., 2018). HIMMELI is driven with soil temperature, water table depth, the leaf area index and anoxic respiration. These parameters are provided to HIMMELI from JSBACH, which models hydrology, vegetation and soil carbon input from litter and root exudates. $CH_4$ emission and uptake of mineral soils are calculated applying the method by Spahni et al. (2011) based on soil moisture estimated by JSBACH.

The distribution of terrestrial vegetation types in JSBACH-HIMMELI is adopted from CORINE land cover data and from native JSBACH land cover for the areas that CORINE does not cover. The HIMMELI methane model is applied for peatlands and the mineral soil approach for the rest. The map of inland water $CH_4$ emissions has been combined with JSBACH-HIMMELI land use map so that the map of inland waters is preserved and JSBACH grid-based fractions of different land use categories adjusted accordingly. In order to avoid double-counting the terrestrial $CH_4$ flux estimates have been normalized by the ratio of the two inland water body distributions.

**Uncertainties:** As in any process modeling the uncertainties of the bottom up modeling of $CH_4$ arise from three primary sources: parameters, forcing data (including spatial and temporal resolution), and model structure. An important source of uncertainty in the case of terrestrial $CH_4$ flux modeling is the spatial distribution of peatlands. The uncertainties of JSBACH-HIMMELI peatland emissions were estimated by comparing the annual totals of measured and simulated methane fluxes at five European observation sites. Two of the sites are located in Finnish Lapland, one in middle Sweden, one in southern Finland and one in Poland.

For the sensitivity of mineral soil fluxes Spahni et al. (2011) tested two soil moisture thresholds, 85% or 95% of water holding capacity, below which mineral soils were assumed to be only $CH_4$ sinks, above which sources. We used the higher value, 95% of water holding capacity. The uncertainty was estimated using $CH_4$ flux simulations of one year (2005). We did two new model runs, using moisture thresholds 95±15%, and derived the uncertainty from the resulting range in the annual emission sum.

***Geological fluxes***

To calculate geological $CH_4$ emissions we used literature data for geological emissions on land (excluding marine seepage) (Etiope et al., 2019; Hmiel et al., 2020). Geological emissions were calculated by scaling the regional emissions from Etiope et al. 2019 (37.4 Tg $CH_4$ yr$^{-1}$) to the global ratio of emissions from Hmiel et al. (2020), obtaining an estimate of 1.3 Tg $CH_4$ yr$^{-1}$ (marine and land geological). Marine seepage emissions were excluded.

***Top-down $CH_4$ emissions estimates***



***Global Carbon Project - Global Methane Budget (Saunois et al., 2020)***

GMB uses an ensemble of thirteen monthly gridded estimates of wetland emissions based on different land surface models as calculated for Saunois et al. (2020). Each model conducted a 30-year spin-up and then simulated net methane emissions from wetland ecosystems over 2000-2017. The models were forced by CRU-JRA reconstructed climate fields (Harris, 2019), and by the remote sensing-based wetland dynamical area dataset WAD2M (Wetland Area Dynamics for Methane Modeling). This data set provides monthly global areas over 2000-2017 based on a combination of microwave remote sensing data from Schroeder et al. (2015) and various regional inventory data sets. More information is available in Saunois et al. (2020) and more details will be presented in a future publication led by Poulter et al., 2017 and colleagues.

**Uncertainty:** As described by Saunois et al. (2020) uncertainties are reported as minimum and maximum values of the available studies, in brackets. They do not take into account the uncertainty of the individual estimates, but rather express the uncertainty as the range of available mean estimates, i.e., the standard error across measurements/methodologies considered.

## A3: Anthropogenic and natural N$_2$O emissions

***Bottom-up N$_2$O emission estimates***

***UNFCCC NGHGI (2019), EDGAR v5.0 and CAPRI:*** descriptions are found in Appendix A1.

***ECOSSE***

ECOSSE is a biogeochemical model that is based on the carbon model ROTH-C (Jenkinson and Rayner, 1977; Jenkinson et al. 1987; Coleman and Jenkinson, 1996) and the nitrogen-model SUNDIAL (Bradbury et al. 1993; Smith et al. 1996). All processes of the carbon and nitrogen dynamics are considered (Smith et al., 2010a,b). Additionally, in ECOSSE processes of minor relevance for mineral arable soils are implemented as well (e.g. methane emissions) to have a better representation of processes that are relevant for other soils (e.g. organic soils). ECOSSE can run in different modes and for different time steps. The two main modes are site specific and limited data. In the later version, basis assumptions/estimates for parameters can be provided by the model. This increases the uncertainty but makes ECOSSE a universal tool that can be applied for large scale simulations even if the data availability is limited. To increase the accuracy in the site-specific version of the model, detailed information about soil properties, plant input, nutrient application and management can be added as available.

During the decomposition process, material is exchanged between the SOM pools according to first order rate equations, characterised by a specific rate constant for each pool, and modified according to rate modifiers dependent on the temperature, moisture, crop cover and pH of the soil. The N content of the soil follows the decomposition of the SOM, with a stable C:N ratio defined for each pool at a given pH, and N being either mineralised or immobilised to maintain that ratio. Nitrogen released from decomposing SOM as ammonium (NH4+) or added to the soil may be nitrified to nitrate (NO3-).



For spatial simulations the model is implemented in a spatial model platform. This allows us to aggregate the input parameter for the needed resolution. ECOSSE is a one-dimensional model and the model platform provides the input data in a spatial distribution and aggregates the model outputs for further analysis. While climate data are interpolated, soil data are represented by the dominant soil type or by the proportional representation of the different soil types in the spatial simulation unit (this is in VERIFY a grid cell).

**Uncertainties** in ECOSSE arise from three primary sources: parameters, forcing data (including spatial and temporal resolution), and model structure.

*DayCent*

DayCent was designed to simulate soil C dynamics, nutrient flows (N, P, S) and trace gas fluxes ($CO_2$, $CH_4$, $N_2O$, $NO_x$, $N_2$) between soil, plants and the atmosphere at daily time-step. Submodels include soil water content and temperature by layer, plant production and allocation of net primary production (NPP), decomposition of litter and soil organic matter, mineralization of nutrients, N gas emissions from nitrification and denitrification, and $CH_4$ oxidation in non-saturated soils.

The DayCent modelling application at the EU level is a consolidated model framework running on LUCAS point (Orgiazzi, 2018) which was extensively explained in previous works (Lugato et al., 2017, 2018; Quemada et al., 2020) where a detailed description of numerical and geographical datasets and uncertainty estimations is reported.

Information directly derived from LUCAS (2009-2015) included the soil organic carbon content (SOC), particle size distribution and pH. Hydraulic properties and bulk density was also calculated with an empirically-derived pedotransfer. Management information was derived from official statistics (Eurostat, 2019) and included crop shares at NUTS2 level. The amount of mineral N was partitioned according to the regional crop rotations and agronomic crop requirements. Organic fertilization and irrigated areas were derived from the 'Gridded Livestock of the World' FAO dataset and the FAO-AQUASTAT product.

Meteorological data were downloaded from the E-OBS gridded dataset ([http://www.ecad.eu](http://www.ecad.eu)) at 0.1° resolution. For the climatic projection, the gridded data from CORDEX database ([https://esgf-node.ipsl.upmc.fr/search/cordex-ipsl/](https://esgf-node.ipsl.upmc.fr/search/cordex-ipsl/)) were used. The average annual (2006-2010) atmospheric N deposition from the EMEP model (rv 4.5) were also implemented into the simulations.

**Uncertainty:** The starting year of the simulation was set in 2009 and projected in the future. The uncertainty analysis, based on the Montecarlo approach, was done running the model 52 times in each point and, contemporary, randomly sampling model inputs from probability density functions for: SOC pool partition, irrigation and both mineral and organic fertilization rates. The model outputs (including uncertainties) at point level were up-scaled regionally at 1 km resolution by a machine learning approach based on Random Forest regression.

*$N_2O$ emissions from inland waters*

The $N_2O$ estimate represents a climatology of average annual $N_2O$ emissions from rivers, lakes, reservoirs and estuaries at the spatial resolution of 0.1°. Based on a spatially explicit representation of water bodies and point and non-point sources of N and P, this model quantifies the global scale spatial patterns in inland water $N_2O$ emissions





in a consistent manner at 0.5° resolution, which were then downscaled to 0.1° using the spatial distribution of European inland water bodies. The procedure to calculate the cascading loads of N and P delivered to each water body along the river–reservoir–estuary continuum and to topologically connect 1.4 million lakes (extracted from the HYDROLAKES database) is described in Maavara et al., 2019 and Lauerwald et al., 2019. The methodology to quantify $N_2O$ emissions is based on the application of a mechanistic stochastic model (MSM) to estimate inland water C-N-P cycling as well

as $N_2O$ production and emission generated by nitrification and denitrification. Using a Monte Carlo analysis, the MSM allows to generate relationships relating N processes and $N_2O$ emissions to N and P loads and water residence time from the mechanistic model outputs, which are subsequently applied for the spatially resolved upscaling. For the estimation of $N_2O$ emission, we ran two distinct model configurations relying on EFs scaling to denitrification and nitrification rates: one assuming that $N_2O$ production equals $N_2O$ emissions, the other taking into account the kinetic

limitation on $N_2O$ gas transfer and progressive $N_2O$ reduction to $N_2$ during denitrification in water bodies with increasing residence time (Maavara et al., 2019). The model outputs from the two scenarios are used to constrain uncertainties in $N_2O$ emission estimates.

### *GAINS*

Specific sectors and abatement technologies in GAINS vary by the specific emitted compound, with source sector definition and emission factors largely following the IPCC methodology at the Tier 1 or Tier 2 level. GAINS includes in general all anthropogenic emissions to air, but does not cover emissions from forest fires, savannah burning and land use / land use change. Emissions are estimated for 174 countries/regions, with the possibility to aggregate to a global emission estimate, and spanning a timeframe from 1990 to 2050 in five-year intervals. Activity drivers for

macroeconomic development, energy supply and demand, and agricultural activities are entered externally, GAINS extends with knowledge required to estimate "default" emissions (emissions occurring due to an economic activity without emission abatement) and emissions and costs of situations under emission control (Amann et al., 2001).

Emissions of nitrous oxide derive from energy, industry, agriculture, and waste. Land use change emissions are not included. In the energy sector, certain technologies implemented to improve air quality affect N2O emission

factors (like catalytic converters in vehicles), sometimes also negatively. That is also the case for non-selective catalytic reduction devices for NOx abatement in power plants, or for fluidized bed combustion. Relevant industrial processes cover nitric acid and adipic acid, with other processes (glyoxal, if relevant, or caprolactam) included. Both processes allow for two different levels of abatement technologies, which both are relatively easily accessible and low cost. The use of $N_2O$ in gaseous form, often as an anesthetic for medical purposes, is associated with population

numbers and scaled by availability of hospital beds. Marked emission reductions (at low costs) as well as complete phase out of emissions (high costs) are implemented as technologies. Agricultural emissions in part derive from manure handling, where different management strategies have repercussions on emissions. The larger fraction of emission is from application of nitrogen compounds in different forms to grassland, crops and rice, with rice using a different emission factor. Application of manure and of mineral fertilizer in GAINS can be reduced by advanced

computer technology such as automatic steering and variable rate application, or by agrochemistry (nitrification inhibitors). Costs of implementation are considered to depend on the size of a farm, hence farm size is an important





parameter. In the waste sector, composting and wastewater treatment are considered relevant sources. For wastewater treatment, GAINS also considers a specific emission reduction option when optimizing processes towards N2O reduction (e.g. via favoring the anammox process). All details have been reported by Winiwarter et al. (2018) in their

supplementary material.

**Uncertainties**: The same paper provides full information on the uncertainty of $N_2O$ emissions in the GAINS model, which is a consequence of uncertainty provided in the activity data, in the emission factors, and in the actual structure of the respective management strategies that also include the share of abatement technology already implemented. Further parameters also described (on uncertainty of future projections and on costs) are not relevant here.


### *FAOSTAT*

FAOSTAT: Statistics Division of the Food and Agricultural Organization of the United Nations, provides $N_2O$ emissions from agriculture: http://www.fao.org/faostat/en/#data/GT/visualize and its sub-domains, as well as $N_2O$ emissions from land use linked to biomass burning: http://www.fao.org/faostat/en/#data/GI (metadata:

http://fenixservices.fao.org/faostat/static/documents/GT/GT_e_2019.pdf                                and http://fenixservices.fao.org/faostat/static/documents/GL/GL_e_2019.pdf). The FAOSTAT emissions database is computed following Tier 1 IPCC 2006 Guidelines for National GHG Inventories (http://www.ipcc-nggip.iges.or.jp/public/2006gl/index.html). Country reports to FAO on crops, livestock and agriculture use of fertilizers are the source of activity data. Geospatial data are the source of AD for the estimates from cultivation of

organic soils, biomass and peat fires. $N_2O$ emissions are provided by country, regions and special groups, with global coverage, relative to the period 1961-present (with annual updates, currently 2017) and with projections for 2030 and 2050 for agriculture only, expressed in both $CO_2e$ and $N_2O$ by underlying agricultural and land use emission sub-domain and by aggregate (agriculture total, agriculture total plus energy, agricultural soils). The main $N_2O$ emissions are reported for the following agricultural activities: manure management, synthetic fertilizers, manure applied to the

soils, manure left in pasture, crop residues, cultivation of organic soils and burning crop residues. LULUCF emissions consist of $N_2O$ associated with burning biomass and peat fires, as well as from the drainage of organic soils.

**Uncertainties** were computed by Tubiello et al., 2013 but are not available in the FAOSTAT database.

### *Top-down N₂O emission estimates*
### *FLEXINVERT*

The FlexInvert framework is based on Bayesian statistics and optimizes surface-atmosphere fluxes using the maximum probability solution (Rodgers 2000). Atmospheric transport is modelled using the Lagrangian model FLEXPART (Stohl et al. 2005; Pisso et al. 2019) run in the backwards time mode to generate a so-called Source-Receptor Matrix (SRM). The SRM describes the relationship between the change in mole fraction and the fluxes discretized in space and time (Seibert and Frank, 2004) and was calculated for 7 days prior to each observation. For

use in the inversions, FLEXPART was driven using ECMWF Era Interim wind fields.

The state vector consisted of flux increments (i.e. offsets to the prior fluxes) discretized on an irregular grid based on the SRMs (Thompson et al. 2014). This grid has finer resolution (in this case the finest was 0.5°×0.5°) where the fluxes have a strong influence on the observations and coarser resolution where the influence is only weak (the





coarsest was 2°×2°). The flux increments were solved at 2-weekly temporal resolution. The state vector also included
scalars for the background mole fractions. The optimal (posterior) fluxes were found using the Conjugate Gradient
method (e.g. Paige and Saunders, 1975).

The background mole fractions, i.e., the contribution to the modelled mole fractions that is not accounted for
in the 7-day SRMs, was estimated by coupling the termination points of backwards trajectories (modelled using virtual
particles) to initial fields of mole fractions from the optimized Eulerian model LMDz (i.e. the CAMS $N_2O$ mole
fraction product v18r1) following the method of Thompson et al. 2014.

**Uncertainties:** The posterior fluxes are subject to systematic errors primarily from: 1) errors in the modelled
atmospheric transport; 2) aggregation errors, i.e. errors arising from the way the flux variables are discretized in space
and time; 3) errors in the background mole fractions; and 4) the incomplete information from the observations and
hence the dependence on the prior fluxes. In addition, there is, to a smaller extent, some error due to calibration offsets
between observing instruments, which is more pertinent for $N_2O$ than for other GHGs. Random uncertainties are
calculated from a Monte Carlo ensemble of inversions following Chevallier et al. (2007) and uncertainties in the
observation space were inflated to take into account the model representation errors.

### *Global N₂O Budget – GCP (Tian et al., 2020)*

#### *PyVAR*

Within the GCP 2019 results, $N_2O$ fluxes are estimated using the atmospheric inversion framework, PyVAR-
$N_2O$. Atmospheric inversions use observations of atmospheric mixing ratios, in this case, of $N_2O$, and provide the
fluxes that best explain the observations while at the same time being guided by a prior estimate of the fluxes. In other
words, the fluxes are optimized to fit the observations within the limits of the prior and observation uncertainties. To
produce the optimized (*a posteriori*) fluxes a number of steps are involved: first, the observations are pre-processed,
second, a prior flux estimate is prepared, third mixing ratios are simulated using the prior fluxes and are used to
estimate the model representation error, and fourth, the inversion is performed.\ In total 140 ground-based sites, ship
and aircraft transects are included in the inversion. The term "site" refers to locations where there is a long-term record
of observations and includes ground-based measurements, both from discrete samples (or "flasks") and quasi-
continuous sampling by in-situ instruments, as well as aircraft measurements. A prior estimate of the total $N_2O$ flux
with monthly resolution and inter-annually varying fluxes is prepared from a number of models and inventories. For
the soil fluxes (including anthropogenic and natural) an estimate from the land surface model OCN-v1.2 is used, which
is driven by observation-based climate data, N-fertilizer statistics and modelled N-deposition (Zaehle et al. 2011). For
the ocean fluxes, an estimate from the ocean biogeochemistry model PlankTOM-v10.2 is used, which is a prognostic
model (Buitenhuis et al. 2018). Atmospheric transport is modelled using an offline version of the Laboratoire de
Meteorologie Dynamique model, LMDz5, which computes the evolution of atmospheric compounds using archived
fields of winds, convection mass fluxes and planetary boundary layer (PBL) exchange coefficients that have been
calculated using the online version nudged to ECMWF ERA interim winds.

PyVAR-$N_2O$ uses the Bayesian inversion method to find the optimal fluxes of $N_2O$ given prior information about the
fluxes and their uncertainty, and observations of atmospheric $N_2O$ mole fractions. The method is the same as that used
in Thompson et al. (2014)



**Uncertainty:** Uncertainties in PyVAR simulations pertain to observation space and to state space. Uncertainty in the observation space is calculated as the quadratic sum of the measurement and transport uncertainties. The measurement uncertainty is assumed to be 0.3 ppb (approximately 0.1%) based on the recommendations of data providers. The transport uncertainty includes estimates of uncertainties in advective transport (based on the method of Rödenbeck et al. (2003)) and from a lack of subgrid-scale variability (based on the method of Bergamaschi et al. (2010)). For the error in each land grid cell, the maximum magnitude of the flux in the cell of interest and its 8 neighbours is used, while for ocean grid cells the magnitude of the cell of interest only is used. Posterior flux uncertainties are calculated from a Monte Carlo ensemble of inversions, based on the method of Chevallier et al. (2005).

### TOMCAT-INVICAT

TOMCAT-INVICAT (Wilson et al., 2014) is a variational inverse transport model, which is based on the global chemical transport model TOMCAT, and its adjoint. It uses a 4-D variational (4D-VAR) optimization framework based on Bayesian theory which seeks to minimize model-observation differences by altering surface fluxes, while allowing for prior knowledge of these fluxes to be retained. TOMCAT (Monks et al., 2017) is an offline chemical transport model, in which meteorological data is taken from ECMWF ERA-Interim reanalyses (Dee et al., (2011)). The model grid resolution, and therefore the optimised surface flux estimates, have a horizontal resolution of 5.6 x 5.6 degrees. The model has 60 vertical levels running from the surface to 0.1 hPa. For each individual year's fluxes, which are optimised on a monthly basis, 30 minimisation iterations are carried out.

**Uncertainty:** Uncertainties in TOMCAT-INVICAT $N_2O$ inversions are described as follows and further in Thompson et al., (2019). Uncertainty in the observations is calculated as the quadratic sum of the measurement and transport uncertainties. The measurement uncertainty for each observation is assumed to be 0.4 ppb. For the transport error for each observation is assumed to be the mean difference between the observation grid cell and its 8 neighbours. Prior flux errors are assumed to be 100% or the prior estimate, and are uncorrelated in space and time. Posterior flux uncertainties are not currently able to be calculated.

### MIROC4-ACTM

The MIROC4-ACTM time dependent inversion for 84 regions (TDI84) framework is based on Bayesian statistics and optimizes surface-atmosphere fluxes using the maximum probability solution (Rodgers 2000). Atmospheric transport is modelled using the JAMSTEC's Model for Interdisciplinary Research on climate, version 4 based atmospheric chemistry-transport model (MIROC4-ACTM) (Watanabe et al. 2008; Patra et al. 2018). The Source-Receptor Matrix (SRM) is calculated by simulating unitary emissions from 84 basis regions, for which the fluxes are optimised. The SRM describes the relationship between the change in mole fraction at the measurement locations for the unitary basis region fluxes (similar to Rayner et al., 1997). The MIROC4-ACTM meteorology was nudged to the JMA 55-year reanalysis (JRA55) horizontal wind fields and temperature.

The simulated mole fractions for the total a priori fluxes are subtracted from the observed concentrations before running the inversion calculation (as in Patra et al., 2016 for $CH_4$ inversion).

**Uncertainties:** The posterior fluxes are subject to systematic errors primarily from: 1) errors in the modelled atmospheric transport; 2) aggregation errors, i.e. errors arising from the way the flux variables are discretized in space (84 regions) and time (monthly-means); 3) errors in the background mole fractions (assumed to be a minor factor);



and 4) the incomplete information from the sparse observational network and hence the dependence on the prior fluxes. In addition, there is, to a much smaller extent, some error due to calibration offsets between observing instruments, which is more pertinent for $N_2O$ than for other GHGs. We have validated model transport in the troposphere using $SF_6$ for the inter-hemispheric exchange time, and the using $SF_6$ and $CO_2$ for the age of air in the stratosphere. The simulated $N_2O$ concentrations are also compared with aircraft measurements in the upper troposphere and lower

stratosphere for evaluating the stratosphere-troposphere exchange rates. Comparisons with ACE-FTS vertical profiles in the stratosphere and mesosphere indicate good parameterisation of $N_2O$ loss by photolysis and chemical reactions, and thus the lifetime, which affect the global total $N_2O$ budgets.

        Random uncertainties are calculated by the inverse model depending on the prior flux uncertainties and the observational data density and data uncertainty. Only 37 sites are used in the inversion and thus the reduction in priori

flux uncertainties have been minimal. The net fluxes from the inversion from individual basis regions are less reliable compared to the anomalies in the estimated fluxes over a period of time.

## Appendix B
### B1: Overview figures

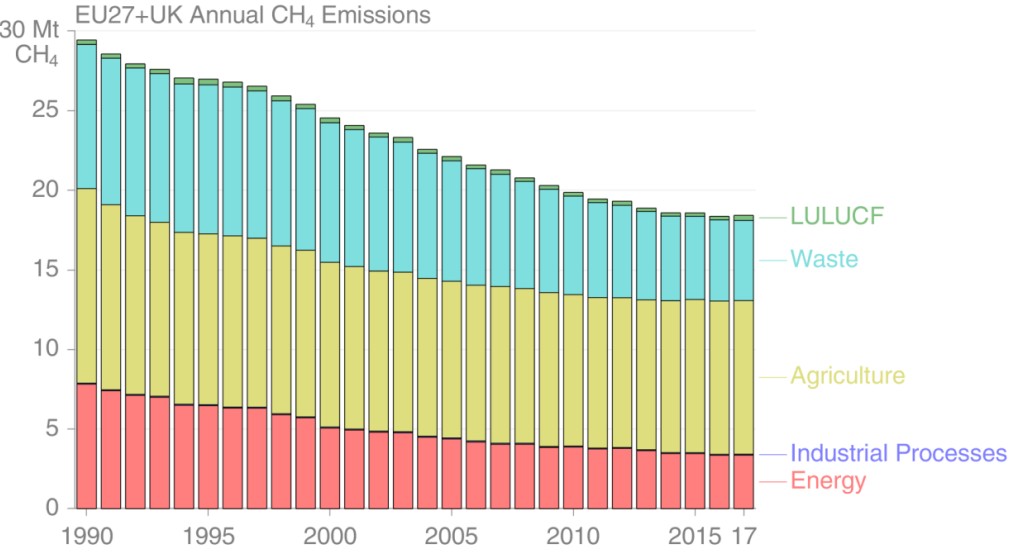

*Figure B1a: EU27+UK total CH₄ emissions time series per sectors as reported by UNFCCC NGHGI (2019).*

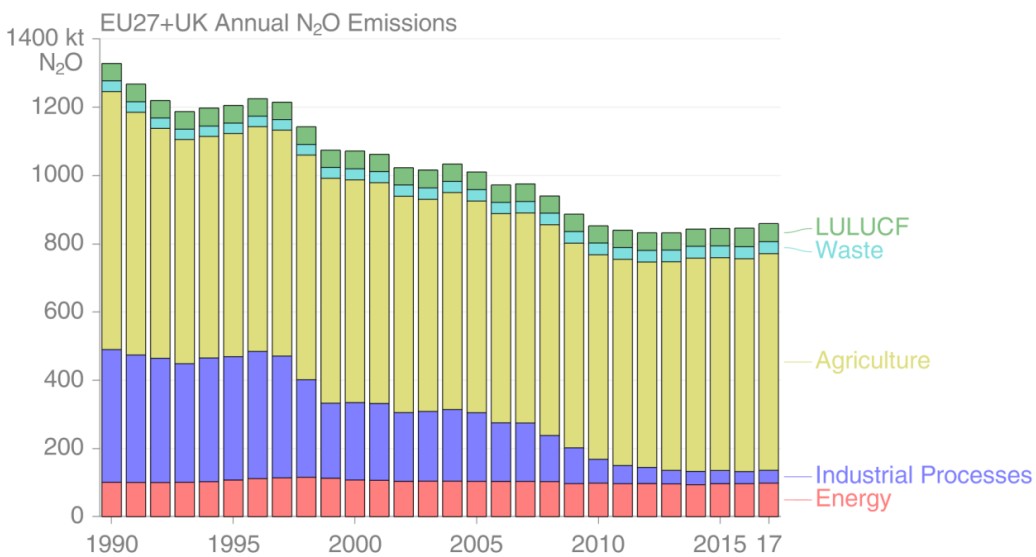

*Figure B1b: EU27+UK total N$_2$O emissions time series per sectors as reported by UNFCCC NGHGI (2019).*

### B2: Source specific methodology: AD, EF and uncertainties

*Table B1: Source specific activity data (AD), emission factors (EF) and uncertainty methodology for all current VERIFY and non-VERIFY 2019 data product collection.*

| CH$_4$ bottom-up anthropogenic emissions | | | | |
|---|---|---|---|---|
| **Data source** | **AD/Tier** | **EFs/Tier** | **Uncertainty assessment method** | **Emission data availability** |
| **UNFCCC NGHGI (2019)** | Country-specific information consistent with the IPCC GLs. | IPCC GLs/country-specific information for higher tiers. | IPCC GLs (https://www.ipcc-nggip.iges.or.jp/public/ 2006gl/, last access: December 2019) for calculating the uncertainty of emissions based on the uncertainty of AD and EF, two different approaches: (1) error propagation and (2) Monte Carlo simulation. | NGHGI official data (CRFs) are found at https://unfccc.int/process-and-meetings/ transparency-and-reporting/ reporting-and-review-under-the-convention/ greenhouse-gas-inventories-annex-i-parties/ submissions/ national-inventory-submissions-2019 (last access: September 2020). |
| **UNFCCC MS-NRT** | Country-specific information consistent with the IPCC GLs. | IPCC GLs/country-specific information for higher tiers. | n/a | EEA Report, Approximated EU GHG inventory: proxy GHG estimates for 2018, https://www.eea.europa.eu/publications/approximated-eu-ghg-inventory-proxy |



| EDGAR v5.0 | International Energy Agency (IEA) for fuel combustion Food and Agricultural Organisation (FAO) for agriculture US Geological Survey (USGS) for industrial processes (e.g. cement, lime, ammonia and ferroalloys) GGFR/NOAA for gas flaring World Steel Association for iron and steel production International Fertilisers Association (IFA) for urea consumption and production Complete description of the data sources can be found in Janssens-Maenhout et al. and in Crippa et al. (2019b). | IPCC 2006, Tier 1 or Tier 2 depending on the sector | Tier 1 with error propagation by sectors for $CH_4$ | https://edgar.jrc.ec.europa.eu/overview.php?v=50_GHG |
|---|---|---|---|---|
| CAPRI | Farm and market balances, economic parameters, crop areas, livestock population and yields from EUROSTAT, parameters for input-demand functions at regional level from FADN (EC), data on trade between world regions from FAOSTAT, policy variables from OECD. | IPCC 2006: Tier 2 for emissions from enteric fermentation of cattle and from manure management of cattle. Tier 1 for all other livestock types and emission categories. N-flows through agricultural systems (including N excretion) calculated endogenously. | N/A | Detailed gridded data $CH_4$ and $N_2O$ emissions can be obtained by contacting the data provider: Adrian.Leip@ec.europa.eu |



| Data source | AD/Tier | EFs/Tier | Uncertainty assessment method | Emission data availability |
|---|---|---|---|---|
| **GAINS** | Livestock numbers by animal type (FAOSTAT, 2010; EUROSTAT, 2009; UNFCCC, 2010) Growth in livestock numbers from FAOSTAT (2003), CAPRI model (2009) Rice cultivation Land area for rice cultivation (FAOSTAT, 2010) Projections for EU are taken from the CAPRI Model | Country-specific information and: Livestock - Implied EFs reported to UNFCCC and IPCC Tier 1 (2006, Vol.4, Ch. 10) default factors Rice cultivation - IPCC Tier 1–2 (2006, Vol. 4, p. 5.49 Agricultural waste burning - IPCC Tier 1 (2006, Vol. 5, p. 520 | IPCC (2006, Vol.4, p.10.33) uncertainty range | Detailed gridded data $CH_4$ and $N_2O$ emissions can be obtained by contacting the data providers: for $CH_4$, contact Lena Höglund Isaksson (hoglund@iiasa.ac.at); for $N_2O$, contact Wilfried Winiwarter (winiwart@iiasa.ac.at). |
| **FAOSTAT** | FAOSTAT Crop and Livestock Production domains from country reporting; FAOSTAT Land Use Domain; Harmonized world soil; ESA CCI and Copernicus Global Land Cover Service (C3S) maps; MODIS MCD12Q1 v6; FAO Gridded Livestock of the World; MODIS MCD64A1.006 burned area products | IPCC guidelines Tier 1 | IPCC (2006, Vol.4, p.10.33) Uncertainties in estimates of GHG emissions are due to uncertainties in emission factors and activity data. They may be related to, inter alia, natural variability, partitioning fractions, lack of spatial or temporal coverage, or spatial aggregation. | Agriculture total and subdomain specific GHG emissions are found for download at http://www.fao.org/faostat/en/#data/GT (last access: June 2020). |
| **CH₄ bottom-up natural emissions** | | | | |
| **Data source** | **AD/Tier** | **EFs/Tier** | **Uncertainty assessment method** | **Emission data availability** |
| | | | | |



| Mechanistic Stochastic Model CH$_4$ emissions from inland waters | Hydrosheds 15s (Lehner et al., 2008) and Hydro1K (USGS, 2000) for river network, HYDROLAKES for lakes and reservoirs network and surface area (Messager et al., 2016); Worldwide Typology of estuaries by Dürr et al. (2011) | N/A | Four model configurations for CH$_4$ | Detailed gridded data can be obtained by contacting the data providers: Ronny Lauerwald Ronny.Lauerwald@ulb.ac.be Pierre Regnier Pierre.Regnier@ulb.ac.be |
|---|---|---|---|---|
| **JSBACH-HIMMELI** | JSBACH vegetation and soil carbon and physical parameters provided to HIMMELI to simulate wetland methane fluxes HydroLAKES database (Messager et al., 2016). CORINE land cover data VERIFY climate drivers 0.1◦ × 0.1 ◦ | CH$_4$ fluxes from peatlands | the standard deviation and the resulting range in the annual emission sum represents a measure of uncertainty. | Detailed gridded data CH$_4$ emissions can be obtained by contacting the data providers: Tuula.Aalto@fmi.fi tiina.markkanen@fmi.fi |
| Geological emissions, including marine and land geological) | Areal distribution activity: 1◦ × 1 ◦ maps include the four main categories of natural geo-CH$_4$ emission: (a) onshore hydrocarbon macro-seeps, including mud volcanoes, (b) submarine (offshore) seeps, (c) diffuse microseepage and (d) geothermal manifestations. | CH$_4$ fluxes, measurements and estimates based on size and activity | 95% confidence interval of the median emission-weighted mean sum of individual regional values | (Etiope et al, 2019 and Hmiel et al., 2020) |
| **CH$_4$ Top-down inversions** | | | | |
| **Regional inversions over Europe ( high transport model resolution )** | | | | |



| Data source | AD/Tier | EFs/Tier | Uncertainty assessment method | Emission data availability |
|---|---|---|---|---|
| FLEXPART - FLExKF | Extended Kalman Filter in combination with backward Lagrangian transport simulations using the model FLEXPART Atmospheric observations ECMWF Era Interim meteorological fields | FLExKF-TM5-4DVAR_EMPA specific background | The random uncertainties are represented by the posterior error covariance matrix provided by the Kalman Filter, which combines errors in the prior fluxes with errors in the observations and model representation (see description in Appendix A1) | Detailed gridded data can be obtained by contacting the data provider: Dominik.Brunner@empa.ch |
| TM5-4DVAR | Global Eulerian models with a zoom over Europe, ERA-Interim reanalysis | 4DVAR variational techniques | Uncertainty was calculated as 1σ estimate. See descriptions in Appendix A1 | Detailed gridded data can be obtained by contacting the data provider: Peter.BERGAMASCHI@ec.europa.eu |
| FLEXINVERT_NILU | Bayesian statistics Atmospheric transport is modelled using the Lagrangian model FLEXPART | prior fluxes from LPX-Bern DYPTOP, EDGAR v4.2 FT2010 GFED v4 Termites and ocean fluxes ground-based surface $CH_4$ observations. Background fields based on nudged FLEXPART-CTM simulations (Groot Zwaaftink et al., 2018) | | Detailed gridded data $CH_4$ emissions can be obtained by contacting the data provider: Christine Groot Zwaaftink cgz@nilu.no |
| CTE-FMI | Ensemble Kalman filter Eulerian transport model TM5 ECMWF ERA-Interim meteorological data | prior fluxes from LPX-Bern DYPTOP, EDGAR v4.2 FT2010 GFED v4 Termites and ocean fluxes ground-based surface $CH_4$ observations GOSAT $XCH_4$ retrievals from NIES v2.72 | The prior uncertainty is assumed to be a Gaussian probability distribution function The posterior uncertainty is calculated as standard deviation of the ensemble members, where the posterior error covariance matrix are driven by the ensemble Kalman filter. | Detailed gridded data can be obtained by contacting the data provider: aki.tsuruta@fmi.fi |





| InTEM-NAME | Atmospheric Lagrangian transport model analysis 3-D meteorology from the UK Met Office Unified Model | a)the UK National Atmospheric Emissions Inventory (NAEI) 2015 within the UK b)Outside the UK - EDGAR 2010 emissions distributed uniformly over land (excluding the UK) | derived from the variability of the observations within each 2-hr period<br>a) 40%,<br>b) 50% | Detailed gridded data can be obtained by contacting the data provider: Manning, Alistair alistair.manning@metoffice.gov.uk |
|---|---|---|---|---|
| InGOS | 18 European monitoring stations EDGARv4.2FT-InGOS wetland inventory of J.Kaplan and LPX-Bern v1.0 ERA-Interim reanalysis Met Office Unified Model | For Priors please see Table B4 | The uncertainty of the model ensemble was calculated as 1σ estimate. Individual models use Bayes' theorem to calculate the reduction of assumed a priori emission uncertainties by assimilating measurements. | Detailed gridded data can be obtained by contacting the data provider: Peter.BERGAMASCHI@ec.europa.eu |
| **Global inversions from the Global Carbon Project CH₄ budget (Saunois et al. 2020)** | | | | |
| **GCP-CH₄ 2019 anthropogenic and natural partitions from inversions** | ensemble of inversions gathering various chemistry transport models surface or satellite data | For Priors please see Table B4 | Uncertainties are reported as minimum and maximum values of the available studies, as the range of available mean estimates, i.e., the standard error across measurements/methodologies considered. Posterior uncertainty mostly use Monte Carlo methods | Detailed gridded data can be obtained by contacting the data provider: Marielle Saunois marielle.saunois@lsce.ipsl.fr |
| **N₂O bottom-up anthropogenic emissions** | | | | |
| **Data source** | **AD/Tier** | **EFs/Tier** | **Uncertainty assessment method** | **Emission data availability** |
| **UNFCCC NGHGI (2019), MS-NRT (2018), EDGAR v5.0, CAPRI, GAINS and FAOSTAT see above** | | | | |
| ECOSSE | The model is a point model, which provides spatial results by using spatial | IPCC 2006: Tier 3 The simulation results will be allocated due to the available information (size of spatial unit, | N/A | Detailed gridded data can be obtained by contacting the data provider: Kuhnert, Matthias matthias.kuhnert@abdn.ac.uk |





| | distributed input data (lateral fluxes are not considered). The model is a TIER 3 approach that is applied on grid map data, polygon organized input data or study sites. | representation of considered land use, etc.). | | |
|---|---|---|---|---|
| **DayCent** | Spatial explicit simulations at point level, up-scaled at 1km for agricultural areas. | Tier 3; Land management and input factors for the cropland remaining cropland category based on datasets covering the 2005-2015 period. | Monte Carlo | Detailed gridded data can be obtained by contacting the data provider: Emanuele.LUGATO@ec.europa.eu |
| **N₂O bottom-up natural emissions** | | | | |
| **Mechanistic Stochastic Model for N₂O emissions from inland waters** | Hydrosheds 15s (Lehner et al., 2008) and Hydro1K (USGS, 2000) for river network, HYDROLAKES for lakes and reservoirs network and surface area (Messager et al., 2016); Worldwide Typology of estuaries by Dürr et al. (2011); terrestrial N and P loads by Global-NEWS (Van Drecht et al., 2009; Bouwman et al., 2009), resdistributed at 0.5° resolution by Maavara et al., 2019. | EFs applied to denitrification and nitrification rates for N2O emissions. Values constrained from the range reported in Beaulieu et al., 2011. | Two model configurations for N2O | Detailed gridded data can be obtained by contacting the data providers: Ronny Lauerwald Ronny.Lauerwald@ulb.ac.be Pierre Regnier Pierre.Regnier@ulb.ac.be |
| **Regional N₂O inversions over Europe ( high transport model resolution )** | | | | |
| **FLEXINVERT_NILU** | Bayesian statistics Atmospheric transport is modelled using the Lagrangian model FLEXPART | background mole fractions | Random uncertainties are calculated from a Monte Carlo ensemble of inversions | Detailed gridded N2O data can be obtained by contacting the data provider: Rona Thompson rlt@nilu.no |
| **Global N₂O inversions over Europe from GN₂OB (Tian et al., 2020)** | | | | |





| **PYVAR** | Bayesian inversion method observations of atmospheric mixing ratios fluxes from ground-based sites, ship and aircraft transects soil fluxes OCN-v1.2 ocean biogeochemistry model PlankTOM-v10.2 GFED-v4.1s EDGAR-4.32 ECMWF ERA interim | Fires emission factors from Akagi et al., 2011 | Uncertainty in the observation space is calculated as the quadratic sum of the measurement and transport uncertainties. For the error in each land grid cell, the maximum magnitude of the flux in the cell of interest and its 8 neighbours is used; for ocean grid cells the magnitude of the cell of interest only is used. | Detailed gridded $N_2O$ data can be obtained by contacting the data provider: Rona Thompson rlt@nilu.no |
|---|---|---|---|---|
| **TOMCAT-INVICAT** | Variational Bayesian inverse model assimilating surface flask observations of atmospheric mixing ratios. ECMWF ERA-Interim meteorological driving data. | Prior emissions estimates are from OCN-v1.1 model (soils), EDGARv4.2FT2010 (anthro. non-soil), PlankTOM5 (oceans) and GFEDv4.1s (biomass burning). | Uncertainty in the observation space is calculated as the quadratic sum of the measurement and transport uncertainties. For the error in each land grid cell, the maximum magnitude of the flux in the cell of interest and its 8 neighbours is used. Prior emission uncertainties are 100% and uncorrelated. | Detailed gridded $N_2O$ data can be obtained by contacting the data provider: Christopher Wilson [GEO] C.Wilson@leeds.ac.uk |
| **MIROC4-ACTM** | Matrix inversion for calculation of fluxes from 53 and 84 partitions of the globe for $CH_4$ and $N_2O$, respectively. Forward model transport is nudged to JRA-55 horizontal winds and temperature. | Fire emissions for $CH_4$ are taken from GFEDv4s | A posteriori uncertainties are obtained from the Bayesian statistics model. A priori emissions uncertainties are uncorrelated. | Detailed gridded data can be obtained by contacting the data provider: Prabir Patra prabir@jamstec.go.jp |

*Table B2: Biogeochemical models that computed wetland emissions used in this study. Runs were performed for the whole period 2000-2017. Models run with prognostic (using their own calculation of wetland areas) and/or diagnostic (using WAD2M) wetland surface areas (see Sect 3.2.1) From Saunois et al., 2020.*





| Model | Institution | Prognostic | Diagnostic | References |
|---|---|---|---|---|
| CLASS-CTEM | Environment and Climate Change Canada | y | y | Arora, Melton and Plummer (2018) Melton and Arora (2016) |
| DLEM | Auburn University | n | y | Tian et al., (2010;2015) |
| ELM | Lawrence Berkeley National Laboratory | y | y | Riley et al. (2011) |
| JSBACH | MPI | n | y | Kleinen et al. (2019) |
| JULES | UKMO | y | y | Hayman et al. (2014) |
| LPJ GUESS | Lund University | n | y | McGuire et al. (2012) |
| LPJ MPI | MPI | n | y | Kleinen et al. (2012) |
| LPJ-WSL | NASA GSFC | y | y | Zhang et al. (2016b) |
| LPX-Bern | University of Bern | y | y | Spahni et al. (2011) |
| ORCHIDEE | LSCE | y | y | Ringeval et al. (2011) |
| TEM-MDM | Purdue University | n | y | Zhuang et al. (2004) |
| TRIPLEX_GHG | UQAM | n | y | Zhu et al., (2014;2015) |
| VISIT | NIES | y | y | Ito and Inatomi (2012) |

*Table B3: Top-down studies used in our new analysis, with their contribution to the decadal and yearly estimates noted. For decadal means, top down studies have to provide at least 8 years of data over the decade to contribute to the estimate, from Saunois et al., 2020*

| Model | Institution | Observation used | Time period | Number of inversions | References |
|---|---|---|---|---|---|
| Carbon Tracker-Europe CH$_4$ | FMI | Surface stations | 2000-2017 | 1 | Tsuruta et al. (2017) |
| Carbon Tracker-Europe CH$_4$ | FMI | GOSAT NIES L2 v2.72 | 2010-2017 | 1 | Tsuruta et al. (2017) |
| GELCA | NIES | Surface stations | 2000-2015 | 1 | Ishizawa et al. (2016) |

| LMDz-PYVAR | LSCE/CEA | Surface stations | 2010-2016 | 2 | Yin et al. (2019) |
|---|---|---|---|---|---|
| LMDz-PYVAR | LSCE/CEA | GOSAT Leicester v7.2 | 2010-2016 | 4 | Yin et al. (2019) |
| LMDz-PYVAR | LSCE/CEA | GOSAT Leicester v7.2 | 2010-2017 | 2 | Zheng et al. (2018a, 2018b) |
| MIROC4-ACTM | JAMSTEC | Surface stations | 2000-2016 | 1 | Patra et al. (2016; 2018) |
| NICAM-TM | NIES | Surface stations | 2000-2017 | 1 | Niwa et al. (2017a; 2017b) |
| NIES-TM-FLEXPART-VAR (NTFVAR) | NIES | Surface stations | 2000-2017 | 1 | Maksyutov et al. (2020); Wang et al. (2019b) |
| NIES-TM-FLEXPART-VAR (NTFVAR) | NIES | GOSAT NIES L2 v2.72 | 2010-2017 | 1 | Maksyutov et al. (2020); Wang et al., (2019b) |
| TM5-CAMS | TNO/VU | Surface stations | 2000-2017 | 1 | Segers and Houweling (2018); Bergamaschi et al. (2010; 2013), Pandey et al. (2016) |
| TM5-CAMS | TNO/VU | GOSAT ESA/CCI v2.3.8 (combined with surface observations) | 2010-2017 | 1 | Segers and Houweling (2018,report); Bergamaschi et al. (2010; 2013), Pandey et al. (2016) |
| TM5-4DVAR | EC-JRC | Surface stations | 2000-2017 | 2 | Bergamaschi et al. (2013, 2018) |
| TM5-4DVAR | EC-JRC | GOSAT OCPR v7.2 (combined with surface observations) | 2010-2017 | 2 | Bergamaschi et al. (2013, 2018) |
| TOMCAT | Uni. of Leeds | Surface stations | 2003-2015 | 1 | McNorton et al. (2018) |

*Table B4: List of prior datasets for natural CH$_4$ emissions used by all inverse models*

| Project | Model | Prior | | | | | | |
|---|---|---|---|---|---|---|---|---|
| | | Wetlands | Geological | Fire | Termites | Soil sink | Ocean/Lakes | Wild animals |
| VERIFY | CTE_FMI | LPX-Bern DYPTOP (Stocker et al., 2014) | | GFED4s | Ito and Inatomi 2012 | LPX-Bern DYPTOP (Stocker et al., 2014) | Tsuruta et al., 2017 | |





| VERIFY | FLEXPART(FLExKF-TM5-4DVAR)_EMPA | JSBACH-HIMMELI | | | GCP | Ridgwell /GCP | GCP/ULB | |
|---|---|---|---|---|---|---|---|---|
| VERIFY | FLEXINVERT_NILU | LPX-Bern DYPTOP (Stocker et al., 2014) | | GFED4s | Ito and Inatomi, 2012 | LPX-Bern DYPTOP (Stocker et al., 2014) | Tsuruta et al., 2017 | |
| VERIFY | TM5_4DVAR JRC | GCP_CH$_4$_2019 | GCP_CH$_4$ 2019 (global total: 15 Tg CH$_4$ yr$^{-1}$) | | GCP_CH$_4$_2019 | GCP_CH$_4$_2019 | GCP_CH$_4$_2019 | |
| InGOS | INGOS-CTE-S4_EC | LPX-Bern v1.0 (Spahni et al., 2013) | | GFED | Ito and Inatomi 2012 | LPX-Bern v1.0 (Spahni et al., 2013) | Tsuruta et al., 2015 | |
| InGOS | INGOS-LMDZEU-S4_EC | wetland inventory of J. Kaplan (Bergamaschi et al., 2007) | | | | | | |
| InGOS | INGOS-TM3STILT-S4_EC | wetland inventory of J. Kaplan (Bergamaschi et al., 2007) | | | | | | |
| InGOS | INGOS-TM5VAR-S4_EC | wetland inventory of J. Kaplan (Bergamaschi et al., 2007) | | | Sanderson /GCP | Ridgwell /GCP | Lambert /GCP | Oslson climatology |
| InGOS | INGOS-NAME-S4_EC | wetland inventory of J. Kaplan (Bergamaschi et al., 2007) | | | | | | |
| GCP | GELCA-SURF_NIES | VISIT (Ito and Inatomi, 2012) | n/a | GFEDv3.1 then GFAS v1.2 after 2011 | Sanderson (TransCom-CH$_4$ / GCP) | VISIT (Ito and Inatomi, 2012) | n/a | |
| GCP | MIROCv4-SURF_JAMASTEC | VISIT (Ito and Inatomi, 2012) (global total range : 173-197 Tg CH$_4$ yr$^{-1}$) | Etiope and Milkov, 2004 (global total: 7.5 Tg CH$_4$ yr$^{-1}$) | GFEDv4s (global total range : 14-35 Tg CH$_4$ yr$^{-1}$) | Sanderson (TransCom-CH$_4$) (global total: 20.5 Tg CH$_4$ yr$^{-1}$) | VISIT (Ito and Inatomi, 2012) | Lambert/Houweling (TransCom-CH$_4$) (global | |

2000




| | | | | | | | | |
|---|---|---|---|---|---|---|---|---|
| | | | | | | | total: 18.5 Tg CH$_4$ yr$^{-1}$) | |
| GCP | NICAM-SURF_NIES | VISIT (Ito and Inatomi, 2012) | GCP based on Etiope 2015 | GFEDv4s / GCP | Sanderson (TransCom-CH$_4$ / GCP) | VISIT (Ito and Inatomi, 2012) | Lambert/Houweling (TransCom-CH$_4$ / GCP) | |
| GCP | TOMCAT-SURF_ECMWF | JULES emissions from Mc Norton 2016a | Tomcat 2006 | GFED V4 | Matthews and Fung 2006 | Patra et al. 2011 | Tomcat 2006 Matthews and Fung 1987 - all emissions total rescaled to Schwietzke et al. 2016 | |
| GCP | NTFVAR-GOSAT_NIES | VISIT (Ito and Inatomi, 2012) | Etiope and Milkov, 2004 | GFAS v1.2 | Ito and Inatomi 2012 | VISIT (Ito and Inatomi, 2012) | TransCom-CH4 | |
| GCP | NTFVAR-SURF_NIES | VISIT (Ito and Inatomi, 2012) | Etiope and Milkov, 2004 | GFAS v1.2 | Ito and Inatomi 2012 | VISIT (Ito and Inatomi, 2012) | TransCom-CH4 | |
| GCP | LMDZ-GOSAT1_LSCE | Bloom 2017 | n/a | GFED V41s | Sanderson /GCP | Ridgwell /GCP | Lambert /GCP | |
| GCP | LMDZ-GOSAT2_LSCE | GCP - ensemble mean ESSD Saunois et al. 2016 | GCP based on Etiope 2015 | GFED V41s | Sanderson /GCP | Ridgwell /GCP | Lambert /GCP | |
| GCP | LMDZ-GOSAT3_CALTECH LMDZ-GOSAT4_CALTECH LMDZ-GOSAT5_CALTECH LMDZ-GOSAT6_CALTECH LMDZ-SURF1_CALTECH LMDZ-SURF2_CALTECH | Kaplan 2002 rescaled by Bergamaschi 2007 | n/a | GFED V41 | Sanderson 1996 /GCP | Ridgwell /GCP | Lambert and Schmidt 1993 | |





| GCP | TM5-CAMS-GOSAT_TNO | Kaplan climatology | n/a | GFED V31 climatology after 2011 | Sanderson /GCP | Ridgwell /GCP | Lambert /GCP | Oslson climatology |
|---|---|---|---|---|---|---|---|---|
| GCP | TM5-GOSAT1_EC | WETCHIMP ensemble mean; | GCP_CH$_4$ 2019 (global total: 15 Tg CH$_4$ yr$^{-1}$) | | Sanderson /GCP | Ridgwell /GCP | Lambert /GCP | Oslson climatology |
| GCP | TM5-GOSAT2_EC | GCP_CH$_4$_2019 | GCP_CH$_4$ 2019 (global total: 15 Tg CH$_4$ yr$^{-1}$) | GCP_CH$_4$_2019 | GCP_CH$_4$_2019 | GCP_CH4_2019 | GCP_CH$_4$_2019 | |
| GCP | TM5-SURF1_EC | WETCHIMP ensemble mean; | GCP_CH$_4$ 2019 (global total: 15 Tg CH$_4$ yr$^{-1}$) | | Sanderson /GCP | Ridgwell /GCP | Lambert /GCP | Oslson climatology |
| GCP | TM5-SURF2_EC | GCP_CH$_4$_2019 | GCP_CH$_4$ 2019 (global total: 15 Tg CH$_4$ yr$^{-1}$) | GCP_CH$_4$_2019 | GCP_CH$_4$_2019 | GCP_CH$_4$_2019 | GCP_CH$_4$_2019 | |
| GCP | CTE-GOSAT_FMI | GCP_CH$_4$_2019 | Etiope 2015 | GCP_CH$_4$_2019 (=GFED4s) | GCP_CH$_4$_2019 | GCP_CH$_4$_2019 | GCP_CH$_4$_2019 | |
| GCP | CTE-SURF_FMI | GCP_CH$_4$_2019 | Etiope 2015 | GCP_CH$_4$_2019 (=GFED4s) | GCP_CH$_4$_2019 | GCP_CH$_4$_2019 | GCP_CH$_4$_2019 | |
| | NAME-SURF_MetOffice | | | | | | | |

**Author contributions**

A.M. R. P., A. J. D. and P.C designed research and led the discussions, A. M. R. P. wrote the initial draft of the paper and edited all the following versions; C. Q. made the figures, P. C. and C. Q. edited and commented to the first versions
of the manuscript, G. P. provided the figures xyz and made a very detailed review on a previous version, P. P. and M. J. M. processed the original data submitted to the VERIFY portal, M. J. M., P. P. and P. B. designed and are managing the web portal, E. S. developed the methodology for the EDGAR uncertainty calculation and provided the CH$_4$ and N$_2$O uncertainties, R. L. T., G. J.-M., F. N. T., P. B., D. B., L. H.-J., P. R., R. L., D. B., W. W., A. J. D., provided in depth advice and commented/edited the initial versions of the manuscript, L. P. and D. G. provided advice and edited
the text related to the UNFCCC and NGHGI information, R. L. T., P. B., D. B., L. H-J., P. R., R. L., A. T., W. W., P. K. P., M. K., G. D. O., M. C., M. S., T. M., T. A., C. D. G. Z., Y. Y., A. L., A. L., G. J.-H., A. J. M., J. McN. are data providers, all co-authors read and made changes to the more advanced versions of the manuscript and provided specific comments related to their data.

*Competing interests*
The authors declare that they have no conflict of interest.



**Acknowledgements**

FAOSTAT statistics are produced and disseminated with the support of its member countries to the FAO regular budget. The views expressed in this publication are those of the author(s) and do not necessarily reflect the views or policies of FAO. Philippe Ciais acknowledges the support of European Research Council Synergy project SyG-2013-610028 IMBALANCE-P and from the ANR CLand Convergence Institute. Prabir Patra acknowledges support of the Environment Research and Technology Development Fund (JPMEERF20172001, JPMEERF20182002,) of the Environmental Restoration and Conservation Agency of Japan. David Basviken acknowledges support of the European Research Council (ERC) under the European Union's Horizon 2020 research and innovation programme (grant agreement No 725546). We acknowledge the work of the entire EDGAR group (Marilena Muntean, Diego Guizzardi, Edwin Schaaf, Jos Olivier). DB was supported by the European Research Council (ERC) under the European Union's Horizon 2020 research and innovation programme (grant agreement No 725546). Tuula Aalto acknowledges support from Caroline-Herschel-FPA under EU Horizon 2020 programme (FPCUP, Grant no 809596), and Academy of Finland (SOMPA, Grant no 312932). CGZ acknowledges support by the Norwegian Research Council (ICOS-Norway, project 245927). JM acknowledges financial support from the Horizon2020 CHE project (776186).

**Financial support**

This research has been supported by the European Commission, Horizon 2020 Framework Programme (VERIFY, grant no. 776810).

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
