# Peer review of "The consolidated European synthesis of CH4 and N2O emissions for EU27 and UK: 1990-2017"

_Earth System Science Data, 2020_

## Referee Comment (RC1) · Anonymous Referee #1 · 25 Jan 2021

The authors present a comprehensive synthesis of CH4 and N2O emissions over Europe from 1990-2017, and Near Real Time estimates from EEA for the year 2018 was used as a supplement. The data compiled in this study will be very useful for scientific community when building their own estimate in the future, and also be relevant for quantifying the progresses towards mitigation target assessed through the global stocktake. Various estimates derived from both bottom-up and top-down methods were compiled and compared with UNFCCC NGHGI for EU27+UK in this manuscript, while data for other sub-regions of Europe were only provided on a website. The manuscript is well structured and well written in general. However, some descriptions and clarifications of all the data provided in this study will be essential for readers to be actually ben-

efit from the compilation done in this study. For example, to facilitate the use of the data by others, it could be very useful to provide a detail instruction explaining how the data shared on zenodo was structured; how to read and process them automatically (e.g., providing a R/python program/package); I also noticed that the files provided on zenodo are only the numbers used to plot the figure shown in the manuscript and were separated to many small csv files with one line only and less than 1k size. I believe the date provided should be well organized for users (e.g., one file with overviews presented and multiple sheets each for a sector etc.). As a reader, I would mostly be interested in the original data presented in this study including country-specific values rather than aggregated to a few regions only. Depositing original values from each methods shown in the manuscript are essential for such publication (especially on journals like ESSD). As mentioned in the introduction, this manuscript focus on three questions. And in the summary and concluding remarks, they are more or less discussed. But I would expect more structural synthesis of the potential answers for these three questions. In addition, the only 2018 value was provided by MS-NRT as preliminary estimate. All the comparisons were done before 2017. Thus I would think the title should be changed to 1990-2017. Other remarks: L66: "compared to 0.9 Tg N2Oyr-1 from the BU data" may not be necessary. L148-160: It is not clear which gas or gases were referred to in this paragraph. L158: The abbreviation "LULUCF" was not explained before this line. L163: "from" is redundant. L171: Should it be "come from" instead "belong to"? L174: Please check the name of Tables in Appendix A. Why not using Table A1 and A2? Footnote 5: Why only Vol 5 of IPCC was cited? I believe methods from many other sectors and Volumes were used. Footnote 6: You mean "With natural CH4, we . . ." L225-230: It still is hard to understand how the emissions were scaled. L235: There is no Table 5. L266: "SURF" was not explained before. L268: The last sentence is not clear. L296: It seems not only anthropogenic emissions were described in section 3.1. L298: What are the differences between European and regional total? It looks like sub-regional total. L342: It is not clear how a less dense network of surface stations cause the BU = TD estimates. Some further explanations are needed. L534: Is there any reason

that the GOSAT-derived estimates are so different from SURF-derived ones? L538-542 (and also across the manuscript): Can the uncertainty of the trends be given? In addition, it would be interesting to compare the trends of the common period 2010-2016. Otherwise, they are not comparable. L555: Can the authors give range or % uncertainty of TD estimates for a comparison? L581: The total EU27+UK GHG emissions in CO2eq here is different from that in section 3.1.2. Please check. Figure 8: Is there any explanation of the sudden shift of EDGARv5.0 estimates in Fig. 8e? L650-652: It is not clear how the higher TD estimate can be attributed to the seasonal cycle. More explanations are needed. L705: This sentence is not correctly written. L706: "than" rather than "then". L710: "fewer observations" in a specific region? I believe for global inversion, the total number of observations are more than those used in regional inversions. Table A: there are so many different grouping of European countries, which is very confusing. Why such groupings are used? Should it be further classified e.g., as countries, geographical sub-regions etc.?

---

## Referee Comment (RC2) · Anonymous Referee #2 · 10 Feb 2021

The authors provide the most comprehensive assessment of CH4 and N2O emissions for EU27 and UK during 1990-2008 using top down (emission inventory, ecosystem modeling) and bottom up (inverse modeling) approaches. The results from the synthesis with uncertainties are then compared with European NGHGI data in 2019. This is a strong paper with valuable information to the scientific community and certainly the one that resonates with the readers of ESSD. I also commend the authors approach to use dozens of datasets to provide the most comprehensive assessment of these fluxes (its never easy to do that). The manuscript is well written (although the text can be trimmed significantly), logically organized and clearly presented. Below are my comments which probably would be helpful for improving the manuscript.

[Figure]

1. The results are presented as individual CH4 and N2O fluxes (figures 1-9). It would be great to have these figures together particularly for both gases as it will increase the readability especially when comparing the CH4 and N2O fluxes for each region. Currently, its hard to look at these figures of CH4 and N2O fluxes separately and derive any conclusion or recommendation on the dominant control of CH4 on N2O fluxes for different regions. 2. I am left wondering about why only CH4 and N2O emissions are included in this study. Since the study uses ground based observation, ecosystem modeling and inverse modeling, there should also be availability of data related to CO2 fluxes and that would provide a big picture of the net GHG for EU27 and UK. Adding CO2 into the current CH4 and N2O fluxes would be valuable not only to identify regions that are GHG sources/sinks but also to carry out large scale mitigation effort depending on the dominant control of individual GHG at the given location. 3. Datasets: Since this is a data paper, I only see the aggregated data provided in Fig 1-9. I strongly suggest the authors to provide these data at pixel level so that it can be meaningful and useful to other colleagues working on CH4 and N2O fluxes for the UK and EU27 region. I am also wondering whether appropriate approach has been made to use data from all the papers that the authors have cited and whether there has been an agreement on making the data open source through this paper. 4. I have also seen unexpected citation approach (for example: line 160-165). The authors cite Yuanzhi Yao as personal communication for 66% of the N2O emitted from rivers are considered anthropogenic. This needs an appropriate citation with 66% of what, and how what is the contribution of the rivers toward total N2O fluxes (I do believe it should be relatively small compared to fluxes from agricultural lands). 5. Uncertainty: I am still not convinced about how the uncertainty was assessed since the data came from different sources. For example, if the authors are using ecosystem models, is it appropriate to use the standard deviation to determine uncertainty in N2O and CH4 fluxes. I do believe that model uncertainty comes from parameter use, the model structure and uncertainty in input datasets. However, the authors have not tried to address this issue in the manuscript. 6. Tables: While there are many details on the datasets used to estimate N2O and

CH4, there is no any tables that shows the emissions from different sector when these datasets are compiled together. I strongly suggest the authors to provide the top-down and bottom up N2O and CH4 fluxes in one table with different sources (agriculture, natural vegetation, wetlands etc). 7. Table 1,2 and 3 all can go in supplementary material. These tables are just taking too much space in the manuscript and given that Table 3 is adopted from some other paper, I do not think it should be in the main content. 8. Seasonal flux estimates: Currently, the manuscript estimates CH4 and N2O fluxes at annual time scale and completely ignore the fact that understanding seasonal dynamics of these fluxes are important and useful for climate mitigation efforts. At least, there should be an acknowledgement on why seasonal fluxes were not estimated.

---

## Author Comment (AC1) · 12 Mar 2021

Dear Topical Editor Nellie Elguindi, Dear Referees and Editorial Board of ESSD,

As requested, we are submitting responses to the referees' comments. We will provide as well a track-change version of the manuscript. We will not refer here to grammar or language corrections, but they will appear in the marked-up manuscript. The lines in the following answers refer to the track-change version of the manuscript.
REPLY TO THE REFEREE #1 The authors thank Referee #1 for the thoughtful and helpful comments and for the fact that the Referee acknowledges the manuscript as being a comprehensive collection of data, very useful for modelers and the whole scientific community and for quantifying the progresses towards mitigation target assessed through the global stocktake. Below we provide answers to the general and specific comments posted by Referee #1. One general comment: as requested by our co-author Rona Thompson, the N2O inversion model PYVAR was renamed with CAMS-N2O (Figure 9).

General evaluation: This study is intended to be updated annually, similar to the GCP papers (Friedlingstein et al., 2019), and to evolve into a complete synthesis of bottom-up and top-down GHG estimates of European countries and ecosystems. While the GCP provides the global carbon budget, the current study starts a series of datasets for EU. These are essential for the GHG Monitoring and Verification Support (MVS) capacity that the EU envisages to build in support of the enhanced transparency framework of the Paris Agreement (e.g. Janssen-Maenhout, 2020). Our data access is similar to that provided by GCP. "The manuscript is well structured and well written in general. However, some descriptions and clarifications of all the data provided in this study will be essential for readers to be actually benefit from the compilation done in this study. For example, to facilitate the use of the data by others, it could be very useful to provide a detail instruction explaining how the data shared on zenodo was structured" The Zenodo data https://doi.org/10.5281/zenodo.4590875 (this is the new doi number, data version v2) are the data for the figures, according to the policy of ESSD, and we believe is sufficient to ensure easy replicability. For most of the raw gridded data underlying these aggregated data (i.e. for EU27+UK) the co-authors would prefer to be first contacted, in line with their own data policy. We chose to add their contact details or the direct link from where the data can be downloaded (if free) in Appendix B, Table B2. As the referee suggested, we compiled all the aggregated data files (associated to each figure) in one aggregated file with different spreadsheets per figure and for the files containing the means we added as well the entire time series from where this

mean value was calculated. Regarding the description of the datasets, in the revised version we clarify the situation of data availability accordingly: Line 154: "References are given in Table 2 and the detailed description of all products in Appendix A1-A3." In Appendix A1-A3 we describe each source containing information on spatial resolution, time steps and we updated as well the data in the excel sheets to ensure a better readability as suggested by the referee. "how to read and process them automatically (e.g., providing a R/python program/package)" Given that all data is already provided in a basic input table csv file(s), we think that this is enough to ensure plotting, and for the precise replicability of the style/layout of the figures, if needed, we could make codes available upon request. We added on L185 the following sentence: "Upon request, we can provide the codes necessary to plot precisely the style/layout of the figures." "I also noticed that the files provided on zenodo are only the numbers used to plot the figure shown in the manuscript and were separated to many small csv files with one line only and less than 1k size. I believe the date provided should be well organized for users (e.g., one file with overviews presented and multiple sheets each for a sector etc.). As a reader, I would mostly be interested in the original data presented in this study including country-specific values rather than aggregated to a few regions only." The scope of this synthesis is to present the data sets for EU27+UK. For original gridded data as well as for the complete data of all other regions/countries not used in this study (italic in Appendix A, Table A1), due to data provider policy, it should be requested directly from the specific contact (Appendix B, table B2). The data policy of the VERIFY project (consortium governing document) which supported most of the research presented here restricts the free use of raw data (gridded products) for the first 12 months after its publication, as it may not be entirely published by the data providers. Therefore, we agreed to only make public aggregated data (time series, means/min/max values and aggregated uncertainties). Given that this paper discusses only EU27+UK grouping we provide the aggregated time series only for that region. We intend to submit to ESSD yearly updates for the European GHG budget and once the new update will be published the old version of previous synthesis will be made publicly available

through the VERIFY web-site (http://verify.lsce.ipsl.fr/index.php/products). For a better explanation we added on L705 the following statement: "The raw gridded data, according to the VERIFY consortium governing document, will be made publicly available 12 months after its publication in ESSD, through the VERIFY web site". We provide as well at http://webportals.ipsl.jussieu.fr/VERIFY/FactSheets/ all figures and country-specific data, which are not included in the study but are specifically requested as a VERIFY project output. This publication represents a scientific compilation of these results and we clearly state (Lines 185-190) that all details as well as data for countries/groups of countries can be found on the VERIFY web portal. To access the web portal one needs to register once in order for us to understand the extent of public interest in the data, but there is no restriction on downloading the data. The figures can be saved and data behind figures can be downloaded.

"As mentioned in the introduction, this manuscript focus on three questions. And in the summary and concluding remarks, they are more or less discussed. But I would expect more structural synthesis of the potential answers for these three questions." For a first compilation and study of this type for the EU27 and UK we tried to identify some questions and, as mentioned in the introduction, at this point in time we are not able to provide complete answers to these specific questions, but they were set in order to provide continuity and give scope to the next synthesis. As we stated "A comprehensive investigation of detailed differences between all datasets is currently beyond the scope of this paper". In the conclusions we discuss the findings in more detail but it will take significantly more research to define concrete actions on how these issues can be solved. "In addition, the only 2018 value was provided by MS-NRT as preliminary estimate. All the comparisons were done before 2017. Thus I would think the title should be changed to 1990-2017." It is true and we agree with the reviewer's comment that the actual comparison is done until 2017. Next to MS-NRT values, only ECOSSE (Fig. 6d Agriculture N2O emissions) provided us with time series until 2018. Therefore, we will change the title and text accordingly (L46, L149, L152) We will upload next to our responses the track-changes version of the manuscript. Response to specific

comments and changes in manuscript: Other remarks: L66: "compared to 0.9 Tg N2Oyr-1 from the BU data" may not be necessary. We deleted this statement. L148-160: It is not clear which gas or gases were referred to in this paragraph. The first sentence of the paragraph L148 states that we refer to CH4 and N2O, but to clarify on line 155 we added the following: "For both CH4 and N2O BU approaches, we used inventories of anthropogenic emissions covering all sectors (EDGAR v5.0 and GAINS) and inventories limited to agriculture (CAPRI and FAOSTAT). For CH4 we used one biogeochemical...."

L158: The abbreviation "LULUCF" was not explained before this line. On Line 160 we added the following explanation: "Biomass burning emissions of CH4 from land use, land use change and forestry (LULUCF) sector account for..." L163: "from" is redundant. Was deleted L171: Should it be "come from" instead "belong to"? Thank you, we replaced with "come from" L174: Please check the name of Tables in Appendix A. Why not using Table A1 and A2? This is true, thank you, we replaced Table A with Table A1 and Table AA with Table A2. Footnote 5: Why only Vol 5 of IPCC was cited? I believe methods from many other sectors and Volumes were used. We do cite here IPCC 2006 and IPCC refinement 2019, and not only vol. 5. Footnote 6: You mean "With natural CH4, we . . ." We added the following explanation: The term natural refers here to unmanaged natural CH4 emissions (wetlands, geological, inland waters) not reported under the UNFCCC LULUCF sector.

L225-230: It still is hard to understand how the emissions were scaled. Thank you, We added in Appendix A2, geological flux description (L 1137) the following explanation: "To calculate geological CH4 emissions we used literature data for geological emissions on land (excluding marine seepage) (Etiope et al., 2019; Hmiel et al., 2020). From the gridded geological CH4 emissions by Etiope et al., and using the land-sea mask for EU27+UK (to exclude marine seepage), geological CH4 emissions from the land of EU27+UK are 8.83 Tg/yr. Then we scaled this number by the ratio of global geological CH4 emissions estimated by Hmiel et al. and by Etiope et al., thus obtaining a value of 8.83*5.4/37.4=1.3 Tg/yr (marine and land geological).. The global total geological CH4 emissions reported by Etiope et al. and Hmiel et al. are 37.4 Tg/yr and 5.4 Tg/yr, respectively." In his paper Hmiel et al 2020 compared his numbers to those of Etiope et al 2019 and wrote the following explanation: "(1) the uncertainties associated with global upscaling of geological emissions from discrete measurements result in overestimation by an order of magnitude, or (2) geological CH4 emissions quantified by these measurements were not present in the preindustrial era and may have been triggered by fossil fuel extraction from hydrocarbon reservoirs or other anthropogenic activity such as groundwater aquifer depletion. If the latter is true, such emissions cannot be considered natural."

L235: There is no Table 5. That is correct, in L272 we corrected the number to now it refer to Table 2. L266: "SURF" was not explained before. We added on L 269 the surface station (SURF) explanation. L268: The last sentence is not clear. We think the sentence the referee refers to is: "None of the regional inversions use GOSAT prior data as all base their prior data on SURF stations" In the previous paragraph L264-L269 we explain that global inversions base their prior information on both GOSAT and SURF data. We made this statement as regional inversions will not use GOSAT but only SURF. We changed on L271 to: "All regional inversions use observations from SURF stations as a base of their emission calculation".

L296: It seems not only anthropogenic emissions were described in section 3.1. This is true, we deleted the word anthropogenic, as natural emissions appear as well in 3.1.3 and 3.1.4. We did the same for the N2O section 3.2.

L298: What are the differences between European and regional total? It looks like sub-regional total. We are sorry but we do not understand this comment and what sub-regional total refers to. If the comment addresses results from Figure 1, our aim was not to compare the regions to the total EU but instead to highlight differences between BU and TD estimates within the region and between regions.

L342: It is not clear how a less dense network of surface stations cause the BU = TD estimates. Some further explanations are needed. We replaced this sentence: "For Eastern Europe we note that BU anthropogenic estimates have the same magnitude as the TD. We hypothesize that this could be due to a less dense network of surface stations.". with the following explanation (L348-353): "For Eastern Europe we note that BU anthropogenic estimates have the same magnitude as the TD. One possible explanation is linked to the fact that in TD estimates (i.e. using atmospheric inversions) the fluxes are strongly constrained in regions with a high density of observations. Where there are few or no observations, the fluxes in the inversion will stay close to the prior estimates, since there is little or no information to adjust them. The prior estimates are, in fact, the BU estimates, which means the TD and BU estimates will be similar."

L534: Is there any reason that the GOSAT-derived estimates are so different from SURF-derived ones? Regarding the difference between global and regional inverse estimates, one possible explanation is that satellites (including GOSAT) are not as sensitive to the lower troposphere (the part of the atmosphere closest to the fluxes) as ground-based observations. Therefore, it can be that the inversions with GOSAT miss some of the information that is available from the ground-based observation.

L538-542 (and also across the manuscript): Can the uncertainty of the trends be given? In addition, it would be interesting to compare the trends of the common period 2010-2016. Otherwise, they are not comparable. We thank referee for noticing this inconsistency. Looking at Figure 5, we feel that neither SURF nor GOSAT ensemble shows a linear reduction from 2010-2016, therefore we replaced the paragraph: "Regarding trends, for total CH4 emissions (Figure 5a), the SURF and GOSAT ensemble show a decreasing trend of -1.2 % yr-1 and -0.6 % yr-1, respectively, over the period covered by each of them (SURF: 2000-2016; GOSAT: 2010-2017). For anthropogenic CH4 emissions (Figure 5b), the SURF ensemble shows a decreasing trend of -1.4 % yr-1 compared to -1.5 % yr-1 for the NGHGI over 2000-2016, while the GOSAT ensemble shows a decreasing trend of -0.8 % yr-1 compared to -0.9 % yr-1 for the NGHGI

over 2010-2017." and instead of providing the annual decreasing rates, we only provide the change from 2010 to 2016 (L546-L551) as following: "For the 2010-2016 common period, the two ensembles of regional and global models give an anthropogenic CH4 emission mean (Figure 5b) of 17.4 Tg CH4 yr-1 (GOSAT) and 23.7 Tg CH4 yr-1 (SURF) compared to 19.0 ± 1.7 Tg CH4 yr-1 for NGHGI (Fig. 5b). For the same period total CH4 emissions (Figure 5a) from the SURF and GOSAT ensemble decrease by 0.5% and 4.6%, respectively. For anthropogenic CH4 emissions (Figure 5b), the SURF and GOSAT ensemble show a decrease of 1.1 % and 6.3%, respectively, compared to 7.3% for the NGHGI from 2010 to 2016".

We also added to the Figure 5 caption the following sentence related to the common period to calculate the means: "Two out of 11 SURF products (GELCA-SURF_NIES, TOMCAT-SURF_UOL) were not available for 2016".

L555: Can the authors give range or % uncertainty of TD estimates for a comparison? We added the following sentence on L572: "for the EU27+UK, global inversions show a min/max range of 25-32 % while regional inversions show a variability range of 9-11 % compared to the mean 2011-2015 value". L581: The total EU27+UK GHG emissions in CO2eq here is different from that in section 3.1.2. Please check. Thank you for your observation. It is different because in section 3.1.2 the values include emissions from LULUCF while in section 3.2.2 exclude LULUCF. We changed accordingly, referring in both sections to "include LULUCF" (L596). Figure 8: Is there any explanation of the sudden shift of EDGARv5.0 estimates in Fig. 8e? Thank you. Yes, the sudden shift in total emissions is driven by a shift after the year 2000 for EDGAR v5.0 waste N2O emissions is due to the waste water treatment domestic (WWT.DOM.N2O) activity data. For the v5.0_FT2018, version used in this plot, EDGAR was updated for the period 2000-2016 using FAO statistics on "protein_supply_kg_cap_yr". For the previous period 1970-1999 the time series from EDGAR v4.3.2 were kept unchanged. This particular jump is due to France. We added this explanation in the manuscript, L636

L650-652: It is not clear how the higher TD estimate can be attributed to the seasonal

cycle. More explanations are needed. "The higher emissions from the TD estimates could be attributed to the seasonal cycle (e.g. fertilizer application) not accounted for in the NGHGI reporting." We have changed the above sentence to the following (L679-L683): "The higher emissions from TD estimates may be at least in part due to the fact that they include natural emissions of $N_2O$, which are not considered in NGHGI reporting. One estimate (from the O-CN land ecosystem model) is that the natural emissions could amount to 11% of those reported in NGHGI for the EU27+UK region. In addition, the EFs used in NGHGI reporting are very uncertain (up to 300% for direct agricultural emissions) so there may be a systematic error in these"

L705: This sentence is not correctly written. On new L737 we reformulate as following: "Some studies (Fronzek et al., 2018) show that model ensembles work well in simulating highly uncertain variables." L706: "than" rather than "then". We corrected. L710: "fewer observations" in a specific region? I believe for global inversion, the total number of observations are more than those used in regional inversions. "The global models are less well constrained as they have lower resolution (hence larger representation errors) and often use fewer observations." We replaced the above yellow sentence to the following (L743-747): "The global models use fewer observations for Europe compared to the European regional inversions, and thus are expected to have larger uncertainties for the European fluxes. In addition, the global models are at coarser resolution, and thus likely have larger model representation errors compared to the regional ones, which may contribute to further systematic uncertainty for the European fluxes."

Table A: there are so many different grouping of European countries, which is very confusing. Why such groupings are used? Should it be further classified e.g., as countries, geographical sub-regions etc.? It is true that this study focuses only on EU27 and UK data with aggregated data for the five regions presented in Figures 1 ($CH_4$) and 6 ($N_2O$). In the revised version, this purpose is made clearer to the reader of Table A1 in its caption: "Table A1: Country grouping used for reconciliation purposes between BU and TD estimates. The countries and groups of countries in italic are not

directly used by this study but their figures and data are available on the VERIFY project web portal at: http://webportals.ipsl.jussieu.fr/VERIFY/FactSheets/." Because VERIFY is the funder of this work and the project committed to provide European Commission with a regional and country based analysis of GHG estimates, we see the need of mentioning all those diverse groups and countries in the Appendix table A1.

――――――――――――――――――

---

## Author Comment (AC2) · 12 Mar 2021

Dear Topical Editor Nellie Elguindi, Dear Referees and Editorial Board of ESSD,

As requested, we are submitting responses to the referees' comments. We will provide as well a track-change version of the manuscript. We will not refer here to grammar or language corrections, but they will appear in the marked-up manuscript. The lines in the following answers refer to the track-change version of the manuscript.
REPLY TO THE REFEREE #2 The authors thank Referee #2 for acknowledging this study as being the most comprehensive assessment of CH4 and N2O emissions for EU27+UK, as well as being very useful for the modelers and the whole scientific community. We indeed agree that such a comparison was not easy and straight forward and we thank Referee #2 for the comments to which we answer below. 1. The results are presented as individual CH4 and N2O fluxes (figures 1-9). It would be great to have these figures together particularly for both gases as it will increase the readability especially when comparing the CH4 and N2O fluxes for each region. Currently, its hard to look at these figures of CH4 and N2O fluxes separately and derive any conclusion or recommendation on the dominant control of CH4 on N2O fluxes for different regions.

We thank the Referee for this comment. We mainly chose this structure of the paper (section 3.1 to address CH4 and section 3.2 to address N2O) to better focus on differences we've found between BU and TD estimates for each gas. If we would have presented in parallel both CH4 and N2O, would have been difficult to individualize and discuss specific findings and the overall discussion would have been confusing (we tried this before in an early version of the manuscript and most of co-authors suggested and agreed on keeping separate the two gases, each with its own section).

2. I am left wondering about why only CH4 and N2O emissions are included in this study. Since the study uses ground based observation, ecosystem modeling and inverse modeling, there should also be availability of data related to CO2 fluxes and that would provide a big picture of the net GHG for EU27 and UK. Adding CO2 into the current CH4 and N2O fluxes would be valuable not only to identify regions that are GHG sources/sinks but also to carry out large scale mitigation effort depending on the dominant control of individual GHG at the given location. In the same time with this study, the companion paper dedicated to CO2 was/is as well in review in ESSD (we mention it on Line 185). Please find below the link to access the manuscript. https://essd.copernicus.org/preprints/essd-2020-376/

3. Datasets: Since this is a data paper, I only see the aggregated data provided in Fig
1-9. I strongly suggest the authors to provide these data at pixel level so that it can be meaningful and useful to other colleagues working on CH4 and N2O fluxes for the UK and EU27 region. I am also wondering whether appropriate approach has been made to use data from all the papers that the authors have cited and whether there has been an agreement on making the data open source through this paper. Similar to the comment received from Referee #1, we believe that the data behind figures is sufficient for a correct replicability of the figures and represents the complete country/regional information. More detailed data at pixel level (if available, e.g., some models only provide country totals) should be asked from co-authors in accordance with their individual data policy (Appendix B2, Table B1). On the VERIFY web portal we present as well figures at the country level, which can be used by interested parties to check accuracy of the regional totals. This information is freely accessible with only a simple registration needed to keep track with whom is using the data, which sometimes is updated on the website but not yet published. http://webportals.ipsl.jussieu.fr/VERIFY/FactSheets/ Regarding the agreement and the open source data, the data policy of the VERIFY project (consortium governing document), which supported most of the research presented here, restricts the free use of raw data (gridded products) for the first 12 months after its publication, as it may not be entirely published by the data providers. Therefore, we agreed to only make public aggregated data. We intend to submit to ESSD yearly updates for the European GHG budget and once the new update will be published the old version of previous synthesis will be released as publicly available.

4. I have also seen unexpected citation approach (for example: line 160-165). The authors cite Yuanzhi Yao as personal communication for 66% of the N2O emitted from rivers are considered anthropogenic. This needs an appropriate citation with 66% of what, and how what is the contribution of the rivers toward total N2O fluxes (I do believe it should be relatively small compared to fluxes from agricultural lands). The 66 % was calculated for this study by our co-author, Yuanzhi Yao. It is not published in a peer-reviewed study and we wanted to acknowledge his work. We changed accordingly throughout the text "pers. comm." with "in this study". (L166, L245, L282, L687).

On L281 we also completed the sentence as following (in bold): "Note that the estimates of Maavara et al. (2019) and Lauerwald et al. (2019) include anthropogenic emissions from N-fertilizer leaching accounting for 66% of the inland water emissions in EU27+UK. In 2016, emissions from rivers represent 2.2 % of the total UNFCCC NGHGI (2019) N2O emissions."

5. Uncertainty: I am still not convinced about how the uncertainty was assessed since the data came from different sources. For example, if the authors are using ecosystem models, is it appropriate to use the standard deviation to determine uncertainty in N2O and CH4 fluxes. I do believe that model uncertainty comes from parameter use, the model structure and uncertainty in input datasets. However, the authors have not tried to address this issue in the manuscript. We agree that BU and TD approaches differ as well as the way uncertainties are quantified. Therefore, for TD results we agreed to use a neutral approach and define as uncertainty the variability we've seen for the model ensembles, calculating the mean, and set the uncertainty range as the min/max values. For BU models and inventories, only EDGAR v5.0 provided us for 2015 with an uncertainty estimate which, for comparability purposes, was calculated with the same method (the error propagation method (95% confidence interval) according to IPCC (2006, chap. 3, Eq. 3.7).) as the UNFCCC NGHGIs. We refer to Petrescu et al., 2020 for the EDGAR uncertainty methodology calculation https://essd.copernicus.org/articles/12/961/2020/#section8

6. Tables: While there are many details on the datasets used to estimate N2O and CH4, there is no any tables that shows the emissions from different sector when these datasets are compiled together. I strongly suggest the authors to provide the top-down and bottom up N2O and CH4 fluxes in one table with different sources (agriculture, natural vegetation, wetlands etc). Yes, we added the two tables below to the Appendix B1 Overview tables: Table B1a, B1b. Table B1a: Comparison of CH4 results from the BU and TD methods for common periods: BU Anthropogenic 1990-2015, BU and TD natural 2005-2011 and TD total 2006-2012 representing the common period between all data sets and the last year available. All values are in kton CH4 per year. The UNFCCC NGHGI uncertainties represent the 95% confidence interval; uncertainty for EDGAR v5.0 was calculated for 2015 and the min/max values for all sectors are as following: Energy: 33/37, IPPU: 39/34, Agriculture: 18/18, Waste: 32/38; the uncertainty represents the 95 % confidence interval of a lognormal distribution. The other uncertainties represent the variability of the model ensembles (TD) and are the min and max of the averaged result over the time period. All values are rounded to the nearest 0.1 kton CH4 and therefore columns do not necessarily add up. Bottom-up EU27+UK CH4 emissions Sector Data source Mean flux CH4 (kton)* BU Anthropogenic 1990-2013 2005-2011 2006-2012 2010-2016 Last available year** Energy UNFCCC NGHGI 5262.6 ± 1205.1 4022.6 ± 920.2 3938.7 ± 902 3641.2 ± 833.8 3398.5 ± 778.2 GAINS 4661.5 3336.5 3237.6 n.a. 2460.2 EDGAR v5.0 5438.4 (+2012.2; -1794.7) 4464.6 (+1651.9; -1473.3) 4401.7 (+1628.6; -1452.6) n.a. 4276.9 (+1582.4; -1411.4) IPPU UNFCCC NGHGI 69.6 ± 18.3 70.3 ± 19 68.1 ± 18.4 63.2 ± 17.1 63.4 GAINS n.a. n.a. n.a. n.a. n.a. EDGAR v5.0 26.2 (+10.2; -8.9) 26.7 (+10.4; -9.1) 26.2 (+10.2; -8.9) n.a. 25.0 (+9.8; -8.5) Agriculture UNFCCC NGHGI 10284.3 ± 998.8 9682.9 ± 992 9622.1 ± 985.3 9512.3 ± 974.1 9671.9 ± 17.1 GAINS 10791.3 9730.9 9632.6 n.a. 9441.3 EDGAR v5.0 10816.4 (± 1947) 10165.7 (± 1829.8) 10125.9 (± 1822.6) n.a. 10178.5 (± 1832.1) CAPRI 9915.3 9049.2 8975.9 n.a. 8834 FAOSTAT 10864.8 10067.5 9990.7 9814.4 9870.6 LULUCF UNFCCC NGHGI 278.5 ± 136.5 245.1 ± 118.3 244.7 ± 120.0 227.1 ± 111.3 320.6 ± 157.1 Waste UNFCCC NGHGI 8010.7 ± 1629.4 6735.9 ± 1549.3 6483.5 ± 1491.2 5564.1 ± 1279.7 5018.7 ± 1154.3 GAINS 8364.8 7691.1 7562.91 n.a. 6546.3 EDGAR v5.0 8792.9 (+3341.3; -2813.7) 7717.7 (+2932.7; -2469.6) 7501.7 (+2850.6; -2400.5) n.a. 6103.6 (+2319.4; -1953.2) Total anthropogenic BU - UNFCCC NGHGI 23905.7 ± 2220.8 20756.8 ± 1928.3 20357.2 ± 1891.2 19007.9 ± 1765.8 18473.1 ± 1716.1 BU natural CH4 emissions JSBACH-HIMMELI peatlands n.a. 1446.4 1423.0 1442.0 1345.9 Lakes_reservoires n.a. 2531.6 2531.6 n.a. 2531.6 Geological flux n.a. 1275.0 1275.0 1275.0 1275.0 TOTAL natural BU n.a. 5253 5229.6 2717 TD natural CH4 emissions GCP-CH4 wetlands from inversions n.a. 1519 (+4649.1; -462) 1486.5 (+4825.4; -464.3) 1355.9 (+5188.2; -431.5) 1248.1 (+2608.5; -272) Top-down EU27+UK total CH4 emissions TD regional total FLEXPART - FLExKF-TM5-4DVAR n.a. 30486.8 30047.3 27062.3 24594.83 TM5-4DVAR n.a. 28770.7 29308.9 29431.5 29144.0 FLEXINVERT_NILU n.a. 33190.6 32714.9 32434.1 31343.8 CTE-CH4 n.a. 32836.8 32213.3 30246.7 33483.5 InGOS inversions n.a. n.a. 29496 (+6115.4; -1305.1) n.a. 27467.5 (+1913; -4857.4) TD global total Total SURF n.a. 24702.1 (+10174.7; -5083.2) 24308.5 (+9593.2; -4529.1) 23719.0 (+9195.4; -4605.7) 26175.7 (+4798.9; -6474)) Total GOSAT n.a. n.a. n.a. 22689.3 (+8190.9; -3240.3) 22651.4 (+8511.5; -10759.0) *The three periods were chosen based on the availability of data. The common period between all data sets is 2006-2012. **Last available year as following: UNFCCC NGHGI 2017, EDGAR v5.0 2015, GAINS 2015, CAPRI 2013, FAOSTAT 2017, JSBACH-HIMMELI 2017, Lakes_reservoires 2011, geological (one value for 2005-2017), GCP-CH4 natural wetlands partition from TD 2015, FLEXPART - FLExKF-TM5-4DVAR 2017, FLEXINVERT_NILU 2017, CTE-CH4 2017, InGOS 2012, GCP ensemble total 2017, total SURF 2017, total GOSAT 2017. For details on model estimates and yearly values please download the data behind the figures on Zenodo https://doi.org/10.5281/zenodo.4590875

Table B1b: Comparison of N2O results from the BU and TD methods for different periods: BU Anthropogenic 1990-2015, TS total 2005-2014 and the common period between all data sets 2010-2014 and the last year available. All values are in kton N2O per year. The UNFCCC NGHGI uncertainties represent the 95% confidence interval; uncertainty for EDGAR v5.0 was calculated for 2015 and the min/max values for all sectors are as following: Energy: 12/250, IPPU: 13/19, Agriculture: 74/191, Waste: 63/166; the uncertainty represents the 95 % confidence interval of a lognormal distribution. The other uncertainties represent the variability of the model ensembles (TD) and are the min and max of the averaged result over the time period. All values are rounded to the nearest 0.1 kton N2O and therefore columns do not necessarily add up. Bottom-up EU27+UK N2O emissions Sector Data source Mean flux

N2O (kton)* BU Anthropogenic 1990-2013 2005-2014 Last year available** Energy UNFCCC NGHGI 102.6 ± 24.1 99.0 ± 23.2 97.9 ± 23.0 GAINS*** 85.3 100.1 97.7 EDGAR v5.0 90.3 (+225.75; -10.8) 86.1 (+215.3; -10.3) 77.9 (+194.8; -9.3) IPPU UNFCCC NGHGI 231.5 ± 37.1 103.1 ±16.5 37.1 ± 5.9 GAINS 259.9 147.8 69.5 EDGAR v5.0 234.5 (+44.5; -30.5) 155.7 (+29.6 ; -20.2) 141.7 (+26.9; -18.4) Agriculture**** UNFCCC NGHGI 636.4 ± 636.4 603.7 ± 603.7 627.7 ± 627.7 GAINS 704.7 665.7 669.8 EDGAR v5.0 612.3 (+1169.5; -453.1) 578.0 (+1104; -427.7) 587.7 (+1122.5; -434.9) CAPRI 637.9 n.a. 639.1 FAOSTAT 689.7 653.8 670.1 ECOSSE 429.0 425.6 386.0 DayCentˆ n.a. 643.9 ± 60.0 643.9 ± 60.0 LULUCF UNFCCC NGHGI 60.7 ± 29.7 64.9 ± 31.8 61.2 ± 30.0 Waste**** UNFCCC NGHGI 32.1 ± 32.1 34.0 ± 34.0 35.2 ± 35.2 GAINS 61.6 63.3 68.4 EDGAR v5.0 49.0 (+81.3; -30.9) 58.1 (+96.4; -36.6) 59.7 (+99.1; -37.6) Total BU anthropogenic UNFCCC NGHGI 1063.3 ± 853.6 904.7 ± 726.2 859.2 ± 689.7 BU natural N2O emissions Lakes, rivers, reservoirs n.a. 2.7 2.7 Top-down EU27+UK total N2O emissions TD total (2005-2014) MACTM-JAMSTEC (global) n.a. 1535.7 1577.3 PYVAR_NILU (global) n.a. 1024.2 1401.6 TOMCAT_LEEDS (global) n.a. 1369.7 1411.0 FLEXINVERT_NILU (regional) n.a. 1541.7 1228.5 Total TD min and max n.a. 1362.9 (+181.2; -340.3) n.a. *The three periods were chosen based on the availability of data. The common period between all data sets is 2010-2014. **Last available year as following: UNFCCC NGHGI (2019) 2017, EDGAR v5.0 2015, GAINS 2015, CAPRI 2013, FAOSTAT 2017, ECOSSE 2018, DayCent average 2011-2015, Lakes_rivers_reservoires (one value 2010-2014), FLEXINVERT_NILU 2017, TOMCAT_LEEDS 2014, PYVAR_NILU 2017, MACTM-JAMSTEC 2016. For details on model estimates and yearly values please download the data behind the figures on Zenodo https://doi.org/10.5281/zenodo.4590875 ***GAINS reports one value for every five years ****UNFCCC uncertainties for Agriculture and Waste were capped at 100%, but the actual reported values are much higher: 626% for Waste and 107 % for Agriculture. ˆDayCent 2011-2015

7. Table 1,2 and 3 all can go in supplementary material. These tables are just taking too much space in the manuscript and given that Table 3 is adopted from some other
paper, I do not think it should be in the main content. Thank you for your comment. We would like to keep these tables in the text as there are many data sets and we believe these tables offer from the beginning of the paper a good overview on data availability, periods covered and references. Table 3 indeed is an updated version from Petrescu et al 2020 AFOLU paper, and relates to the discussion on N2O; we moved it into Appendix B1, Table B1c.

8. Seasonal flux estimates: Currently, the manuscript estimates CH4 and N2O fluxes at annual time scale and completely ignore the fact that understanding seasonal dynamics of these fluxes are important and useful for climate mitigation efforts. At least, there should be an acknowledgement on why seasonal fluxes were not estimated.

The seasonality discussion is a very important point in estimating correctly the CH4 and N2O emissions, both anthropogenic and natural. This paper in its core is a comparison of scientific data with UNFCCC national submissions. Unfortunately, the latter does not take into account any seasonality from emissions, and we agree that in climate mitigation efforts these should be indeed taken up and sent as a message to the national inventory agencies for further improvement of their reporting. As NGHGIs follows the IPCC guidelines and apply consistently low-tier methods to estimating emissions and uncertainties for all countries, there is a long journey needed for such a change to happen, but we hope that this sort of synthesis will increase awareness in both scientific and political communities. Inventories like EDGAR or FAOSTAT do not account for seasonal changes. We will take on board this comment and aim to include in the next synthesis the seasonality simulated by some process models. We included in the conclusion (L769) a statement on acknowledging the importance of estimating seasonal fluxes, as follows. "Additionally, we advocate the need of analyzing the seasonality of emissions, which are of great importance for CH4 (wetland emission estimates have large uncertainties and show large variability in the spatial (seasonal) distribution ) and N2O (agriculture fertilizer application). This information is largely included in the prior flux estimates for TD approaches but not included in the reported IPCC guidelines. In

the climate mitigation process these seasonal variations may play an important role for a better quantification of sector specific uncertainties"

References: Etiope, G., Ciotoli, G., Schwietzke, S., and Schoell, M.: Gridded maps of geological methane emissions and their isotopic signature, Earth Syst. Sci. Data, 11, 1–22, https://doi.org/10.5194/essd-11-1-2019, 2019.

Hmiel, B., V. V. Petrenko, M. N. Dyonisius, C. Buizert, A. M. Smith, P. F. Place, C. Harth, R. Beaudette, Q. Hua, B. Yang, I. Vimont, S. E. Michel, J. P. Severinghaus, D. Etheridge, T. Bromley, J. Schmitt, X. Faïn, R. F. Weiss & E. Dlugokencky: Preindustrial 14CH4 indicates greater anthropogenic fossil CH4 emissions, Nature 578, 409–412, 2020.

Janssens-Maenhout, G., Pinty, B., Dowell, M., Zunker, H., Andersson, E., Balsamo, G., Bézy, J.-L., Brunhes, T., Bösch, H., Bojkov, B., Brunner, D., Buchwitz, M., Crisp, D., Ciais, P., Counet, P., Dee, D., Denier van der Gon, H., Dolman, H., Drinkwater, M., Dubovik, O., Engelen, R., Fehr, T., Fernandez, V., Heimann, M., Holmlund, K., Houweling, S., Husband, R., Juvyns, O., Kentarchos, A., Landgraf, J., Lang, R., Löscher, A., Marshall, J., Meijer, Y., Nakajima, M., Palmer, P., Peylin, P., Rayner, P., Scholze, M., Sierk, B., and Veefkind, P.: Towards an operational anthropogenic CO2 emissions monitoring and verification support capacity, B. Am. Meteorol. Soc., https://doi.org/10.1175/BAMS-D-19-0017.1, online first, 2020.